# Multigenerational paternal obesity enhances the susceptibility to male subfertility in offspring via *Wt1* N6-methyladenosine modification

Yong-Wei Xiong [1,2,7], Hua-Long Zhu[1,2,7], Jin Zhang[1,2,7], Hao Geng[3,4,5,7], Lu-Lu Tan[1,2], Xin-Mei Zheng[1,2], Hao Li[1,2], Long-Long Fan[1,2], Xin-Run Wang[1,2], Xu-Dong Zhang[1,2], Kai-Wen Wang[1,2], Wei Chang[1,2], Yu-Feng Zhang[1,2], Zhi Yuan[1,2], Zong-Liu Duan[3,4,5], Yun-Xia Cao[3,4,5], Xiao-Jin He [4,6] ✉, De-Xiang Xu [1,2,5] ✉ & Hua Wang [1,2,5] ✉

There is strong evidence that obesity is a risk factor for poor semen quality. However, the effects of multigenerational paternal obesity on the susceptibility to cadmium (a reproductive toxicant)-induced spermatogenesis disorders in offspring remain unknown. Here, we show that, in mice, spermatogenesis and retinoic acid levels become progressively lower as the number of generations exposed to a high-fat diet increase. Furthermore, exposing several generations of mice to a high fat diet results in a decrease in the expression of Wt1, a transcription factor upstream of the enzymes that synthesize retinoic acid. These effects can be rescued by injecting adeno-associated virus 9-*Wt1* into the mouse testes of the offspring. Additionally, multigenerational paternal high-fat diet progressively increases METTL3 and *Wt1* N6-methyladenosine levels in the testes of offspring mice. Mechanistically, treating the fathers with STM2457, a METTL3 inhibitor, restores obesity-reduced sperm count, and decreases *Wt1* N6-methyladenosine level in the mouse testes of the offspring. A case-controlled study shows that human donors who are overweight or obese exhibit elevated N6-methyladenosine levels in sperm and decreased sperm concentration. Collectively, these results indicate that multigenerational paternal obesity enhances the susceptibility of the offspring to spermatogenesis disorders by increasing METTL3-mediated *Wt1* N6-methyladenosine modification.

In the last 45 years (1973–2018), global semen quality has decreased by more than 62%[1]. Globally, the number of people with obesity and overweight are on the rise[2]. Body mass index (BMI) and the quality of semen were negatively correlated in population surveys[3,4]. Animal experiments also confirmed that a high-fat diet (HFD) induced testicular injury and spermatogenesis impairment in mice[5,6]. Paternal Origins of Health and Disease (POHaD) theory manifests that paternal exposure to adverse factors leads to the occurrence and development of adult offspring with chronic diseases[7,8]. Studies showed that paternal obesity impaired the structure of testicular seminiferous tubules

and reduced the number of epididymal sperm in offspring[9,10]. Based on the above studies, paternal obesity may lead to spermatogenesis disorders in offspring. Recent study found that parental obesity intergenerationally induced reproductive damages in offspring[11]. Nevertheless, the intergenerational effect of multigenerational paternal obesity on the susceptibility to spermatogenesis disorders in offspring and its mechanism remain unknown.

N6-methyladenosine (m6A) is the most common RNA modification, dynamically regulates post-transcriptional processes[12]. In mammals, the modification of m6A is conferred by demethylases (ALKBH5 and FTO) and methylases, including METTL3 and METTL4[13]. Increasing evidences have presented that homeostasis of m6A modification is essential for spermatogenesis in mammals[14–16]. An earlier study found that the expression of methylases in mouse testes gradually increased from embryo to adult, while the expression of demethylases gradually decreased[17]. We previously demonstrated that HFD markedly increased m6A modification levels in contemporary mouse testes[18]. However, the effect of paternal HFD on the level of m6A modification in offspring testes is unknown. It's well known that sperm plays an indispensable role in transmitting paternal phenotypes[19]. Previous studies found that sperm epigenetic modifications, such as small non-coding RNA, histone modification and DNA methylation, were involved in the intergenerational transmission of paternally acquired diseases[20–22]. It has not been clarified whether sperm m6A modifications contribute to multigenerational paternal obesity-induced susceptibility to spermatogenesis disorders in offspring.

In this work, we report that multigenerational paternal obesity enhances the susceptibility to male subfertility in offspring via Wilms tumor 1 (Wt1) m6A modification. A multigenerational paternal HFD mouse model was initially used to study the susceptibility of offspring to cadmium (Cd)-induced spermatogenesis disorder. Subsequently, we investigated the effect of WT1 overexpression on paternal susceptibility to spermatogenesis disorder in mouse offspring by injecting adeno-associated virus 9 (AAV9)-Wt1. STM2457, a specific METTL3 active inhibitor, was used to explore the role of sperm-derived modification of m6A in paternal HFD-downregulated testicular Wt1 expression in offspring. Lastly, the relationship among sperm m6A level, sperm concentration and BMI were verified in a human case-control study.

## Results

### Multigenerational paternal HFD progressively enhances susceptibility to spermatogenesis disorder in their offspring

The obesity phenotype was first identified in male mice. As presented in Supplementary Figs. 1a–c, body weight and epididymal white fat weight in male mice increased after HFD exposure. Furthermore, HFD exposure induced fasting blood glucose elevation, glucose tolerance impairment, hepatic lipid deposition and serum triglyceride (TG) increase in mice (Supplementary Figs. 1d–h). The effects of multigenerational paternal HFD exposure on the fertility rate, pregnancy rate and litter size in mice were then investigated. As presented in Fig. 1a–c and Supplementary Fig. 2a, the gradual reductions of fertility rate were observed in mice with the increase of HFD generation, with no effect on litter size. The effect of paternal HFD exposure on the susceptibility of environmental stress-induced spermatogenesis impairment in offspring was also explored. Figure 1d, e showed that sperm counts were obviously reduced in the HFD1D (the offspring were treated with Cd after paternal exposure to one-generational HFD) group, and persistently lowered in the HFD2D (the offspring were treated with Cd after paternal exposure to bi-generational HFD) group compared to that of the NC and NCD mice. The rates of sperm motility were evidently

decreased in HFD1D and HFD2D groups compared to the NCD group (Fig. 1f). Additionally, testicular HE staining showed that the gradual reductions of mature seminiferous tubules were observed in offspring with the increase of HFD generation (Fig. 1g, h). The above results indicate that multigenerational paternal HFD progressively enhances the susceptibility to spermatogenesis disorder in their offspring.

### Multigenerational paternal HFD progressively exacerbates environmental stress-impaired testicular germ cell development in offspring

It is well known that spermatogenesis is determined by the development of testicular germ cells. Compared to the NCD group, testicular weight and DDX4 (marker of testicular germ cell) protein level were obviously lowered in the HFD1D group, which was further reduced in the HFD2D group (Fig. 2a–c). Furthermore, the mRNA levels of Izumo3 (elongated spermatids marker), Acrv1 (round spermatids marker), Smc3 (spermatocytes marker) and C-kit (differentiating-spermatogonia marker) were reduced in HFD1D and HFD2D groups compared to NCD group, and Smc3, Acrv1 and Izumo3 mRNA levels were further decreased in HFD2D groups compared to HFD1D group (Fig. 2d). Correspondingly, the C-KIT and SYCP3 (spermatocytes marker) protein expressions were downregulated in the HFD1D and HFD2D groups compared with that of NCD group, and SYCP3 expression was persistently downregulated in the HFD2D group compared to the HFD1D group (Fig. 2e, f). Also, a gradual reduction of testicular SYCP3-positive cells was observed in the offspring with the increase of HFD generation (Fig. 2g, h). As mentioned, multigenerational paternal HFD progressively exacerbates environmental stress-impaired testicular germ cell development in offspring.

### Multigenerational paternal HFD progressively aggravates environmental stress-inhibited testicular retinoic acid synthesis in offspring

To investigate the mechanism by which paternal exposure to HFD exacerbates environmental stress-impaired testicular germ cell development in offspring, testicular RNA sequencing was performed in HFD1D and HFD2D groups. Compared with HFD1D group and HFD2D group, 229 mRNAs were upregulated and 268 mRNAs were downregulated, screened for a 1.2-fold change and adjusted with $P < 0.05$ (Supplementary Figs. 3a–c). As illustrated in Fig. 3a, b, GO analysis revealed that downregulated mRNAs were related to multiple biological processes, including the "retinol metabolic process", the "reproductive process", and the "spermatogenesis". The retinol metabolic-related genes were presented in supplementary Fig. 3d. Results showed that the expressions of Aldh1a1, Aldh1a2, Aldh1a3, Rarα and Stra8 were downregulated in the HFD2D group compared with the HFD1D group. Nevertheless, the levels of serum vitamin A (retinol) and testicular retinol-binding protein 4 (RBP4) were not reduced among the four groups (Supplementary Figs. 4a–c). Notably, a gradual decrease of testicular retinoic acid level was observed in the offspring with the increase of HFD generation (Fig. 3c). Further experiments showed that RARα and STRA8 proteins expression were progressively downregulated in offspring testes with the increase of HFD generation (Fig. 3d, e). To further explore and verify the expression of retinoic acid synthetase and retinoic acid metabolic enzyme in the four groups, the analysis of gene expression was conducted using RT-qPCR. The results presented that Aldh1a1, Aldh1a2 and Aldh1a3 mRNA levels were lowered in HFD1D and HFD2D groups, and Aldh1a1 and Aldh1a2 levels in HFD2D group were less than those in HFD1D group (Fig. 3f). Similarly, the gradual downregulation in the expression of

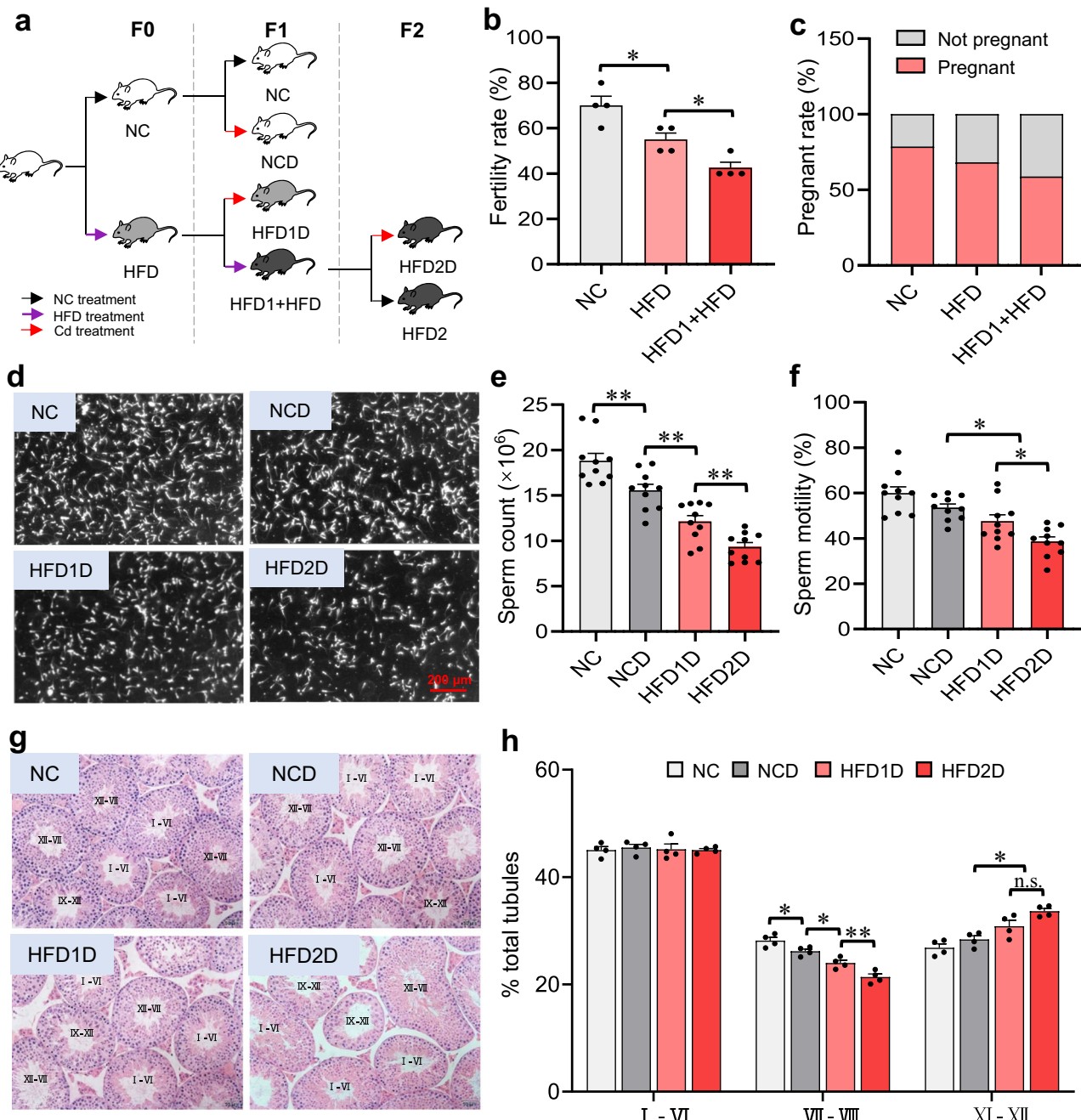

**Fig. 1 | Multigenerational paternal HFD enhances susceptibility to spermatogenesis disorder in offspring.** F0 generation male mice were fed NC or HFD from 5 weeks to 15 weeks old, and then mated with NC-fed female mice to breed F1 generation. Similarly, a subset of the males in F1 generation were continued to be treated with HFD for 10 weeks, and mated with normal female to breed F2 generation. Male mice of F1 and F2 generations were exposed to CdCl$_2$ for 10 weeks, and named NC, NCD, HFD1D or HFD2D group respectively. All mice were euthanized at 15 weeks of age. **a** Experimental design flowchart of Cd-induced spermatogenesis impairment in mice with HFD-feeding. The black arrow indicated NC treatment. The purple arrow indicated HFD treatment. The red arrow indicated Cd treatment. **b** The fertility rate was calculated. $n = 4$ independent experiments, Degree of Freedom (DOF) = 11, $F = 18.20$, $P = 0.0007$. **c** The pregnancy rate was counted. $n = 4$ independent experiments, $P = 0.3911$. **d**, **e** Epididymal sperm counts were measured. $n = 10$ mice, DOF = 39, $F = 37.68$, $P < 0.0001$. **f** Sperm motility was recorded; $n = 10$ mice, DOF = 39, $F = 14.01$, $P < 0.0001$. **g** Testicular H&E staining. **h** The number of Testicular seminiferous tubules number at different stages were evaluated. $n = 4$ mice, DOF = 15, $F = 30.25$, $P < 0.0001$. n.s. not significant. *$P < 0.05$; **$P < 0.01$. In regard to Fig. 1b, e, f, h, statistical significance was evaluated by two-sided one-way *ANOVA* with post hoc *LSD* tests. Data are presented as mean ± SEM. Source data are provided with this paper.

ALDH1A1 and ALDH1A2 proteins was observed in offspring testes with the increase of HFD generation (Fig. 3g–i). The compilation of these studies suggests that multigenerational HFD progressively exacerbates the inhibition of synthesis of testicular retinoic acid in offspring exposed to environmental stress.

## Multigenerational paternal HFD progressively exacerbates environmental stress-downregulated testicular WT1 expression in offspring

To investigate the potential mechanism for paternal HFD-inhibited testicular retinoic acid synthesis in offspring, the

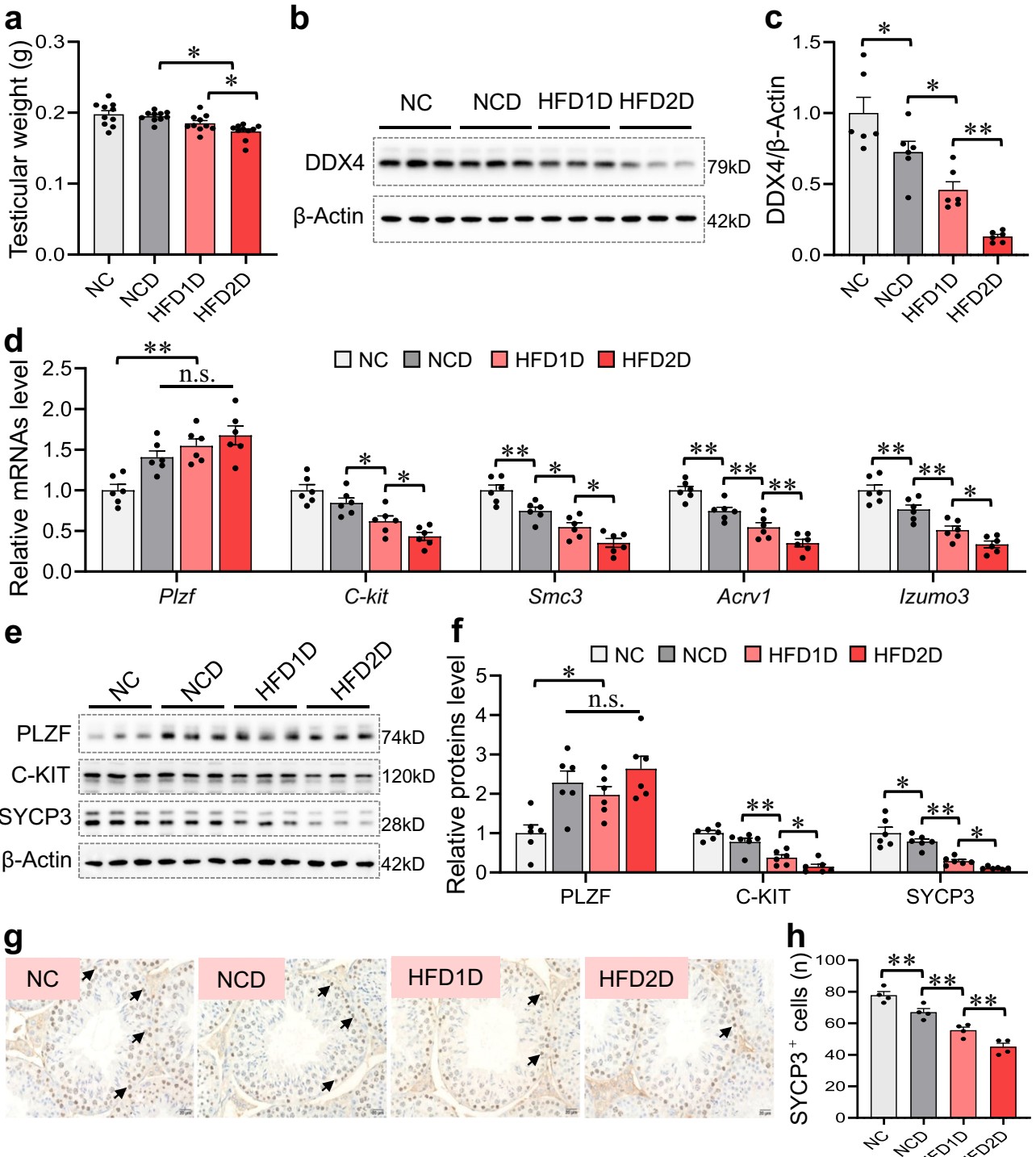

**Fig. 2 | Multigenerational paternal HFD exacerbates environmental stress-impaired testicular germ cell development in offspring.** F0 generation male mice were fed NC or HFD from 5 weeks to 15 weeks old, and then mated with normal female to breed F1 generation. Similarly, a subset of the males in F1 generation were continued to be treated with HFD for 10 weeks, and mated with normal female to breed F2 generation. Male mice of F1 and F2 generations were exposed to CdCl₂ for 10 weeks, and named NC, NCD, HFD1D or HFD2D group respectively. All mice were euthanized at 15 weeks of age. **a** Testicular weight. $n = 10$ mice, DOF = 39, $F = 8.12$, $P = 0.0003$. **b**, **c** Testicular DDX4 protein expression was detected by immunoblotting. $n = 6$ mice, DOF = 23, $F = 25.05$, $P < 0.0001$. **d** Testicular *Izumo3*, *Acrv1*, *Smc3*, *C-kit* and *Plzf* mRNA levels were detected by RT-qPCR. $n = 6$ mice, DOF = 23,

$F = 10.68$ and $P = 0.0002$ for *Plzf*; $F = 16.53$ and $P < 0.0001$ for *C-kit*; $F = 25.24$ and $P < 0.0001$ for *Smc3*; $F = 31.55$ and $P < 0.0001$ for *Acrv1*; $F = 29.16$ and $P < 0.0001$ for *Izumo3*. **e**, **f** Testicular PLZF, C-KIT and SYCP3 protein expression were detected by immunoblotting. $n = 6$ mice, DOF = 23, $F = 7.15$ and $P = 0.002$ for PLZF; $F = 27.78$ and $P < 0.0001$ for C-KIT; $F = 24.44$ and $P < 0.0001$ for SYCP3. **g** The expression of testicular SYCP3 was measured via immunohistochemistry. **h** SYCP3-positive cells were counted. $n = 4$ mice, DOF = 15, $F = 39.87$, $P < 0.0001$. n.s. not significant. *$P < 0.05$; **$P < 0.01$. In regard to Fig. 2a, c, d, f, h, statistical significance was evaluated by two-sided one-way *ANOVA* with post hoc *LSD* tests. Data are presented as mean ± SEM. Source data are provided with this paper.

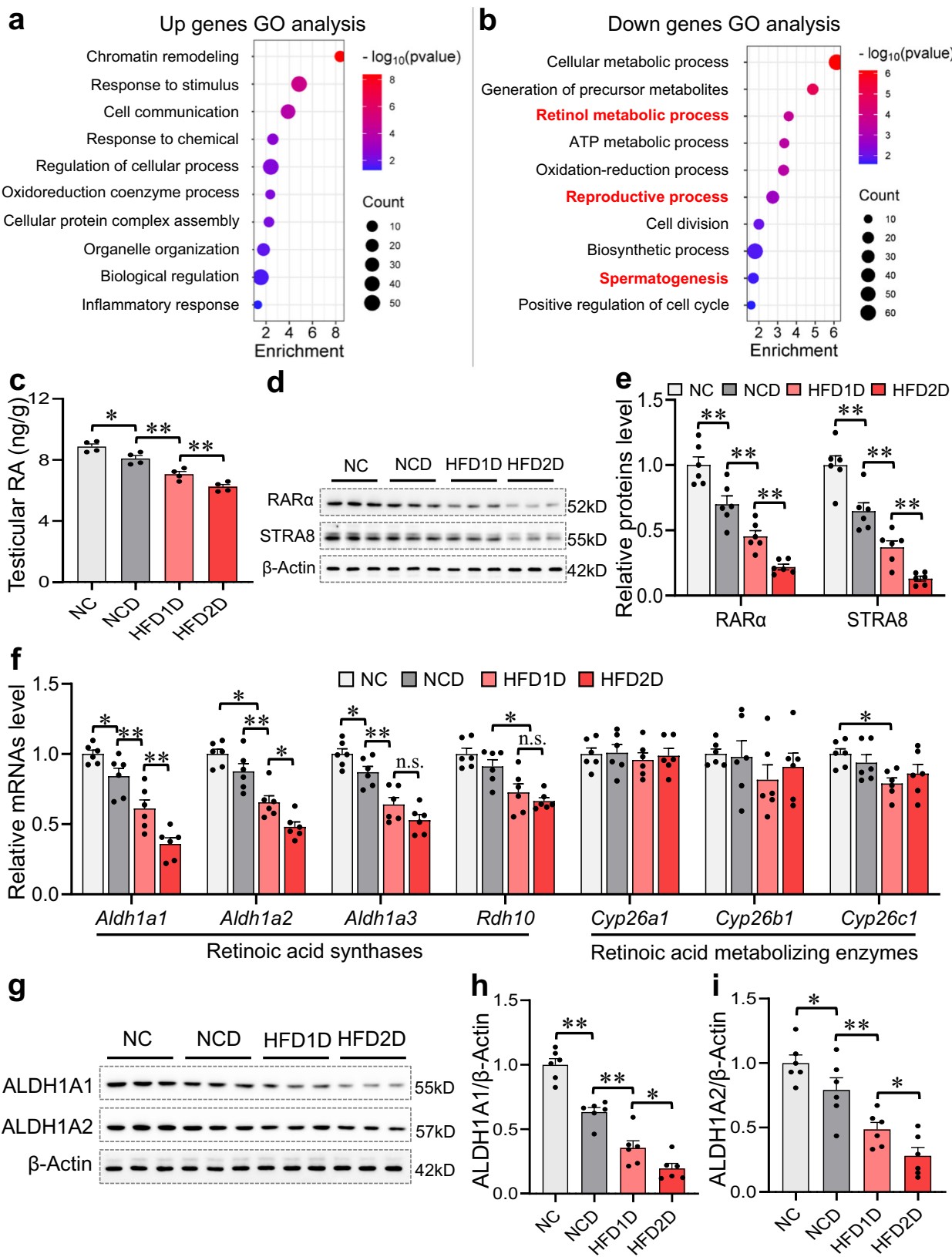

upstream transcription factors of *Aldh1a1*, *Aldh1a2* and *Aldh1a3* mRNAs were predicted from the ChEA3 database. Among all predictive transcription factors, Wilms tumor 1 (WT1) had the highest score (Fig. 4a). As shown in Fig. 4b–d, the mRNA and

protein levels of WT1 were significantly reduced in the HFD1D group, and persistently lowered in the HFD2D group compared to those of the NC and NCD mouse testes. In line with this, the gradual reductions of testicular WT1-positive cells were observed

**Fig. 3 | Multigenerational paternal HFD progressively aggravates environmental stress-inhibited testicular retinoic acid synthesis in offspring.** F0 generation male mice were fed NC or HFD from 5 weeks to 15 weeks old, and then mated with normal mice to breed F1 generation. Similarly, a subset of the males in F1 generation were continued to be treated with HFD for 10 weeks, and mated with normal female to breed F2 generation. Male mice of F1 and F2 generations were exposed to CdCl$_2$ for 10 weeks, and named NC, NCD, HFD1D or HFD2D group respectively. All mice were euthanized at 15 weeks of age. **a**, **b** GO enrichment analysis of differentially expressed mRNAs in mouse testes between HFD1D and HFD2D groups. **c** Testicular retinoic acid level was detected by ELISA. $n = 4$ mice, DOF = 15, $F = 41.13$, $P < 0.0001$. **d**, **e** Testicular RARα and STRA8 protein expressions were measured by immunoblotting. $n = 6$ mice, DOF = 23, $F = 43.08$ and $P < 0.0001$

for RARα; $F = 46.65$ and $P < 0.0001$ for STRA8. **f** Testicular *Aldh1a1*, *Aldh1a2*, *Aldh1a3*, *Rdh10*, *Cyp26a1*, *Cyp26b1* and *Cyp26c1* mRNA levels were tested by RT-qPCR. $n = 6$ mice, DOF = 23, $F = 32.26$ and $P < 0.0001$ for *Aldh1a1*; $F = 27.78$ and $P < 0.0001$ for *Aldh1a2*; $F = 25.82$ and $P < 0.0001$ for *Aldh1a3*; $F = 11.75$ and $P = 0.0001$ for *Rdh10*; $F = 0.18$ and $P = 0.9063$ for *Cyp26a1*; $F = 0.78$ and $P = 0.5217$ for *Cyp26b1*; $F = 3.20$ and $P = 0.0459$ for *Cyp26c1*. **g**–**i** Testicular ALDH1A1 and ALDH1A2 proteins were measured using immunoblotting. $n = 6$ mice, DOF = 23, $F = 61.35$ and $P < 0.0001$ for ALDH1A1; $F = 20.01$ and $P < 0.0001$ for ALDH1A2. n.s. not significant. *$P < 0.05$; **$P < 0.01$. In regard to Fig. 3c, e, f, h, i, statistical significance was evaluated by two-sided one-way *ANOVA* with post hoc *LSD* tests. Data are presented as mean ± SEM. Source data are provided with this paper.

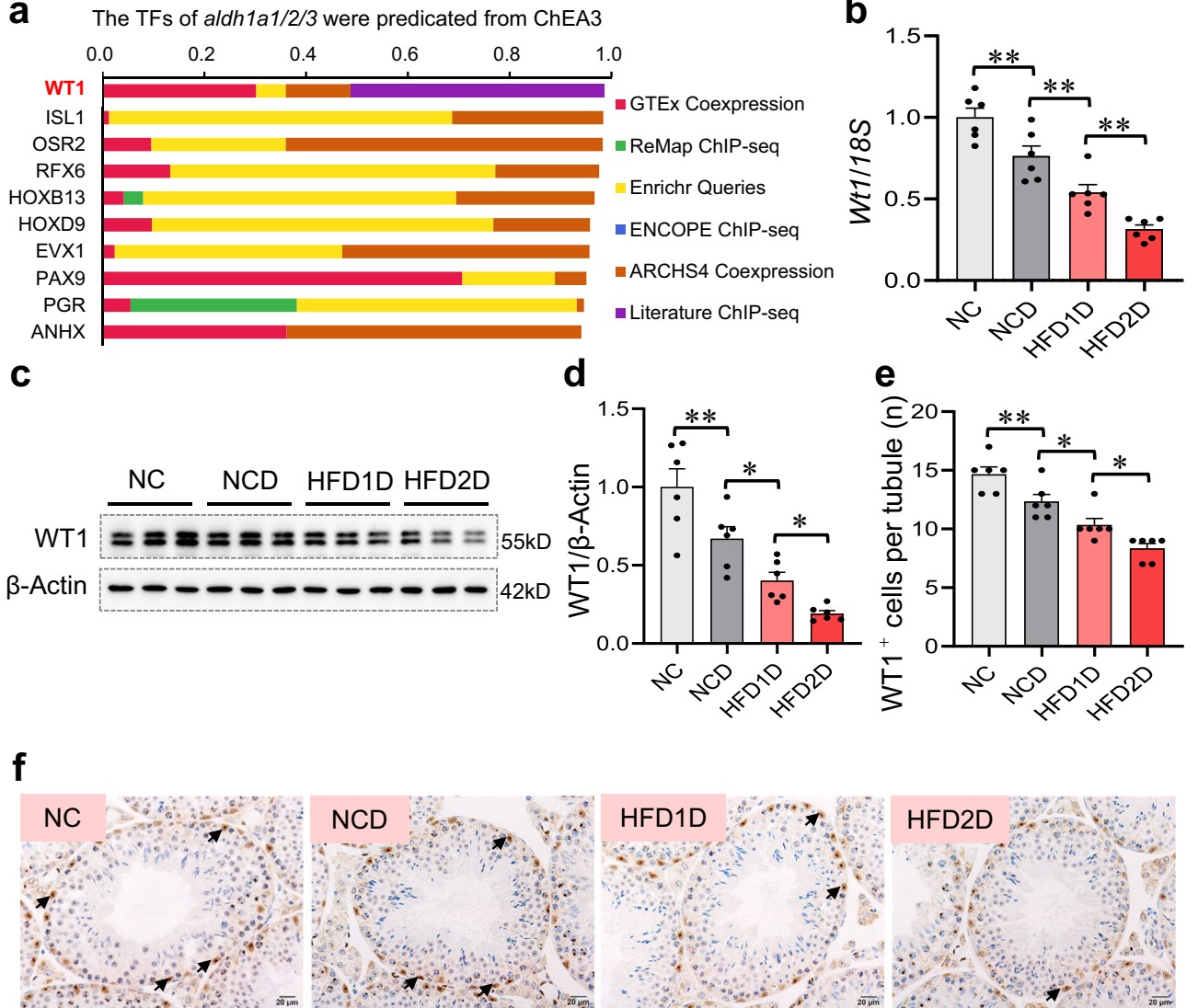

**Fig. 4 | Multigenerational paternal HFD progressively exacerbates environmental stress-downregulated testicular WT1 expression in offspring.** F0 generation male mice were fed NC or HFD from 5 weeks to 15 weeks old, and then mated with normal female to breed F1 generation. Similarly, a subset of the males in F1 generation were continued to be treated with HFD for 10 weeks, and mated with normal female to breed F2 generation. Male mice of F1 and F2 generations were exposed to CdCl$_2$ for 10 weeks, and named NC, NCD, HFD1D or HFD2D group respectively. All mice were euthanized at 15 weeks of age. **a** The upstream transcription factors of *Aldh1a1*, *Aldh1a2* and *Aldh1a3* were predicted from ChEA3

(https://maayanlab.cloud/chea3/). **b** Testicular *Wt1* mRNA level was tested using RT-qPCR. $n = 6$ mice, DOF = 23, $F = 33.75$, $P < 0.0001$. **c**, **d** Testicular WT1 protein expression was measured by immunoblotting. $n = 6$ mice, DOF = 23, $F = 21.59$, $P < 0.0001$. **e**, **f** The number of testicular WT1$^+$ positive cells per tubule were counted by immunohistochemistry. $n = 6$ mice, DOF = 23, $F = 23.66$, $P < 0.0001$. *$P < 0.05$; **$P < 0.01$. In regard to Fig. 4b, d, e, statistical significance was evaluated by two-sided one-way *ANOVA* with post hoc *LSD* tests. Data are presented as mean ± SEM. Source data are provided with this paper.

in the offspring with the increase of HFD generation (Fig. 4e, f). Collectively, these results suggest that multigenerational paternal HFD slows the expression of testicular *Wt1* mRNA and protein in offspring under environmental stress.

### Paternal HFD aggravates environmental stress-impaired testicular spermatogenesis via inhibiting WT1-mediated retinoic acid synthesis in offspring

To explore whether *Wt1* overexpression attenuates paternal HFD-impaired testicular spermatogenesis in offspring, a local testicular injection of AAV9-*Wt1* was performed in offspring whose fathers were exposed to HFD (Fig. 5a). As presented in Fig. 5b, c, sperm count was obviously reduced in HFD1D group compared to that in NCD group. As expected, *Wt1* overexpression markedly reversed HFD1D-induced reduction in sperm production in offspring (Fig. 5b, c). Figure 5d–f showed that *Wt1* overexpression attenuated HFD1D-downregulated the protein expression of RARα and STRA8 in offspring testes. Additionally, the reduced level of retinoic acid and ALDH1A1 protein were restored in HFD1D-treated testes after *Wt1* overexpression (Fig. 5g–i). Further studies found that *Wt1* overexpression restored HFD1D-reduced the level of WT1 in offspring testes (Fig. 5h, j and k). Therefore, paternal HFD aggravates environmental stress-impaired testicular spermatogenesis via inhibiting WT1-mediated retinoic acid synthesis in offspring.

### Multigenerational paternal HFD progressively exacerbates environmental stress-elevated testicular *Wt1* m6A level in offspring

The effect of paternal HFD exposure on the level of testicular m6A in offspring is shown in Fig. 6a. Testicular m6A level was evidently elevated in HFD1D group compared to NCD group, which was further augmented in the HFD2D group. Furthermore, METTL3 protein level was progressively increased in offspring testes over generations of HFD plus Cd stress, but not in METTL14, ALKBH5 and FTO (Fig. 6b, c, Supplementary Fig. 5a–d). The expression of m6A reading proteins was investigated in the follow-up studies. Compared to the NCD group, *Ythdf1* and *Igf2bp1* mRNA expressions were negatively regulated, while *Ythdf2* mRNA was positively regulated in offspring testes of HFD1D and HFD2D groups (Fig. 6d). The altered trend of YTHDF1, YTHDF2 and IGF2BP1 proteins was similar to those of their mRNAs among four groups (Fig. 6e, f). However, it was predicted that only IGF2BP1 had a binding effect on *Wt1* mRNA through the ENCORI database. As displayed in Fig. 6g, the four high credibility m6A modification sites in *Wt1* mRNA were predicted by the online tool SRAMP. The results of MeRIP-qPCR confirmed that a gradual increase in the m6A level of testicular *Wt1* site1 was observed in the offspring with the increase of HFD generation, but not site2, site3 and site4 (Fig. 6h, Supplementary Fig. 6a–c). Altogether, multigenerational paternal HFD progressively exacerbates environmental stress-elevated testicular *Wt1* site1 m6A level in offspring.

### Environmental stress decreases *Wt1* stability in an m6A-dependent manner in Sertoli cells

To verify the m6A modification regulation for environmental stress-downregulated *Wt1* mRNA, testicular Sertoli cells were selected for further experiments. As shown in Supplementary Fig. 7a, the dosage of Cd (20 μM) was selected based on a MTT assay. Supplementary Fig. 7b presented that the total m6A level was elevated in Sertoli cells after exposure to Cd. Additionally, METTL3 protein expression was upregulated, while IGF2BP1 and WT1 proteins expression were downregulated in Cd-treated Sertoli cells (Supplementary Figs. 7c–f). Figure 7a, b showed that *Wt1* mRNA level was reduced, and *Wt1* site1 m6A level was augmented in Cd-treated Sertoli cells. To determine whether Cd downregulated the expression of *Wt1* mRNA through the

methylated site1, luciferase reporters containing mutant (MUT) or wild-type (WT) *Wt1* were constructed. As presented in Fig. 7c, the modification of m6A was abrogated in MUT *Wt1*, due to the replacement of adenine (A) by thymine (T) in consensus sequences (RRACH). Compared with the WT sequence, the MUT sequence markedly reversed Cd-reduced *Wt1* level in Sertoli cells (Fig. 7d). Further studies found *Mettl3* siRNA (siR) pretreatment markedly restored Cd-elevated *Wt1* site1 m6A level, and reversed Cd-lowered WT1 protein and mRNA level in Sertoli cells (Fig. 7e–h). *Mettl3* siR obviously alleviated Cd-augmented the rate of *Wt1* mRNA degradation in actinomycin D-treated Sertoli cells (Fig. 7i). In the present experiment, *Igf2bp1* siR was further performed on Cd-treated Sertoli cells. The results presented that *Igf2bp1* siR exacerbated Cd-decreased WT1 protein level in Sertoli cells (Fig. 7j, k). Additionally, *Igf2bp1* siR markedly aggravated Cd-increased the rate of *Wt1* mRNA degradation in Sertoli cells (Fig. 7m). The IGF2BP1-RIP assay confirmed that the binding capacity of IGF2BP1 to *Wt1* mRNA was decreased after Cd exposure (Fig. 7n). Thus, the above results indicate that Cd induces the reduction of *Wt1* stability in an m6A-dependent manner.

### Multigenerational paternal HFD progressively increases m6A level and downregulates *Wt1* mRNA expression in paternal sperm

The effects of paternal HFD exposure on m6A modification in paternal sperm were investigated. As displayed in Fig. 8a, the gradual increases of m6A modification level were observed in paternal sperm with the increase of HFD generation. Figure 8b showed that sperm *Mettl3* mRNA expression was evidently upregulated in HFD group, which was further upregulated in HFD1 + HFD group. The MeRIP-microarray of sperm was then investigated in NC and HFD groups. GO and KEGG analysis suggest that hyper-methylated mRNAs were involved in multiple biological processes, including the "reproductive process", "gamete generation" and "spermatogenesis" (Fig. 8c, d). Furthermore, the sperm RNA-microarray revealed that 662 mRNAs increased and 530 mRNAs decreased after HFD consumption (Fig. 8e). The intersection between hyper-methylated mRNAs and downregulated mRNAs was identified. As shown in Fig. 8f, g, 72 overlapped mRNAs were found, and *Wt1* was one of them. The qPCR results confirmed that *Wt1* mRNA expression was progressively downregulated in the sperm with the increase of HFD generation (Fig. 8h). Overall, these results suggest that multigenerational paternal HFD progressively increases m6A level and downregulates *Wt1* mRNA expression in paternal sperm.

### Paternal HFD enhances testicular *Wt1* downregulation and spermatogenesis disorder in offspring via increasing METTL3-mediated paternal sperm m6A level

To further investigate the effect of reduced sperm m6A level on paternal HFD-impaired spermatogenesis in offspring, F0 generation male mice were treated with STM2457, a specific inhibitor of METTL3 activity. Results presented that the sperm count was reduced in HFD1D group, and the above effect was significantly restored after STM2457 treatment (Fig. 9a–c). As shown in Fig. 9d–f, the decreased protein and mRNA levels of *Wt1* in HFD1D testes were markedly reversed after STM2457 treatment. In addition, STM2457 treatment evidently restored HFD1D-increased the levels of *Wt1* site1 m6A and METTL3 protein in offspring testes (Fig. 9g–i). Further studies found that STM2457 treatment markedly attenuated HFD-increased m6A and *Mettl3* mRNA levels in paternal sperm (Fig. 9j, k). Furthermore, the decreased sperm level of *Wt1* mRNA in HFD group was also reversed after STM2457 treatment (Fig. 9k). The above results collectively demonstrate that paternal HFD aggravates Cd-induced testicular *Wt1* downregulation and spermatogenesis disorder in offspring via increasing METTL3-mediated paternal sperm m6A level.

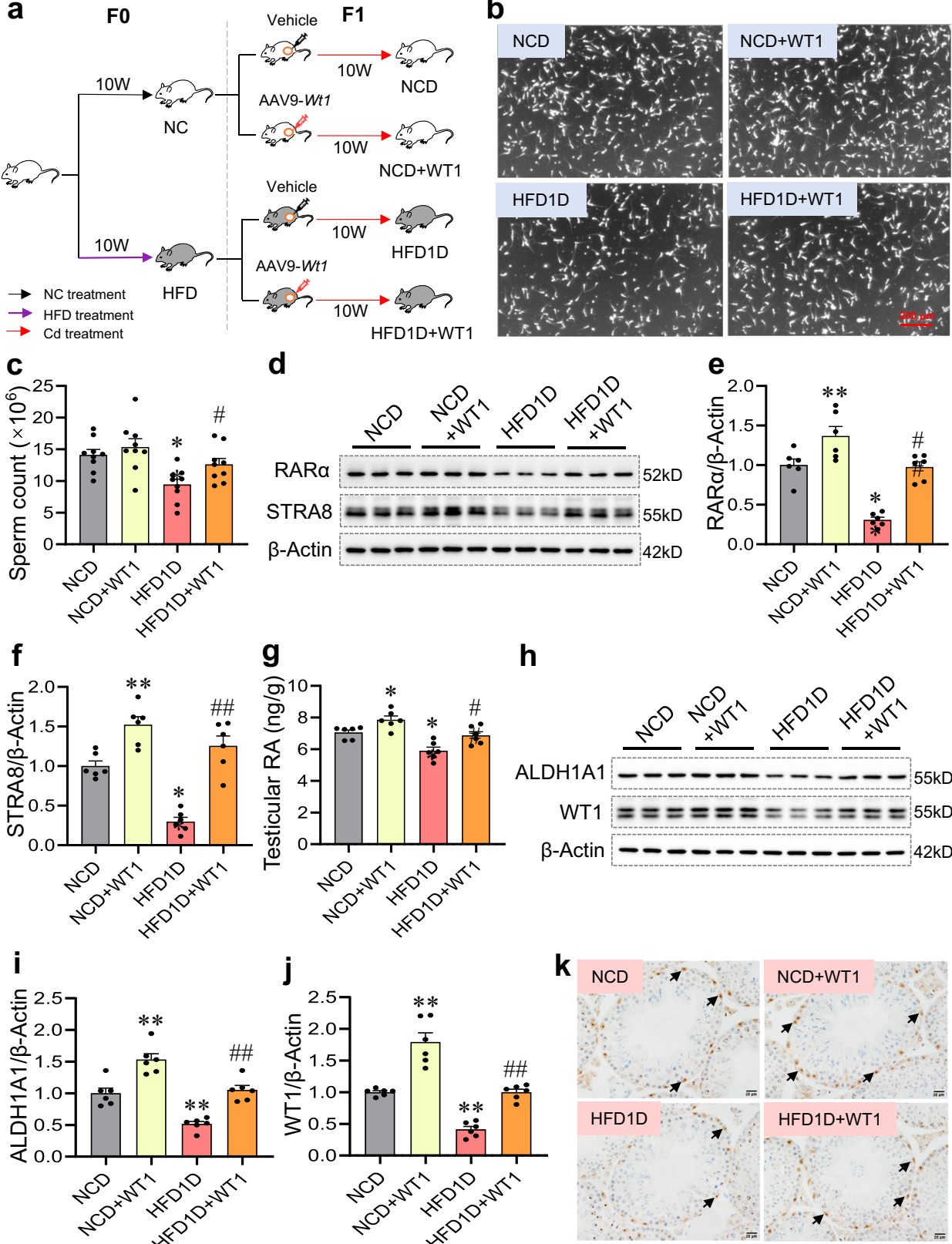

## Elevated sperm m6A level and decreased sperm concentration is observed in donors who were overweight or obese

The effects of paternal single- or double-generation HFD exposure on sperm and testicular parameters, retinoic acid level, WT1 and METTL3 expression in offspring mice were investigated. Analogously, the gradual reduction in sperm count, the impairment in testicular germ cell development, the inhibition in retinoic acid synthesis, the down-regulation in WT1 expression, the upregulation in METTL3 expression and the elevation in m6A level were observed in offspring testes with the increase of HFD generation (Supplementary Figs. 8–10). To further verify the relationship among sperm m6A level, sperm concentration and BMI, a case-control study was established. As presented in Fig. 10a,

**Fig. 5 | Paternal HFD aggravates environmental stress-impaired testicular spermatogenesis via inhibiting WT1-mediated retinoic acid synthesis in offspring.** F0 generation male mice were fed NC or HFD from 5 weeks to 15 weeks old, and mated with normal female to breed F1 generation. At postnatal day 35, *Wt1* was overexpressed in F1 mice by local testicular injection of adeno-associated virus 9 (AAV9). After injection of AAV9-*Wt1* or vehicle, mice were exposed to CdCl$_2$ for 10 weeks, and named NCD, NCD + WT1, HFD1D or HFD1D + WT1 group, respectively. All mice were euthanized at 15 weeks of age. **a** Experimental design flowchart. **b**, **c** Epididymal sperm counts were measured. $n = 9$ mice, DOF = 35, $F = 6.36$, $P = 0.0017$. **d**–**f** Testicular RARα and STRA8 protein expressions were measured by immunoblotting. $n = 6$ mice, DOF = 23, $F = 29.78$ and $P < 0.0001$ for RARα; $F = 33.45$ and $P < 0.0001$ for STRA8. **g** Testicular retinoic acid level was detected by ELISA. $n = 6$ mice, DOF = 23, $F = 13.88$, $P < 0.0001$. **h**–**j** Testicular ALDH1A1 and WT1 protein levels were measured by immunoblotting. $n = 6$ mice, DOF = 23, $F = 30.92$ and $P < 0.0001$ for ALDH1A1; $F = 48.60$ and $P < 0.0001$ for WT1. **k** Testicular WT1 expression was measured by immunohistochemistry. $n = 4$ mice. *$P < 0.05$; **$P < 0.01$ vs NCD. #$P < 0.05$; ##$P < 0.01$ vs HFD1D. In regard to Fig. 5c, e-g, i, j, statistical significance was evaluated by two-sided one-way *ANOVA* with post hoc *LSD* tests. Data are presented as mean ± SEM. Source data are provided with this paper.

b, the BMI of the case group was higher than that of the control group, but the sperm concentration was lower. Figure 10c showed that sperm m6A levels were elevated in the case group compared to the control group. The correlations of sperm m6A level and BMI or sperm concentration were further analyzed. The positive association was observed between sperm m6A level and BMI ($r = 0.57$, $P < 0.01$; Fig. 10e). The negative correlation was also observed between sperm m6A level and sperm concentration ($r = −0.51$, $P < 0.01$; Fig. 10f). In addition, restricted cubic splines were used to describe the nonlinear correlation among sperm concentration, BMI and sperm m6A level. As shown in Supplementary Figs. 11a–c, a linear relationship was observed between the three sets of data ($P$ for overall <0.001, and $P$ for nonlinear â 0.05). The above results suggest that elevated sperm m6A level and decreased sperm concentration is observed in donors who were overweight or obese.

## Discussion

A substantial body of evidence showed that early-life exposure to adverse factors impaired the reproductive development of male offspring[23,24]. Nevertheless, the intergenerational effects of paternal HFD exposure on susceptibility to spermatogenesis disorders in offspring is unknown. The present study revealed that fertility rate and sperm count were progressively reduced in the offspring with the increase of HFD generation. Additionally, mature testicular seminiferous tubules were progressively reduced in the offspring over generations of HFD plus Cd stress. Normal development of testicular germ cells is the premise for spermatogenesis, involving spermatogonia proliferation and differentiation, spermatocyte meiosis and spermatid deformation and elongation[25,26]. This study found that multigenerational paternal HFD persistently lowered testicular weight and testicular germ cell marker in offspring. During successive generations of HFD with Cd stress, offspring testes were progressively depleted of differentiating spermatogonia, spermatocytes, round spermatids, and elongated spermatids. In line with our findings, previous study found that paternal exposure to HFD decreased sperm count and motility in F1 offspring[9]. Taken together these findings suggest that multigenerational paternal HFD enhances susceptibility to testicular spermatogenesis disorder and germ cells impairment in their offspring.

Retinoic acid (RA), a retinol metabolite, plays a crucial role in testicular germ cells development and spermatogenesis[27,28]. In this study, the results of transcriptome sequencing suggested that downregulated mRNAs were related to retinol metabolism. The experimental results showed that multigenerational paternal HFD reduced the level of RA in offspring testes. Analogously, previous studies found that spermatogonia differentiation was blocked in rodents with RA deficiency, whereas normal spermatogenesis could be restored after RA supplementation[29,30]. In the testis, RA synthesis requires a two-step oxidative dehydrogenation reaction. The first step occurs under the action of retinol dehydrogenase RDH10 to form the retinal. Then, the retinal layer is further oxidized to RA by the retinal dehydrogenases ALDH1A1, ALDH1A2 and ALDH1A3[31,32]. Our results demonstrated that the gradual reductions in the level of *Aldh1a1*, *Aldh1a2*, and *Aldh1a3* were observed in offspring testes over generations of HFD plus Cd

stress. During the development of testicular germ cells, RA regulates the expression of its downstream target gene *Stra8* by binding to its receptor RARα[33]. The present study showed that RARα and STRA8 levels were progressively reduced in offspring with increasing HFD generation. In the context of Cd-impaired germ cell formation, multigenerational paternal HFD progressively diminishes the expression of retinal dehydrogenases.

Multiple transcription factors regulate the expression of *Aldh1a1*, *Aldh1a2* and *Aldh1a3*[34,35]. In this study, we discovered that WT1 has the highest score of any predicted transcription factor in the ChEA3 database. WT1 is located in Sertoli cells and participates in testicular development and steroidogenesis[36–38]. The current study found that multigenerational paternal HFD progressively reduced the mRNA and protein level of WT1 in offspring testes. Additionally, *Wt1* overexpression markedly reversed HFD1D-reduced sperm number, attenuated HFD1D-downregulated RARα and STRA8 expression, restored HFD1D-decreased RA and ALDH1A1 levels in offspring testes. An earlier study discovered that Sertoli cells *Wt1* knockout resulted in abnormal germ cell differentiation and meiosis in mouse testes[39]. Overall, the above results indicate that multigenerational paternal HFD progressively aggravates Cd-impaired testicular RA synthesis and spermatogenesis by inhibiting the *Wt1* mRNA expression.

Methylases (METTL3 and METTL14) and demethylases (ALKBH5 and FTO) regulate the balance of m6A in mammal[40–42]. m6A modification participates in the processes of testicular development and spermatogenesis[43,44]. This study found that multigenerational paternal HFD progressively exacerbated Cd-increased METTL3 and m6A in offspring testes. The results of the double luciferase reporter showed that Cd decreased the expression of *Wt1* mRNA by methylating modificatory site1. Furthermore, *Mettl3* siR effectively reversed the Cd-induced increase in the m6A level at *Wt1* site 1, alleviated the Cd-induced downregulation of *Wt1* in Sertoli cells. Reading proteins determine the function and fate of m6A-modified transcripts[45,46]. The present study found that YTHDF1 and IGF2BP1 were progressively downregulated, while YTHDF2 expression was upregulated in offspring testes with the increase of HFD generation. However, the prediction showed that only IGF2BP1 had binding effects on *Wt1* mRNA through the ENCORI database. IGF2BP1 was reported to maintain the stability of m6A-modified transcripts[47–49]. Our results showed that *Igf2bp1* siR markedly exacerbated Cd-induced degradation rate of *Wt1* mRNA in actinomycin D-treated Sertoli cells. The IGF2BP1-RIP assay confirmed that the binding ability of IGF2BP1 to *Wt1* mRNA was decreased after Cd exposure. These results suggest that multigenerational paternal HFD aggravates Cd-induced reduction of *Wt1* stability in an m6A-dependent manner in offspring testes.

Paternally acquired phenotypes are transmitted through sperm to offspring[19,50]. Previous studies found that paternal exposure to harmful factors impaired the health of their offspring by altering sperm DNA methylation and histone modification[51–53]. However, few studies have explored the role of sperm m6A modification in the transmission of paternal-inherited diseases. We found that *Mettl3* mRNA and m6A modification levels in sperm were progressively elevated with the increase of paternal HFD generation. The MeRIP-microarray results showed that hypermethylated mRNAs were

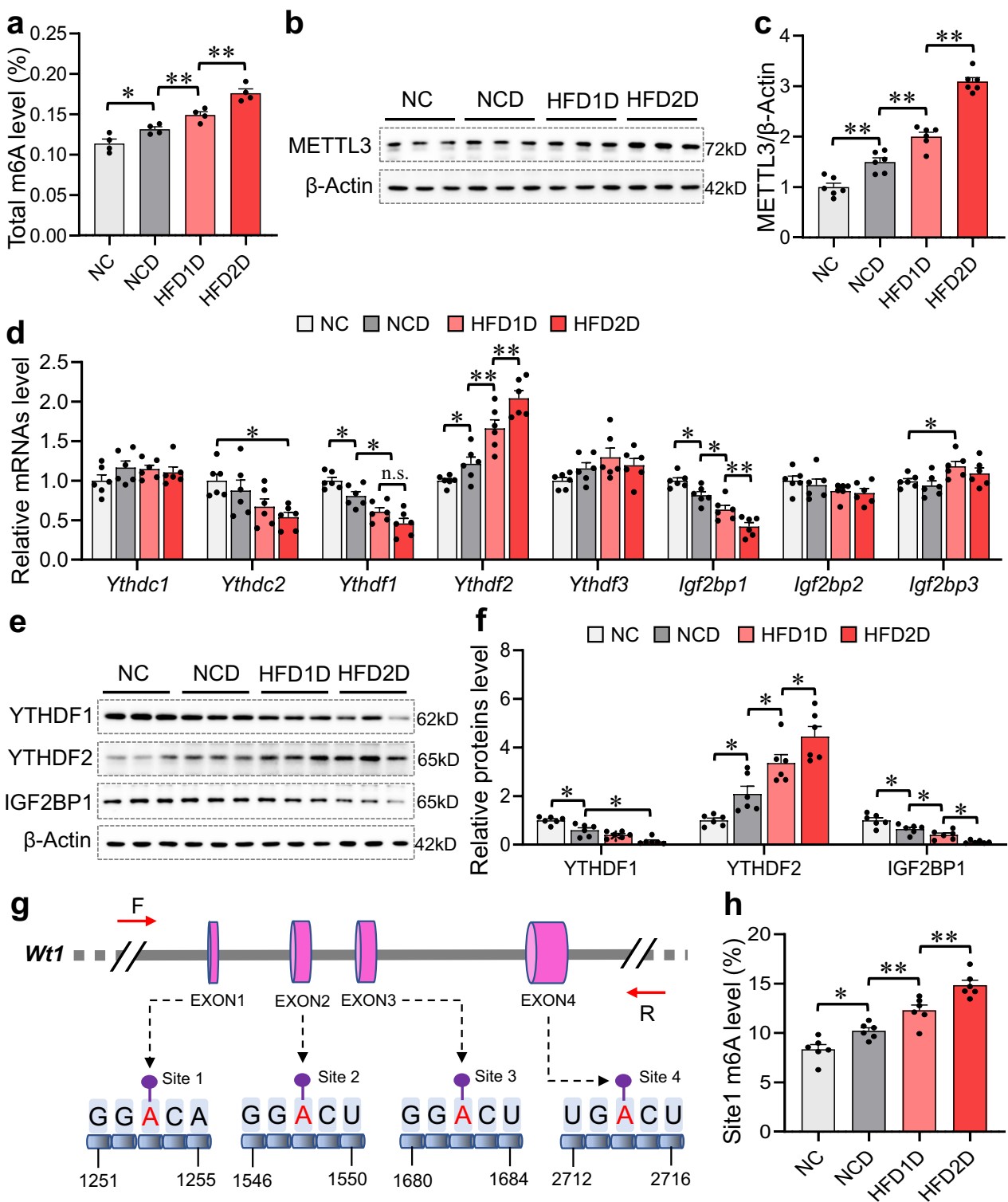

involved in multiple biological processes, including the "reproductive process", the "gamete generation" and the "spermatogenesis". Further experiments presented that paternal STM2457 treatment evidently restored HFD1D-reduced sperm count and *Wt1* level, reversed HFD1D-increased *Wt1* site1 m6A and METTL3 levels in offspring testes. An earlier study found that an increase in m6A modification in human sperm was associated with morbidity of asthenozoospermia[54]. In our case-control study, the elevated sperm m6A level and decreased sperm concentration was observed in donors with overweight/obesity. As thus, the above results suggest

that multigenerational paternal HFD progressively aggravates Cd-induced testicular *Wt1* downregulation and spermatogenesis disorder in offspring via METTL3-mediated sperm m6A.

In conclusion, our results suggest that multigenerational paternal obesity progressively enhances the susceptibility to spermatogenesis disorders in male offspring via downregulating *Wt1* mRNA expression, and METTL3-mediated sperm m6A modification is involved in paternal HFD-induced a decrease of *Wt1* mRNA stability in offspring testes (Fig. 10g). As result of these findings, it opens an avenue for mechanistic exploration of paternally acquired male subfertility and

**Fig. 6 | Multigenerational paternal HFD progressively exacerbates environmental stress-elevated testicular *Wt1* m6A level in offspring.** F0 generation male mice were fed NC or HFD from 5 weeks to 15 weeks old, and then mated with normal female to breed F1 generation. Similarly, a subset of the males in F1 generation were continued to be treated with HFD for 10 weeks, and mated with normal female to breed F2 generation. Male mice of F1 and F2 generations were exposed to CdCl$_2$ for 10 weeks, and named NC, NCD, HFD1D or HFD2D group respectively. All mice were euthanized at 15 weeks of age. **a** Testicular total RNA m6A level was measured. $n = 4$ mice, DOF = 15, $F = 23.52$, $P < 0.0001$. **b, c** Testicular METTL3 protein expression was detected by immunoblotting. $n = 6$ mice, DOF = 23, $F = 117.00$, $P < 0.0001$. **d** Testicular *Ythdc1*, *Ythdc2*, *Ythdf1*, *Ythdf2*, *Ythdf3*, *Igf2bp1*, *Igf2bp2* and *Igf2bp3* mRNA levels were tested by RT-qPCR. $n = 6$ mice, DOF = 23, $F = 1.14$ and $P = 0.3565$ for *Ythdc1*; $F = 4.49$ and $P = 0.0145$ for *Ythdc2*; $F = 18.81$ and $P < 0.0001$ for *Ythdf1*; $F = 29.71$ and $P < 0.0001$ for *Ythdf2*; $F = 2.15$ and $P = 0.1259$ for *Ythdf3*; $F = 28.11$ and $P < 0.0001$ for *Igf2bp1*; $F = 1.46$ and $P = 0.2566$ for *Igf2bp2*; $F = 3.71$ and $P = 0.02858$ for *Igf2bp3*. **e, f** Testicular YTHDF1, YTHDF2 and IGF2BP1 protein expressions were detected by immunoblotting. $n = 6$ mice, DOF = 23, $F = 26.49$ and $P < 0.0001$ for YTHDF1; $F = 22.44$ and $P < 0.0001$ for YTHDF2; $F = 23.94$ and $P < 0.0001$ for IGF2BP1. **g** The four m6A modification sites in *Wt1* mRNA were predicted by online tool SRAMP. **h** *Wt1* site1 m6A level was measured by MeRIP-qPCR. $n = 6$ mice, DOF = 23, $F = 34.76$, $P < 0.0001$. n.s. not significant. *$P < 0.05$; **$P < 0.01$. In regard to Fig. 6a, c, d, f, h, statistical significance was evaluated by two-sided one-way *ANOVA* with post hoc *LSD* tests. Data are presented as mean ± SEM. Source data are provided with this paper.

intergenerational transmission, including m6A modification and *Wt1* expression. Among them, sperm m6A modification may be one of the genetic markers for early screening of paternally acquired male subfertility.

## Methods

### Ethics

All animal treatments were approved by the Laboratory Animal Ethics Committee of Anhui Medical University (ethical approval number: LLSC20220640 and LLSC20190297). The case-control study design complied with all relevant regulations regarding the use of human study participants and was conducted in accordance to the criteria set by the Declaration of Helsinki. This was approved by the Clinical Medical Research Ethics Review Committee of the First Affiliated Hospital of Anhui Medical University (ethical approval number: PJ2023-04-12) and obtained informed consent from the donor.

### Animal treatments

The C57BL/6 N mice were provided by Beijing Vital River Laboratory Animal Technology Co., Ltd (Beijing, China). All animals were accustomed to standard conditions (20–25 °C environment temperature, 50–60% air humidity and 12-h light/dark cycle) for a week. The purified control feed (TP23302) and 60% fat high-fat feed (TP23300) were acquired from Trophic Animal Feed High-Tech Co., Ltd (Nantong, China). In the different generations, 8 to 11 pregnant mice were obtained in each group after mating with different male mice, and the balances of sexes between pups (male: female, 3: 3) were performed on postnatal day (PND) 1. At PND28, one male mouse from each litter was selected. For euthanasia, all mice were injected intraperitoneally with 2,2,2-tribromoe-thanol (250 mg/kg), and then cervical dislocation was executed under anesthesia.

Experiment 1. The experimental design used to investigate the susceptibility to spermatogenesis disorders in the offspring of fathers stressed by multigenerational HFD is shown in Fig. 1a. Briefly, 20 F0 generation male mice were fed with HFD or normal chow (NC) from 5 weeks to 15 weeks old, and then mated with normal female mice to breed F1 generation. After mating, 10 pregnant mice were obtained in each group. Similarly, 10 F1 generation male mice (whose fathers were exposed to HFD) from different litter continued to be fed HFD for 10 weeks, and mated with normal females to breed F2 generation. Then, 10 male mice from each group of F1 and F2 generations were exposed to CdCl$_2$ (0 or 100 mg/L) by drinking water for 10 weeks, and named NC (normal control), NCD (the offspring were treated with Cd after paternal treatment with normal chow), HFD1D (the offspring were treated with Cd after paternal exposure to one-generational HFD) or HFD2D (the offspring were treated with Cd after paternal exposure to bi-generational HFD) groups, respectively. Cd, a well-known reproductive toxicant, impairs testicular development and spermatogenesis. Population-based studies found that Cd concentrations in blood and semen were negatively correlated with sperm quality[55,56]. A series of animal experiments also confirmed that Cd exposure impaired testicular

development and reduced sperm count in mice[18,57]. Here, Cd was used as an inducer of reproductive toxicity to induce spermatogenesis disorders in mice. It's well-known that Cd exposure impairs testicular spermatogenesis mainly via the dose of internal exposure. In the present study, the dosage of 100 mg/L CdCl$_2$ was chosen according to the internal exposure level in population studies and previous research[18,58]. Our pervious study showed the sera Cd concentration when mice were exposed to CdCl$_2$ (100 mg/L) through drinking water was close to the blood Cd of patients with low semen quality and was related to Cd level of human in severe Cd-contaminated areas and smokers[58,59]. Meanwhile, single- and double-generations of HFD-fed fathers named HFD and HFD1 + HFD group, and their offspring named HFD1 and HFD2 group, were created. Euthanasia was performed in all groups at 15 weeks of age. The testes, epididymal sperm and serum from mice were obtained for further analysis.

Experiment 2. The experimental design used to explore the effect of WT1 overexpression on multigenerational paternal HFD-enhanced susceptibility to spermatogenesis disorders in offspring, as presented in Fig. 5a. In summary, F0 generation male mice were fed with HFD or NC from 5 weeks to 15 weeks old, and mated with normal female mice to breed F1 generation. After mating, 9 pregnant mice were obtained in each group. At PND35, 9 F1 generation mice from each group overexpressed WT1 or NC by local testicular injection of adeno-associated virus 9 (AAV9). After injection of AAV9-*Wt1* or *NC*, mice were exposed to CdCl$_2$ for 10 weeks, and named the NCD, NCD + WT1, HFD1D or HFD1D + WT1 group, respectively. All mice were euthanized at 15 weeks of age. Mouse epididymal sperm and testes were obtained.

Experiment 3. To investigate the effect of reduced sperm m6A level on paternal HFD-inhibited testicular WT1 expression and spermatogenesis in offspring, F0 generation male mice were treated with STM2457 by local testicular injection. As shown in Fig. 9a, F0 generation male mice were fed with HFD or NC from 5 weeks to 15 weeks old, and treated with STM2457 once a week from 10 weeks to 15 weeks old. After mating, 8–11 pregnant mice were obtained in each group. Then, 8–11 F1 generation male mice from each group were exposed to CdCl$_2$ from 5 weeks to 10 weeks old, and named NCD, NCD + STM, HFD1D or HFD1D + STM group, respectively. Mouse serum, epididymal sperm and testes were obtained for further analysis.

### Case-control study

To investigate the effect of overweight/obesity on sperm m6A levels, a case-control study was established. A total of 428 human sperm were obtained from the reproductive medicine center of the First Affiliated Hospital of Anhui Medical University with the donor's informed consent. After removal of smoking or alcohol drinking donors, 168 sperm samples were available. Finally, 30 pairs of cases with overweight/obesity and corresponding controls were obtained by matching age. Semen specimens were centrifuged at 600 g for 10 min to remove seminal plasma. After PBS resuspension, sperm were incubated in somatic cell lysate (0.5% Triton X and 0.1% sodium dodecyl sulfate in

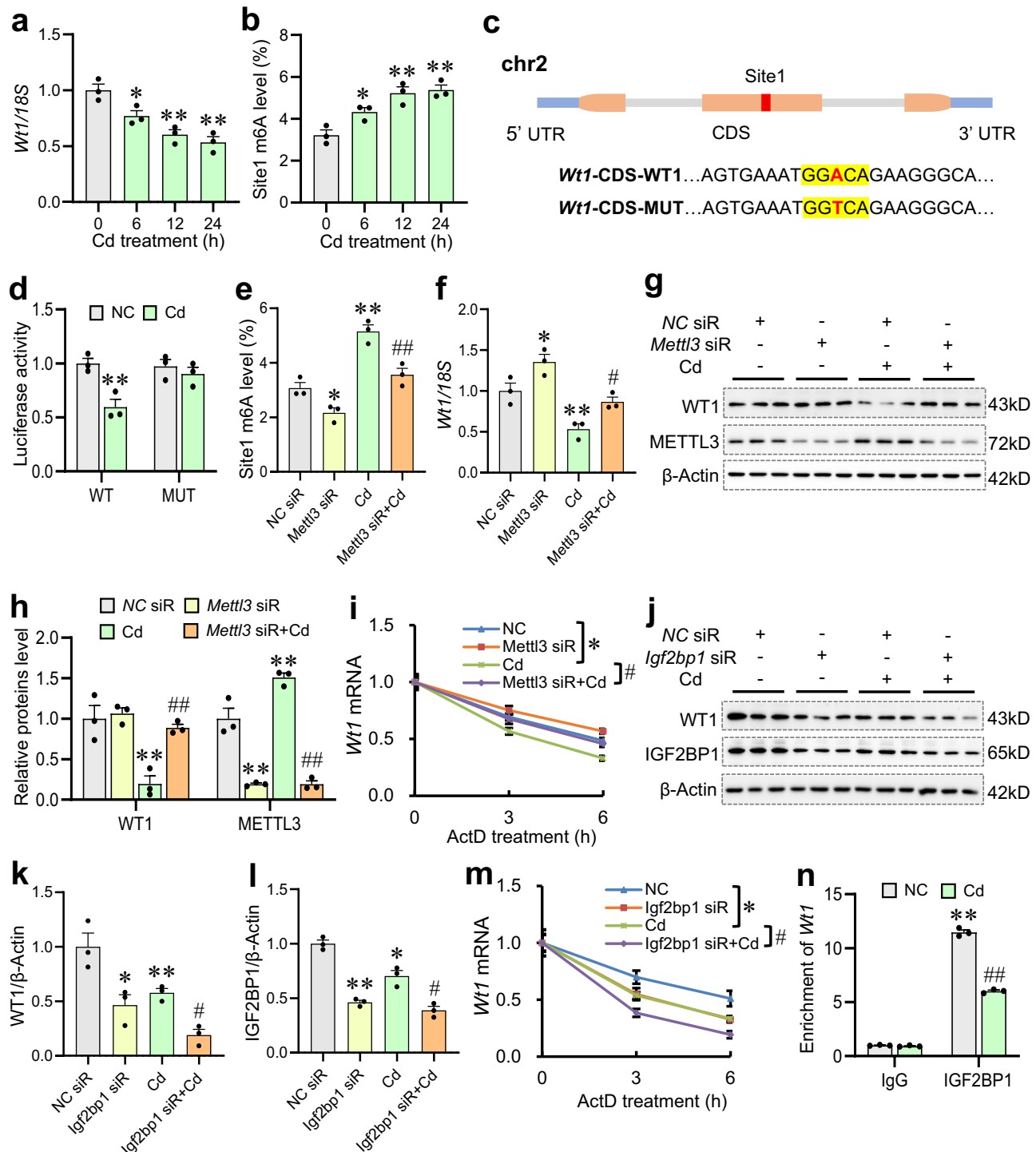

nuclease-free water) at 4 °C for 10 min to remove somatic cells. After washing, sperm pellets were collected for m6A detection.

## Cell culture

TM4 cells, the normal mouse testicular Sertoli cell lines, were from the Cell Bank of the Chinese Academy of Sciences (catalog number: GNM41). TM4 cells were cultured using DMEM/F-12 medium and supplemented with 5% horse serum, 2.5% fetal bovine serum, and 1% penicillin-streptomycin solution. To determine the dose of Cd-impaired TM4 cell growth, cells were exposed to $CdCl_2$ at different concentrations, and cell viability was detected by the MTT assay at different time points. To investigate the effects of Cd exposure on WT1

expression and m6A modification level, TM4 cells were exposure to $CdCl_2$ for 0–24 h. To investigate the effect of *Mettl3* or *Igf2bp1* siR on Cd-downregulated WT1 expression, cells were exposure to $CdCl_2$ for 24 h after *Mettl3* or *Igf2bp1* siR treatment. To determine whether Cd downregulated *Wt1* mRNA expression by methylating site1, cells were exposure to $CdCl_2$ for 24 h after wild type (WT)-*Wt1* or mutant (MUT)-*Wt1* plasmids treatment.

## Fertility test

For fertility rate, one male mouse was mated with two female mice at night, and vaginal plug were checked in the next morning. The number of fertile male mice was recorded and divided by the total number of

**Fig. 7 | Environmental stress decreases *Wt1* stability in an m6A-dependent manner in Sertoli cells. a, b** Treatment of TM4 cells with $CdCl_2$ (20 μM) for 0, 6, 12 or 24 h. **a** Cellular *Wt1* mRNA expression was detected by RT-qPCR. $n = 3$ biologically independent samples, DOF = 11, $F = 17.24$, $P = 0.0007$. **b** *Wt1* site1 m6A level was measured using MeRIP-qPCR. $n = 3$ biologically independent samples, DOF = 11, $F = 15.54$, $P = 0.0011$. **c** Schematic representation of m6A motif and mutation position in CDS within *Wt1* mRNA were presented. **d** The luciferase activities of the mutant and wild-type *Wt1*-CDS were measured after Cd exposure by luciferase reporter assay. $n = 3$ biologically independent samples, DOF = 11, $F = 8.79$, $P = 0.0065$. **e–h** Treatment of TM4 cells with $CdCl_2$ for 24 h after *Mettl3* siR. **e** *Wt1* site1 m6A level was detected. $n = 3$ biologically independent samples, DOF = 11, $F = 33.43$, $P < 0.0001$. **f** *Wt1* mRNA expression was measured. $n = 3$ biologically independent samples, DOF = 11, $P = 0.0006$, $F = 18.46$. **g, h** WT1 and METTL3 protein levels were detected via immunoblotting. $n = 3$ biologically independent samples, DOF = 11, $F = 15.21$ and $P = 0.0011$ for WT1; $F = 78.05$ and $P < 0.0001$ for METTL3.

**i** *Wt1* mRNA level was measured after treating with ActD for 3 or 6 h using RT-qPCR. $n = 3$ biologically independent samples, $F = 8.00$, $P = 0.0001$. **j–l** Treatment of TM4 cells with $CdCl_2$ for 24 h in the within or without of *Igf2bp1* siR. WT1 and IGF2BP1 protein levels were detected via immunoblotting. $n = 3$ biologically independent samples, DOF = 11, $F = 22.13$ and $P = 0.0003$ for WT1; $F = 10.07$ and $P = 0.0043$ for IGF2BP1. **m** *Wt1* mRNA level was detected after treating with ActD for 3 or 6 h using RT-qPCR. $n = 3$ biologically independent samples, $F = 5.04$, $P = 0.0051$. **n** The binding ability of IGF2BP1 and *Wt1* was evaluated after Cd exposure by RIP assays. $n = 3$ biologically independent samples, DOF = 11, $F = 75.36$, $P < 0.0001$. *$P < 0.05$; **$P < 0.01$ vs NC. #$P < 0.05$; ##$P < 0.01$ vs Cd. In regard to Fig. 7a, b, d–f, h, k–l, n, statistical significance was evaluated by two-sided one-way *ANOVA* with post hoc *LSD* tests. In regard to Fig. 7i, m, statistical significance was evaluated by two-sided two-way *ANOVA*. Data are presented as mean ± SEM. Source data are provided with this paper.

male mice to calculate the fertility rate of each group on that day. This test was repeated twice again every 4 days, a total of 4 repetitions. For pregnant rate, female mice with vaginal plug were fed at single caged, and the pregnancy results were recorded. The litter sizes were also observed after delivery.

### Glucose tolerance test (GTT)
The F0 generation male mice were intraperitoneally injected with 2.0 g/kg glucose solution after fasting (maintaining normal water intake) for 12 h. The blood glucose was then detected in the caudal vein using a glucometer, and recorded at different time points.

### Local testicular injection
Local testicular injections were performed according to previous study with corresponding modification[60]. In brief, the mice were injected intraperitoneally with 250 mg/kg 2,2,2-tribromoe-thanol. After anesthesia, both testes of the mice were found by touch pressure. Then, 20 μL AAV9 or STM2457 were injected into the middle of each testes using a micropipette. After injection, the mice were placed flat in cage and observed for 30 min until they were fully awake and returned to normal. Among them, $1.03 \times 10^{12}$ vg/ml AAV9-*Wt1* were injected once into mice at PND35. The control was injected with vehicle. AAV9-*Wt1* and vehicle were synthesized by General Biol Co., Ltd. STM2457 were injected at a dose of 50.0 mg/kg, once a week for 5 weeks. 20% SBE-β-CD was used as control. STM2457 and SBE-β-CD were purchased from MedChemExpress. The dose of STM2457 was chosen according to previous research[61].

### Evaluation of sperm count
Sperm motility and count were evaluated by an automatic sperm analyzer (Hamilton Thorne, USA). Mouse left epididymis was sectioned into slices and placed in a preheated medium. The medium was then incubated at 37 °C for 5 min until sperm was completely released. In the final step, the sperm suspension was diluted with preheated medium, and 50 μL of suspension was dropped into a slide for parameter evaluation.

### Sperm isolation
To obtain sperm, fresh mouse epididymis was completely cut into pieces in preheated medium and incubated at 37 °C for 10 min until mature sperm were fully released. Then, the tissue debris were removed by 40-μm cell strainer. The sperm mixture was incubated with somatic cell lysate (0.5% Triton X and 0.1% sodium dodecyl sulfate in nuclease-free water) on ice for 40 min to clear somatic cell. Next, the mixture was centrifuged at 600 g for 5 min and the precipitates were resuspended with PBS. After washing twice, sperm pellets were collected for the methylated RNA microarray.

### MTT assay
First, TM4 cells were treated with different concentrations $CdCl_2$ for 6, 12, or 24 h after culture in 96 well plates for 12 h. Following that, the cells were treated with 5 mg/ml MTT, and were persistently cultured for 2 h in a 37 °C incubator. Prior to this, the cells were grown in DMEM/F-12 medium and supplemented with 5% horse serum, 2.5% fetal bovine serum and 1% penicillin-streptomycin solution. After aspirating the medium, DMSO was added to the culture plate. As a final step, the cellular absorbance at 570 nm was detected by a multimode reader (SYNERGY4, Biotek, USA).

### Immunohistochemistry
Dewaxing, rehydrating, quenching endogenous peroxidase, and recovery of antigens were performed in testicular sections before immunostaining. The antibodies specific for SYCP3 and WT1 were subsequently incubated with sections overnight. On the second day, secondary antibody incubation, diaminobenzidine-staining, and hematoxylin-counterstaining were performed, respectively. An Olympus BX53F microscope was used to count positive cells in the testes.

### Immunoblotting
The protein lysates from TM4 cells and mouse testes were obtained after homogenizing with RIPA buffer, and Immunoblotting was implemented as previously described[62]. Briefly, cellular and testicular protein extracts (6–30 ug) were separated by 10.0–12.5% SDS-PAGE electrophoresis and then transferred into PVDF membrane. After blocking with 5% skim milk for 1.5 h, the membranes were incubated with antibodies of rabbit anti-DDX4 (1:1000, Abcam, ab13840), mouse anti-PLZF (1:200, Santa Cruz Biotechnologies, sc-28319), mouse anti-C-KIT (1:200, Santa Cruz Biotechnologies, sc-365504), mouse anti-SYCP3 (1:200, Santa Cruz Biotechnologies, sc-74569), rabbit anti-STRA8 (1:1000, Abcam, ab49602), rabbit anti-RARα (1:1000, Cell Signaling Technology, 62294 S), rabbit anti-RBP4 (1:1000, Abcam, ab188230), rabbit anti-ALDH1A1 (1:1000, Abcam, ab52492), rabbit anti-ALDH1A2 (1:1000, Cell Signaling Technology, 83805 S), rabbit anti-WT1 (1:1000, Abcam, ab89901), rabbit anti-METTL3 (1:2000, Abcam, ab195352), rabbit anti-METTL14 (1:1000, Cell Signaling Technology, 51104 S), rabbit anti-ALKBH5 (1:1000, Abcam, ab195377), mouse anti-FTO (1:1000, Abcam, ab92821), rabbit anti-YTHDF1 (1:1000, Proteintech, 17479-1-AP), rabbit anti-YTHDF2 (1:1000, Proteintech, 24744-1-AP) and rabbit anti-IGF2BP1 (1:1000, Proteintech, 22803-1-AP) for 1–3 h at room temperature. Antibody of β-Actin (1:10000) for loading control. After washing, the corresponding secondary antibody were used to incubate membranes for 1–2 h. Finally, the imaging system Bio-Rad ChemiDoc™ MP was used to detect immunoreactive protein signal. The protein signal was quantified by Image J software (version 1.8.0).

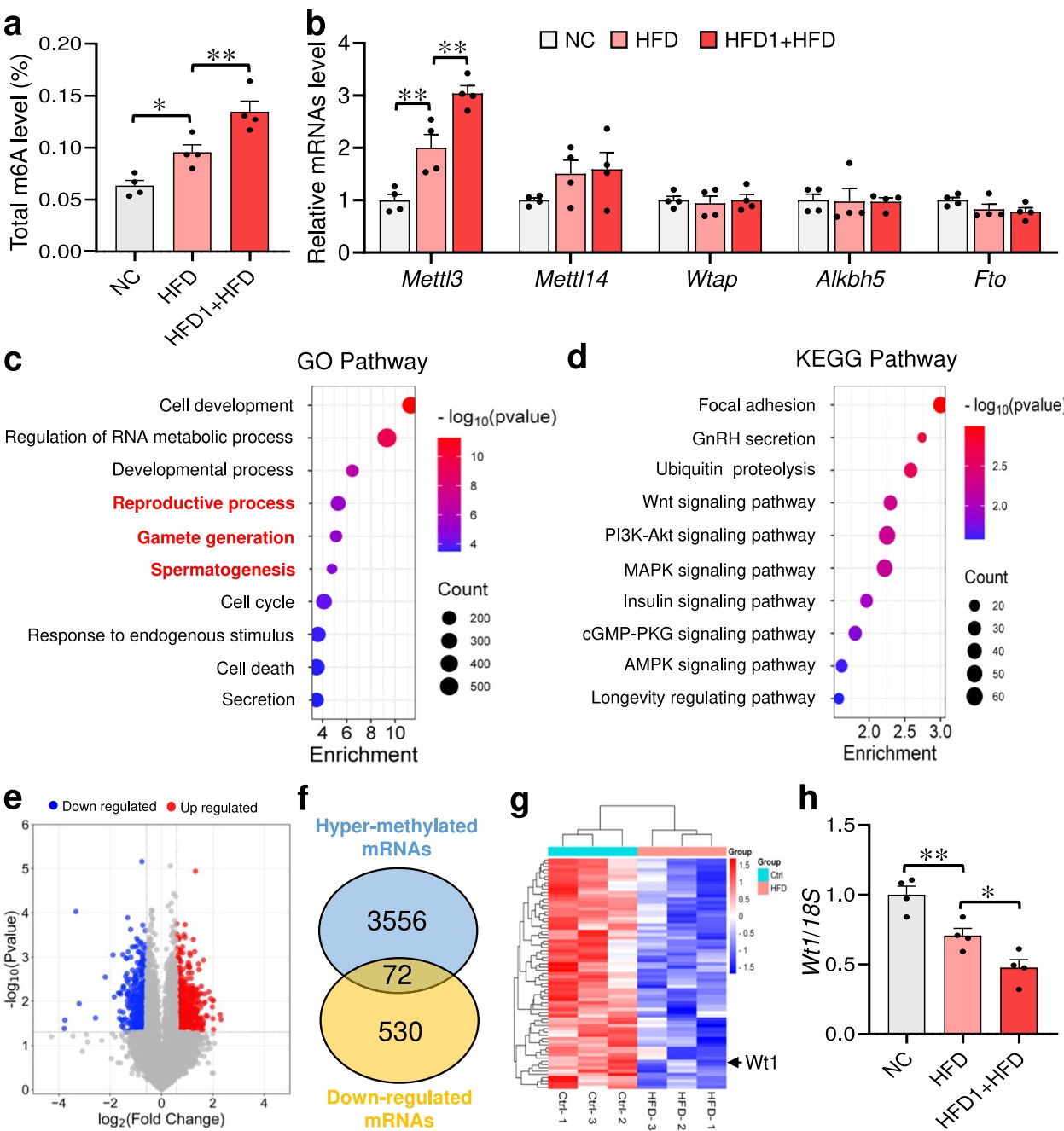

**Fig. 8 | Multigenerational paternal HFD progressively increases m6A level and downregulates *Wt1* mRNA expression in paternal sperm.** F0 generation male mice were fed NC or HFD from 5 weeks to 15 weeks old, and then mated with normal female to breed F1 generation. Similarly, partial F1 generation male mice continued to be fed HFD for 10 weeks. Sperm were collected for MeRIP-microarray and RNA-microarray. **a** The total RNA m6A level of sperm was measured. $n = 4$ mice, DOF = 11, $F = 20.97$, $P = 0.0004$. **b** Sperm *Mettl3*, *Mettl14*, *Wtap*, *Alkbh5* and *Fto* mRNA levels were tested using RT-PCR. $n = 4$ mice, DOF = 11, $F = 31.58$ and $P < 0.0001$ for *Mettl3*; $F = 1.77$ and $P = 0.2242$ for *Mettl14*; $F = 0.10$ and $P = 0.9066$ for *Wtap*; $F = 0.01$ and

$P = 0.9926$ for *Alkbh5*; $F = 2.13$ and $P = 0.1750$ for *Fto*. **c**, **d** GO and KEGG enrichment analysis of hyper-methylated mRNAs in sperm. **e** The distribution of differentially expressed mRNAs in sperm were presented. **f** Hyper-methylated mRNAs and down-regulated mRNAs in sperm were intersected. **g** Heatmap of overlapping mRNAs in sperm was depicted. **h** Sperm *Wt1* mRNA expression was tested using RT-qPCR. $n = 4$ mice, DOF = 11, $F = 21.19$, $P = 0.0004$. *$P < 0.05$. In regard to Fig. 8a, b, h, statistical significance was evaluated by two-sided one-way *ANOVA* with post hoc *LSD* tests. Data are presented as mean ± SEM. Source data are provided with this paper.

## Enzyme-linked immunosorbent assay (ELISA)
The quantity of retinoic acid (RA) in mouse testes and retinol in mouse serum were measured using the mouse RA-ELISA kit (MyBiosource, MBS706971) and the mouse retinol-ELISA kit (CUSABIO, CSB-E07891m), respectively. Assays were implemented according to the manufacturer's specifications. To detect testicular RA, 50 mg of testes was accurately weighed and homogenized in 900 ml ice PBS with

protease inhibitor. After centrifugation at 6000 g for 15 min, the supernatant was collected for RA detection. The concentration of testicular RA was presented as ng/g.

## m6A-mRNA&lncRNA epitranscriptomic microarray
Sperm total RNA was initially extracted with TRIzol reagent, and quantified using the NanoDrop ND-1000. The methylated RNA

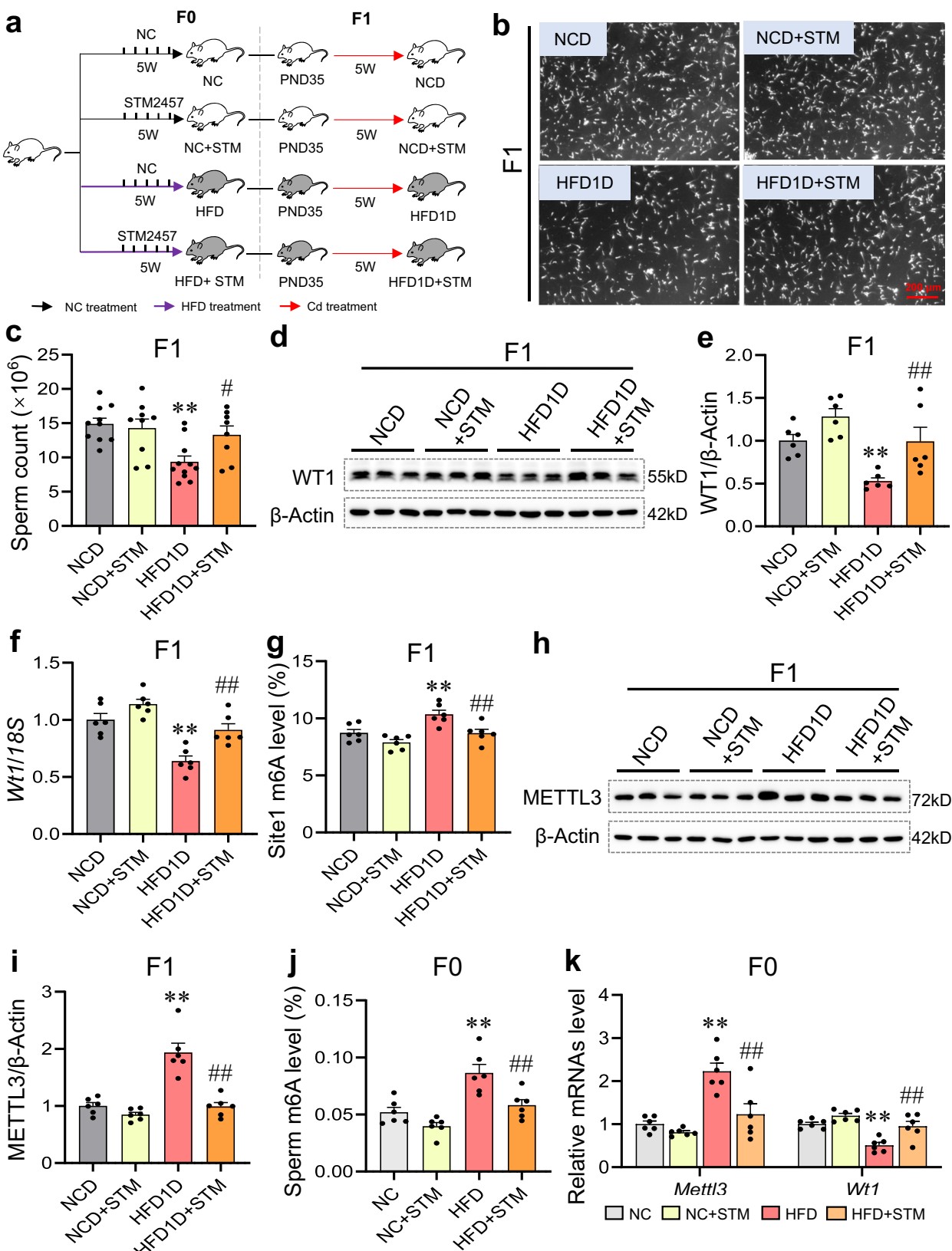

epitranscriptomic microarray were performed at Aksomics (Shanghai, China). Briefly, the total RNAs were immunoprecipitated with anti-m6A antibody. The modified RNAs were eluted from the immunoprecipitated magnetic beads as the IP. The unmodified RNAs were recovered from the supernatant as Sup. After Sup and IP RNAs labeling with Cy3 and Cy5 respectively, the cRNAs were combined together and

hybridized onto Arraystar Mouse mRNA&lncRNA Epitranscriptomic Microarray. Then, the arrays were scanned in two-color channels by an Agilent Scanner G2505C. Agilent Feature Extraction software (version 11.0.1.1) was used to analyze the collected chip fluorescence signals. The raw signal strength of IP and Sup was normalized using the mean value of $\text{Log}_2$-converted spik-in RNA fluorescence intensity. After

**Fig. 9 | Paternal HFD enhances testicular *Wt1* downregulation and spermatogenesis disorder in offspring via increasing METTL3-mediated paternal sperm m6A level.** F0 generation male mice were fed NC or HFD from 5 weeks to 15 weeks old, and treated with STM2457 once a week from 10 weeks to 15 weeks old. F1 generation male mice were exposed to CdCl$_2$ from 5 weeks to 10 weeks old, and named NCD, NCD + STM, HFD1D or HFD1D + STM group, respectively. **a** Experimental design flowchart. **b, c** Epididymal sperm counts were measured. $n = 10$ mice for NCD group; $n = 9$ mice for NCD + STM group; $n = 11$ mice for HFD1D group; $n = 8$ mice for HFD1D + STM group; DOF = 37, $F = 6.10$, $P = 0.0020$. **d, e** Testicular WT1 protein expression was measured by immunoblotting. $n = 6$ mice, DOF = 23, $F = 8.94$, $P = 0.0006$. **f** The mRNA level of testicular *Wt1* was

detected. $n = 6$ mice, DOF = 23, $F = 17.81$, $P < 0.0001$. **g** Testicular *Wt1* site1 m6A level was measured by MeRIP-qPCR. $n = 6$ mice, DOF = 23, $F = 10.20$, $P = 0.0003$. **h, i** The protein level of testicular METTL3 was measured. $n = 6$ mice, DOF = 23, $F = 27.16$, $P < 0.0001$. **j** Sperm total RNA m6A level was tested. $n = 6$ mice, DOF = 23, $F = 14.33$, $P < 0.0001$. **k** Sperm *Mettl3* and *Wt1* mRNA level were detected by RT-qPCR. $n = 6$ mice, DOF = 24, $F = 15.51$ and $P < 0.0001$ for *Mettl3*; $F = 14.90$ and $P < 0.0001$ for *Wt1*. *$P < 0.05$; **$P < 0.01$ vs NCD. #$P < 0.05$; ##$P < 0.01$ vs HFD1D. In regard to Fig. 9c, e–g, i–k, statistical significance was evaluated by two-sided one-way *ANOVA* with post hoc *LSD* tests. Data are presented as mean ± SEM. Source data are provided with this paper.

standardization, the difference in m6A methylation levels between the two groups was calculated using a T-test statistical model. FC (Fold Change) >1.5 and *P*-value < 0.05 were set as screening thresholds for differential expression.

## RNA sequencing
Total RNA of mouse testes was extracted, and quality inspections were performed. The sequencing library was constructed with 2.0 μg total RNA from each sample at Aksomics (Shanghai, China). In brief, mRNA enrichment was performed using NEB Next® Poly(A) mRNA Magnetic Isolation Module kit. The library from enriched RNA was constructed by KAPA Stranded RNA-Seq Library Prep Kit (Illumina) and inspected by Agilent 2100 Bioanalyzer. Then, the final quantification of library was conducted by qPCR. After denaturing with 0.1 M NaOH, 8 pM single stranded DNA was amplified in situ at the NovaSeq S4 Reagent Kit. The end of generated fragment was sequenced using Illumina NovaSeq 6000 sequencer for 150 cycles. After StringTie comparison of the sequencing results to the known transcriptome, the transcriptional abundances at Gene Level and Transcript Level were calculated using Ballgown. The unit of expression quantity was expressed by FPKM (Fragments Per Kilobase of gene/transcript model per Million mapped fragments). A gene or transcript with an expression value of FPKM > 0.5 was considered to be effectively expressed in the group. Finally, Ballgown software was used for statistical analysis by the F-test statistical model, FC > 1.2 and *P*-value < 0.05 were set as the screening threshold for differential expression.

## Isolation of total RNA and real-time RT-PCR
The cellular and testicular total RNA were initially extracted with TRIzol reagent (Invitrogen). The cDNA from 2 μg total RNA was obtained using AMV reverse transcriptase (Promega). The mRNAs expression of targeted genes was then detected by the Lightcycler®480 real-time fluorescence quantitative PCR machine (Roche, Switzerland). The primer sequences in this study are presented in Supplementary Table 1. *18 S* as the internal parameter. The expression of targeted mRNAs was analyzed by comparative the CT method.

## Quantification of m6A modifications
Total RNA in mouse testes and sperm was extracted by TRIzol reagent. The m6A level of total RNA was detected using the EpiQuik m6A RNA Methylation Quantification Kit (Epigentek). A total of 200 ng of RNA or positive controls were added separately to the assay wells, along with the captured antibody solution, the detected antibody solution, the enhanced solution, and the developer solution diluted to a suitable concentration. The absorbance was read at 450 nm, and the m6A levels of the RNA sample were calculated based on the standard curve.

## MeRIP-qPCR
Sperm total RNA was initially extracted with TRIzol reagent, and mRNA was enriched with Arraystar Seq-Star TM poly (A) mRNA isolation kit. The interrupted mRNA (1/10 of which was left as the input control) was mixed with 2 μg anti-m6A rabbit polyclonal antibody and IgG Dyna-beads into a 500 μL IP reaction system (50 mM Tris-HCl, pH 7.4, 0.1%

NP-40, 150 mM NaCl and 40 U/ml RNase inhibitor), and incubated at 4 °C for 2 h. After washing and elution, the binding RNA was leached by TRIzol[16,63]. Afterward, reverse transcription and amplification were performed. The SRAMP website (http://www.cuilab.cn/sramp/) predicted that *Wt1* mRNA has four highly credible m6A-modified sites. The corresponding primer sequences are shown in Supplementary Table 2. The different site m6A level of *Wt1* mRNA was quantified by RT-qPCR, which was calculated with the following formula: Site m6A level (%) = $2^{Ct(Input)-Ct(MeRIP)} \times Fd \times 100\%$. Among them, Fd is the input dilution factor.

## RNA interference
RNA interferences were performed following the approach of our previous study[18]. Briefly, mouse *Mettl3* or *Igf2bp1* specific small interference RNA (siR, GenePharma) were mixed with Lipofectamine 3000 for 15 min in serum-free medium. The mixtures were added to the culture medium for 6 h to transfect TM4 cells. Ather this, the medium was replaced with fresh medium and TM4 cells were cultured for 42 h. The sequence of *Mettl3* siRNA was 5′-GUCAGUAUCUUGGGCAAAUTT-3′ (forward) and 5′-AUUUGCCCAA GAUACUGACTT-3′ (reverse). The sequence of *Igf2bp1* siRNA was 5′- GUCCCAAGGAGGAAGUAAATT-3′ (forward) and 5′- UUUACUUCCUCCUUGGGACTT-3′ (reverse).

## RNA stability assay
Actinomycin D (5 μg/mL) inhibited transcription in TM4 cells. The mRNA expression of *Wt1* was detected by qRT-PCR after 0, 3, and 6 h of Cd treatment.

## RNA immunoprecipitation
Magna RIP RNA-Binding Protein Immunoprecipitation Kit (Millipore) was used to implement RNA immunoprecipitation. After lysing with complete RIP lysis buffer, TM4 cells were incubated with magnetic beads coated with antibodies against IgG or IGF2BP1 for 4 h at room temperature. Then, the complexes were hatched with proteinase K digestion buffer to extract immunoprecipitated-RNA. qPCR was used to determine the relative interaction between *Wt1* and IGF2BP1.

## Luciferase reporter assay
The *Wt1* CDS sequence containing the mutated motif (GAC to GTC) or the wild-type m6A motif was inserted into the pmirGLO-REPORT luciferase reporter vector (Promega). Then, plasmid transfections were conducted in TM4 cells with or without Cd treatment. The firefly and Renilla luciferase activities were detected by a dual-luciferase reporter assay system (Promega).

## Chemicals and reagents
Detailed information on chemical reagents and primary antibodies are listed in Supplementary Table 3.

## Statistical analysis
All statistical analyses were conducted by SPSS23.0 software, and data were expressed as means ± SEM. One-way *ANOVA* was used to compare means in multiple groups. Mean comparisons between the two groups

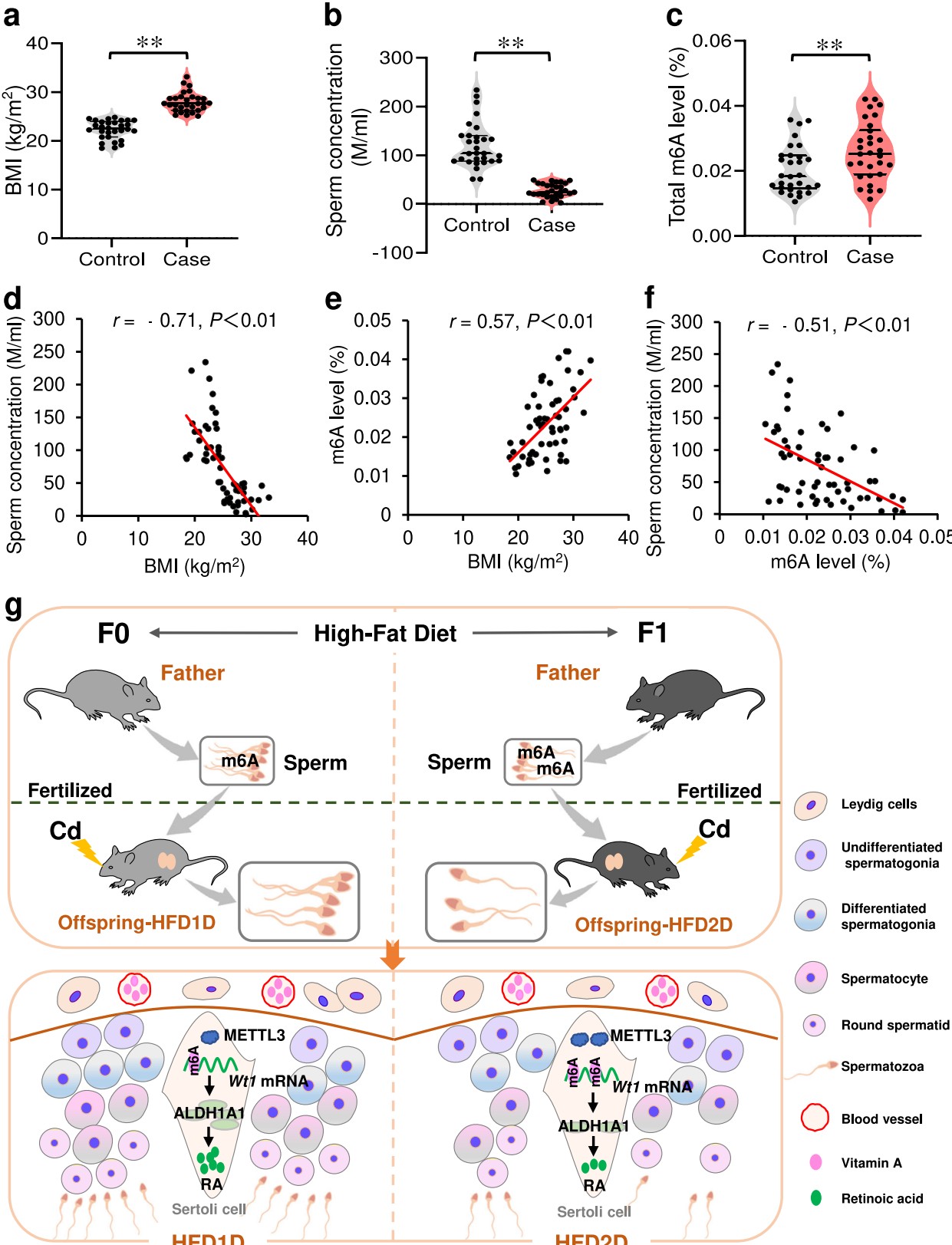

**Fig. 10 | Elevated sperm m6A level and decreased sperm concentration is observed in donors who were overweight or obese.** After removal of smoking or alcohol drinking donors, 30 pairs of cases with overweight/obesity and corresponding controls were obtained by matching age. **a** BMI. $n = 30$ human sperm, $t = -11.20$, $P < 0.0001$. Two-tailed $t$ test was used to analyze the differences. **b** Sperm concentration. $n = 30$ human sperm, $t = 10.50$, $P < 0.0001$. Two-tailed $t$ test was used to analyze the differences. **c** Sperm m6A level. $n = 30$ human sperm, $t = -2.74$, $P = 0.0083$. Two-tailed $t$ test was used to analyze the differences. **d** The association between sperm concentration and BMI. **e** The association between sperm m6A level and BMI. **f** The association between sperm m6A level and sperm concentration. **g** Graphical abstract. **$**P < 0.01$. Source data are provided with this paper.

were performed using two-tailed Student's *t*-test. Repeated-measures *ANOVA* was applied to analyze the GTT data. Two-way *ANOVA* was applied to analyze RNA stability assay and mouse body weight data. The data were graphically presented by GraphPad Prism 8.0 or R (version 3.6.0) processing. Experimental design flowchart (Figs. 1a, 5a, 9a and Supplementary Fig. 8a) and graphical abstract (Fig. 10g) were generated using PowerPoint software (office 2019). $P < 0.05$ meant the difference was statistically significant.

## Reporting summary

Further information on research design is available in the Nature Portfolio Reporting Summary linked to this article.

## Data availability

The m6A-mRNA&lncRNA Epitranscriptomic Microarray data and RNA sequencing used in this study data have been deposited in Gene Expression Omnibus database under accession code GSE241195 and GSE241413, respectively. The ChEA3 database are available at https://maayanlab.cloud/chea3/. The ENCORI database are available at https://starbase.sysu.edu.cn/index.php. The mRNA m6A site was predicted by the online tool SRAMP at http://www.cuilab.cn/sramp/. Source data are provided with this paper.

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

## Acknowledgements

This work was supported by National Key Research and Development Program of China (2022YFC2702904 to H.W.), National Natural Science Foundation of China (82273664 and 81973079 to H.W., 81930093 to D.X.X.), Research Funds of Center for Big Data and Population Health of IHM (JKS2022016 to H.W.), Academic Funding Project for Top Talents in Colleges and Universities (gxbjZD2020059 to H.W.), Anhui Provincial Academic and Technical Leader Reserve Candidate Research Funding (2020H208 to H.W.).

## Author contributions

Project administration: Y.W.X. Resources: H.L.Z, H.G and Y.X.C. Validation: J.Z. Methodology: L.L.T and X.M.Z. Formal analysis: H.L, K.W.W and Z.Y. Investigation: L.L.F. and X.R.W. Visualization: X.D.Z. Data curation: W.C. and Z.L.D. Conceptualization: Y.F.Z. and Y.W.X. Supervision: X.J.H. and D.X.X. Writing-original draft: Y.W.X. Writing-review & editing and Funding acquisition: H.W.

## Competing interests

The authors declare no competing interests.

## Additional information

[1]Department of Toxicology, School of Public Health, Anhui Medical University, Hefei, China. [2]Key Laboratory of Environmental Toxicology of Anhui Higher
Education Institutes, Hefei, China. [3]Reproductive Medicine Center, Department of Obstetrics and Gynecology, The First Affiliated Hospital of Anhui Medical
University, Hefei, China. [4]NHC Key Laboratory of Study on Abnormal Gametes and Reproductive Tract (Anhui Medical University), Hefei, China. [5]Key
Laboratory of Population Health Across Life Cycle (Anhui Medical University), Ministry of Education of the People's Republic of China, Hefei, China.
[6]Reproductive Medicine Center, Department of Obstetrics and Gynecology, Shanghai General Hospital, Shanghai Jiao Tong University School of Medicine,
Shanghai, China. [7]These authors contributed equally: Yong-Wei Xiong, Hua-Long Zhu, Jin Zhang, Hao Geng. ✉e-mail: hxj0117@126.com; xudex@126.com;
wanghuadev@ahmu.edu.cn

