## [Peer Review File NEW · Nature Communications]

Multigenerational paternal obesity enhances the susceptibility to male subfertility in offspring via Wt1 N6-methyladenosine modificationREVIEWER COMMENTS

Reviewer #1 (Remarks to the Author):

The manuscript by Xiong et al investigates the effect of multigenerational high fat diet induced obesity on semen quality in mice subjected to an environmental challenge, via cadmium exposure.

This is a comprehensive set of elegantly presented experiments in mice examining the interaction between paternal HFD consumption and an environmental challenge induced by cadmium in the drinking water.

This paper builds on previous work from the group examining the interaction between cadmium and high fat diet exposures, where males were exposed to cadmium and then offspring were exposed to control or high fat diet.

In that sense, the interaction between these two insults on testicular health has been reported previously.

In a previous paper [Nan et al, Chemosphere 2020 as well as the Xiang reference [18] used here], the authors showed that paternal cadmium exposure increased the susceptibility to high fat diet induced testicular injury and spermatogenic disorders in mouse offspring. In this experiment the exposures are flipped with high fat diet being implemented first, across 1 or 2 generations, to examine whether it exacerbates any damage to testicular development and spermatogenesis in offspring exposed to cadmium. So the incremental advance is that one or two generations of high fat diet in males impacts on subsequent response to cadmium.

The abstract makes no mention of cadmium as the method of testicular insult, which seems strange to me.

There is a detailed suite of interventional experiments which strongly support the mechanisms being proposed by the authors.

There are 9 Figures, 6 supplementary Figures and 3 supplementary Tables; the data are generally clearly presented.

That said, the number of replicates is an issue throughout the manuscript with figures having legends stating $n=3-10$ or $n=4-10$, but it isn't clear which of the panels have low n , or if indeed some groups have low n but others high n . This makes it hard to scrutinise the data and fully understand its message .

This is particularly relevant when considering the statistical significance of the findings – lines 612-616 states one way ANOVA was used, followed by Bonferroni's or Tamhane's post hoc test, where multiple groups are compared. And Students t test for 2 group comparisons. In many cases 3 comparisons are being made across 4 groups - where n is 3-4 across groups, as in Figure 6, and in many cases a range of n is used.

If n is 3 for a particular treatment group, it is not known if that flows through the various assays and outcomes.

Nor is mortality or exclusion of animals documented; Were there deaths of animals that were more pronounced in a particular treatment group for instance?

Thus, the information on the statistical analyses included in the paper needs to be more detailed. I note the instructions to authors states the following –

‘For all statistics (including error bars), provide the EXACT n values used to calculate the statistics (reporting individual values rather than a range if n varied among experiments). Where relevant, provide exact values for both significant and non-significant P values. For ANOVAs, provide F values and degrees of freedom’.

Other Methodological matters

Importantly more methodological detail is required, regarding how the groups were generated. It is important to know how many mice per litter were used in the different generations. Were all 1st (F1) and 2nd (F2) generation mice produced from 3 individual F0 pairs? The best practice to generate the n=10 would be to have 10 separate F0 mating pairs, then exposing a single F1 offspring to cadmium, and mating a littermate to generate an F2 litter (again only one of that litter being exposed to Cd).

Did the HFD impact fertility of the F0 and F1 mice? Were mice bred one on one?

CdCl₂ was provided in the drinking water at 100 mg/L (as used by this group previously, Nan et al, 2020). There is little discussion of this dose other than it has been used previously, or how it compares to other exposures that may impact the sperm/testis.

After WT1 expression was shown to be downregulated, to investigate the role of Wt1 in the response AAV9-Wt1 was injected – the methods states a local testicular injection was performed but no detail is provided - how was this done (how many sites etc); I note the same question applies to STM2457 treatment; site and number of injections, dose used etc would be useful.

Line 516 states that mice were fasted for 12 h prior to GTT without water - is this the case ?? Normally no food is provided during a fast, but mice need to be hydrated.

And GTT glucose data are normally analysed by repeated measures ANOVA.

The gene expression data are key to the argument; did the authors check that there was no influence of any of the interventions on testicular expression level of 18S? Typically, the expression of rRNA is much higher than the target gene and its degradation is reduced compared with mRNA; as there is much more 18S than transcripts of the gene of interest, this contradicts the assumptions for reference genes.

I note that researchers within the field of transgenerational epigenetic inheritance or DOHaD will want to see the HFD1 and HFD2 groups also in most of the figures (or in Supp information). For example, it would be interesting to know whether single- or double-generation HFD have effects on sperm and testicular parameters, retinoic acid metabolism gene expression, WT1 (and Mettl3) in HFD2 groups. Perhaps the authors can comment on this.

Regarding mechanism, others have reported that cadmium exposure alters the sperm methylome (e.g. Saintilnord et al Toxicol Sci. 2021 Apr; 180(2): 262–276 – is this worth adding ?

The study would be strengthened further with data from human studies such as NC, HFD and HFD1+HFD sperm m6A levels in human donors.

The final concluding sentence makes rather a sweeping statement- perhaps this could be modified.

Minor points

The abstract may contain a typo, 'mechanically' should probably be 'mechanistically'.

Line 147 and 149 Fig 2C and 2D should be 3C and 3 D

Line 183 'conformably ' perhaps another word can be substituted there (eg 'In line with this,')

Line 372 'Nevertheless' is repeated

Line 420 'The modification of m6A involves in testicular ...' - review grammar

Line 456 remove 'a'

Why is Cd used to indicate cadmium exposure in Figure 7 but 'D' in the previous figures?

Reviewer #2 (Remarks to the Author):

The study by Xiong et al explores the role that a paternal high fat diet has on offspring reproductive fitness and spermatogenesis in a mouse model. The authors report a role for the transcription factor Wt1 and retinoic acid in regulating offspring sperm production.

While there is merit in the study, I have several major and minor issues that need to be addressed.

First, could the authors provide data on litter sizes, pregnancy rates and general fertility measurements in their mice? While i appreciate there is a decline in sperm number, this doesn't necessarily translate into a decline in fertility. therefore, data showing whether mating, embryo number, litter sizes and or offspring survival are necessary to place the changes in sperm number into context.

While the authors took their experimental offspring groups out to an F2 generation, why did they not do the same with the control (NC) group? I find it odd that they are comparing phenotypic changes in the experimental F2 offspring against a control F1 group.

Minor comments

In line 17, did the authors mean they looked at 'different testicular germ cell markers...'?

In line 23, did the authors mean 'mechanistically' rather than 'mechanically'?

Line 32, did the authors mean that the number of obese people is on the rise, rather than the people them selves being on the rise?

Lines 40-44, the introduction of the transgenerational concept feels very clunky and i feel that a reference focused on transgenerational reproductive changes would be more appropriate.

The rationale for using cadmium in this study is not explained.

In line 72, which 'white fat' depot was measured? And if it was only one then why didn't they look at the others.

Throughout the manuscript, the authors use terms such as '...paternal exposure to multigenerational HFD...' (See line 85). However, this makes it sound like it was the previous generations, before the F0 father, who were exposed to the HFD. This is not the case and so the wording of such phrases needs to be changed to better reflect the design of the study.

Similarly, I am unsure what the authors mean by a 'partial F1 generation' such as in line 91.

I believe that the more accurate name for MVH (line 105) is Ddx4.

IN Figure 3, I find it odd that after conducting RNA-seq on the offspring testes, the authors then conduct additional analysis of gene expression using RT-qPCR. Why didn't they check the expression of the genes in panel F within their RNA-Seq data?

In Figure 4E, is that the average number of WT1+ cells per tubule or per unit area? It would be of interest to see which cells are expressing WT1 through some double staining for Sox9 if the authors think its in the Sertoli cells.

For the analysis of retinoic acid levels, why have the authors normalised the control group to 1? As this was an ELISA, why not display it in the units of RA?

When the authors refer to siR (e.g, line 283) do the authors mean siRNA?

In figure 8F, the genes in blue are regarded as being hypermethylated but the genes in yellow are regarded as being down regulated? This needs amending so that its either hyper/hypo or up/down regulated.

The Materials and Methods lacks significant amounts of information. Line 480, and throughout this section, how were the mice euthanised?

The sections on the AAV9 and STM2457 lack detail. where did the AAV9-wt1 construct come from? How much was used? How did they inject it into the testes, what was the surgical procedure? Was any anaesthetic and analgesia used? Did they inject into both testes? How did they test that it had incorporated into the testicular tissue? What was the control? These questions also apply to the STM2457 construct.

Did the authors normalise their MTT assay to cell number or protein content?

For the RA ELISA, how did authors lyse the testis tissue and did they normalise for protein content?

How were the sperm obtained for the methylated RNA sequencing?

Have the authors deposited their sequencing data into a publicly accessible repository?

What statistical approach was used for the analysis of the sequencing data?

The details on the RNA-Seq is very scant and require a lot more information.

For the MTT assay, what was the medium that the cells were grown in?

For the determination of % RNA methylation by MeRIP qPCR, did the authors use a standard curve? If so this needs to be detailed.

There are no Materials and Methods details on any of the Western blotting experiments.

Manuscript No.: NCOMMS-23-17775A

Title: Multigenerational paternal obesity enhances the susceptibility to poor semen quality in male offspring via *Wt1* m6A modification

Dear reviewers,

Thank you for your letter. According to reviewers' comments and questions, the detailed revisions are as follows,

Reviewer comments:

Reviewer #1(Remarks to the Author):

1. The manuscript by Xiong et al investigates the effect of multigenerational high fat diet induced obesity on semen quality in mice subjected to an environmental challenge, via cadmium exposure. This is a comprehensive set of elegantly presented experiments in mice examining the interaction between paternal HFD consumption and an environmental challenge induced by cadmium in the drinking water. This paper builds on previous work from the group examining the interaction between cadmium and high fat diet exposures, where males were exposed to cadmium and then offspring were exposed to control or high fat diet. In that sense, the interaction between these two insults on testicular health has been reported previously. In a previous paper [Nan et al, Chemosphere 2020 as well as the Xiong reference [18] used here], the authors showed that paternal cadmium exposure increased the susceptibility to high fat diet induced testicular injury and spermatogenic disorders in mouse offspring. In this experiment the exposures are flipped with high fat diet being implemented first, across 1 or 2 generations, to examine whether it exacerbates any damage to testicular development and spermatogenesis in offspring exposed to cadmium. So the incremental advance is that one or two generations of high fat diet in males impacts on subsequent response to

cadmium.

Response: Thanks for reviewer's comments. There is strong evidence that obesity is a risk factor for poor semen quality. However, the effects of paternal multigenerational obesity on the susceptibility to spermatogenesis disorders in offspring remain unknown. On the basis of our previous research, this study further explored the effect of multigenerational paternal high-fat diet (HFD) on the susceptibility to cadmium-induced spermatogenesis disorders in offspring and its mechanism. Results suggested that multigenerational paternal HFD gradually enhanced the susceptibility to spermatogenesis disorders in male offspring via downregulating *Wt1* mRNA expression, and METTL3-mediated sperm m6A modification was involved in paternal HFD-induced a decrease of *Wt1* mRNA stability in offspring testes. As result of these findings, it opens an avenue for mechanistic exploration of paternally acquired male subfertility and intergenerational transmission, including m6A modification and *Wt1* expression. Among them, sperm m6A modification may be one of the genetic markers for early screening of paternally acquired male subfertility.

2. The abstract makes no mention of cadmium as the method of testicular insult, which seems strange to me.

Response: Thanks for reviewer's comment. Information on cadmium as an inducer of testicular toxicity has been added into the revised Abstract section.

The detailed revision is as follows,

In the revised Abstract section (Lines 23-24)

There is strong evidence that obesity is a risk factor for poor semen quality. However, the effects of multigenerational paternal obesity on the susceptibility of cadmium (a reproductive toxicant)-induced spermatogenesis disorders in offspring remain unknown.....

3. There is a detailed suite of interventional experiments which strongly support the mechanisms being proposed by the authors. There are 9 Figures, 6 supplementary Figures and 3 supplementary Tables; the data are generally clearly presented. That said, the number of replicates is an issue throughout the manuscript with figures having legends stating $n=3-10$ or $n=4-10$, but it isn't clear which of the panels have low n , or if indeed some groups have low n but others high n . This makes it hard to scrutinise the data and fully understand its message.

Response: Thanks for reviewer's kind comments. We are sorry for the misunderstanding caused by the unclear sample size. Additional repetitions of animal experiments were added. In the revised manuscript, we replaced all bar graphs with plots that feature information about the distribution of the underlying data. The number of detailed replicates has also been indicated in the revised Figures and Figure legends section.

The detailed revision are as follows,

In the Figures and Figure legends section (**Lines 98-443**)

Fig.1. Multigenerational paternal HFD enhances susceptibility to spermatogenesis disorder in offspring. (A) Experimental design flowchart of Cd-induced spermatogenesis impairment in mice with HFD-feeding. The black arrow indicated NC treatment. The purple arrow indicated HFD treatment. The red arrow indicated Cd treatment. (B) The fertility rate was calculated. $n=4$, Degree of freedom (DOF)=11, $F=18.20$, $P=0.0007$. (C) The pregnancy rate was counted. $n=4$, $P=0.3911$. (D and E) Epididymal sperm counts were measured. $n=10$, DOF=39, $F=37.68$, $P<0.0001$. (F) Sperm motility was recorded. $n=10$, DOF=39, $F=14.01$, $P<0.0001$. (G) Testicular H&E staining. (H) The number of Testicular seminiferous tubules number at different stages were evaluated. $n=4$, DOF=15, $F=30.25$, $P<0.0001$. n.s., not significant. $*P < 0.05$; $**P < 0.01$.

4. This is particularly relevant when considering the statistical significance of the findings – lines 612-616 states one way ANOVA was used, followed by Bonferroni's or Tamhane's post hoc test, where multiple groups are compared. And Students t test for 2 group comparisons. In many cases 3 comparisons are being made across 4 groups - where n is 3-4 across groups, as in Figure 6, and in many cases a range of n is used. If n is 3 for a particular treatment group, it is not known if that flows through the various assays and outcomes. Nor is mortality or exclusion of animals documented; Were there deaths of animals that were more pronounced in a particular treatment group for instance? Thus, the information on the statistical analyses included in the paper needs to be more detailed. I note the instructions to authors states the following – 'For all statistics (including error bars), provide the EXACT n values used to calculate the statistics (reporting individual values rather than a range if n varied among experiments). Where relevant, provide exact values for both significant and non-significant P values. For ANOVAs, provide F values and degrees of freedom'.

Response: Thanks for reviewer's questions and suggestions. To further determine the statistical significance of our findings, additional repetitions were added for all animal experiments whose sample size n was 3. As shown in Figure 6, Figure 9 and other Figures, the sample size for Immunoblotting, RT-PCR, and MeRIP qPCR was increased from 3 to 6. In general, the results of statistical analysis in the revised manuscript were similar to those before. The detailed information on statistical analyses, such as the exact values of *n*, *P*, *F* and degrees of freedom (DOF), has also been added into revised manuscript. Additionally, no animal deaths were observed in the particular treatment group.

The detailed revision are as follows,

In the Figures and Figure legends section (**Lines 98-443**)

The supplemental immunoblotting of Figure 6

Fig.6. Multigenerational paternal HFD gradually exacerbates environmental stress-elevated testicular *Wt1* m6A level in offspring.(A) Testicular total RNA m6A level was measured. $n=4$, $DOF=15$, $F=23.52$, $P<0.0001$. (B and C) Testicular

METTL3 protein expression was detected by immunoblotting. $n=6$, $DOF=23$, $F=117.00$, $P<0.0001$. (D) Testicular *Ythdc1*, *Ythdc2*, *Ythdf1*, *Ythdf2*, *Ythdf3*, *Igf2bp1*, *Igf2bp2* and *Igf2bp3* mRNA levels were tested by RT-PCR. $n=6$, $DOF=23$, $F=1.14$ and $P=0.3565$ for *Ythdc1*; $F=4.49$ and $P=0.0145$ for *Ythdc2*; $F=18.81$ and $P<0.0001$ for *Ythdf1*; $F=29.71$ and $P<0.0001$ for *Ythdf2*; $F=2.15$ and $P=0.1259$ for *Ythdf3*; $F=28.11$ and $P<0.0001$ for *Igf2bp1*; $F=1.46$ and $P=0.2566$ for *Igf2bp2*; $F=3.71$ and $P=0.02858$ for *Igf2bp3*. (E and F) Testicular YTHDF1, YTHDF2 and IGF2BP1 protein expression were detected by immunoblotting. $n=6$, $DOF=23$, $F=26.49$ and $P<0.0001$ for YTHDF1; $F=22.44$ and $P<0.0001$ for YTHDF2; $F=23.94$ and $P<0.0001$ for IGF2BP1. (G) The four m6A modification sites in *Wtl* mRNA were predicted by online tool SRAMP. (H) *Wtl* site1 m6A level was measured by MeRIP-qPCR. $n=6$, $DOF=23$, $F=34.76$, $P<0.0001$. n.s., not significant. $*P<0.05$; $**P<0.01$.

The supplemental immunoblotting of Figure 9

Fig.9. Paternal HFD enhances testicular *Wtl* downregulation and spermatogenesis disorder in offspring via increasing METTL3-mediated paternal sperm m6A level.

F0 generation male mice were fed NC or HFD from 5 weeks to 15 weeks old, and treated with STM2457 once a week from 10 weeks to 15 weeks old. F1 generation male mice were exposed to CdCl₂ from 5 weeks to 10 weeks old, and named NCD, NCD+STM, HFD1D or HFD1D+STM group, respectively. (A) Experimental design flowchart. (B and C) Epididymal sperm counts were measured. $n=10$ for NCD group; $n=9$ for NCD+STM group; $n=11$ for HFD1D group; $n=8$ for HFD1D+STM group; $DOF=37$, $F=6.10$, $P=0.0020$. (D and E) Testicular WT1 protein expression was measured by immunoblotting. $n=6$, $DOF=23$, $F=8.94$, $P=0.0006$. (F) The mRNA level of testicular *Wtl* was detected. $n=6$, $DOF=23$, $F=17.81$, $P<0.0001$. (G) Testicular *Wtl* site1 m6A level was measured by MeRIP qPCR. $n=6$, $DOF=23$, $F=10.20$, $P=0.0003$. (H and I) The protein level of testicular METTL3 was measured. $n=6$, $DOF=23$, $F=27.16$, $P<0.0001$. (J) Sperm total RNA m6A level was tested. $n=6$, $DOF=23$, $F=14.33$, $P<0.0001$. (K) Sperm *Mettl3* and *Wtl* mRNA level were detected by RT-PCR. $n=6$, $DOF=24$, $F=15.51$ and $P<0.0001$ for *Mettl3*; $F=14.90$ and $P<0.0001$ for *Wtl*. * $P<0.05$; ** $P<0.01$ vs NCD. # $P<0.05$ vs HFD1D; ### $P<0.01$ vs HFD1D.

Other Methodological matters:

5. Importantly more methodological detail is required, regarding how the groups were generated. It is important to know how many mice per litter were used in the different generations. Were all 1st (F1) and 2nd (F2) generation mice produced from 3 individual F0 pairs? The best practice to generate the $n=10$ would be to have 10 separate F0 mating pairs, then exposing a single F1

offspring to cadmium, and mating a littermate to generate an F2 litter (again only one of that litter being exposed to Cd).

Response: Thanks for reviewer's suggestion and questions. The details about groups generation of animal experiment 1 are as follows. Twenty F0 generation male mice were fed with HFD or normal chow (NC) from 5 weeks to 15 weeks old, and then mated with normal female mice to breed F1 generation. After mating, 10 pregnant mice were obtained in each group. Similarly, 10 F1 generation male mice (whose fathers were exposed to HFD) from different litters continued to be fed HFD for 10 weeks, and mated with normal females to breed F2 generation. Then, 10 male mice from each group of F1 and F2 generations were exposed to CdCl₂ (0 or 100 mg/L) by drinking water for 10 weeks, and named NC (normal control), NCD (the offspring were treated with Cd after paternal treatment with normal chow), HFD1D (the offspring were treated with Cd after paternal exposure to one-generational HFD) or HFD2D (the offspring were treated with Cd after paternal exposure to bi-generational HFD) groups, respectively. In the different generations, 10 pregnant mice were obtained in each group after mating with different male mice, and the balances of sexes between pups (male: female, 3: 3) were performed on postnatal day (PND) 1. In PND28, one male mouse from each litter was selected. Therefore, F1 and F2 generation mice were produced from 10 individual F0 pairs. The more methodological details have been added into the revised manuscript.

The detailed revision are as follows,

In the revised Materials and Methods section (**Lines 547-566**)

Animal treatments

The C57BL/6N mice were provided by Beijing Vital River Laboratory Animal Technology Co., Ltd (Beijing, China). All animals were accustomed to standard conditions for a week. The purified control feed (TP23302) and 60% fat high-fat feed

(TP23300) were acquired from Trophic Animal Feed High-Tech Co., Ltd (Nantong, China). In the different generations, 8 to 11 pregnant mice were obtained in each group after mating with different male mice, and the balances of sexes between pups (male: female, 3: 3) were performed on postnatal day (PND) 1. At PND28, one male mouse from each litter was selected. For euthanasia, all mice were injected intraperitoneally with 2,2,2-tribromoe-thanol (250 mg/kg), and then cervical dislocation was executed under anesthesia. This study was approved by the Laboratory Animal Ethics Committee of Anhui Medical University (ethical approval number: LLSC82273664).

Experiment 1. The experimental design used to investigate the susceptibility to spermatogenesis disorders in the offspring of fathers stressed by multigenerational HFD is shown in Fig. 1A. Briefly, 20 F0 generation male mice were fed with HFD or normal chow (NC) from 5 weeks to 15 weeks old, and then mated with normal female mice to breed F1 generation. After mating, 10 pregnant mice were obtained in each group. Similarly, 10 F1 generation male mice (whose fathers were exposed to HFD) from different litter continued to be fed HFD for 10 weeks, and mated with normal females to breed F2 generation. Then, 10 male mice from each group of F1 and F2 generations were exposed to CdCl₂ (0 or 100 mg/L) by drinking water for 10 weeks, and named NC (normal control), NCD (the offspring were treated with Cd after paternal treatment with normal chow), HFD1D (the offspring were treated with Cd after paternal exposure to one-generational HFD) or HFD2D (the offspring were treated with Cd after paternal exposure to bi-generational HFD) groups, respectively. Cd, a well-known reproductive toxicant, impairs testicular development and spermatogenesis. Population-based studies found that Cd concentrations in blood and semen were negatively correlated with sperm quality^{55,56}.....

6. Did the HFD impact fertility of the F0 and F1 mice? Were mice bred one on one?

Response: Thanks for reviewer's questions. The effect of HFD exposure on the fertility rate of F0 and F1 mice were investigated. As shown in Fig. 1B, the gradual reductions of fertility rate were observed in

mice with the increase of HFD generation. In fertility test, one male mouse was mated with two female mice at night, and vaginal plug were checked in the next morning. The number of fertile male mice was recorded and divided by the total number of male mice to calculate the fertility rate of each group on that day. This test was repeated twice again every 4 days, a total of 4 repetitions. More details for the fertility test have been added in the revised manuscript.

The detailed revision are as follows,

In the revised Results section (Lines 83-113)

Multigenerational paternal HFD gradually enhances susceptibility to spermatogenesis disorder in their offspring

..... Furthermore, HFD exposure induced fasting blood glucose elevation, glucose tolerance impairment, hepatic lipid deposition and serum triglyceride (TG) increase in mice (Figs. S1D-H). The effect of multigenerational paternal HFD exposure on the fertility rate, pregnancy rate and litter size in mice were then investigated. As presented in Fig. 1A-C and Fig. S2A, the gradual reductions of fertility rate were observed in mice with the increase of HFD generation, but not litter size. The effect of paternal HFD exposure on the susceptibility of environmental stress-induced spermatogenesis impairment in offspring was also explored.....

Fig.1. Multigenerational paternal HFD enhances susceptibility to spermatogenesis disorder in offspring. (A) Experimental design flowchart of Cd-induced spermatogenesis impairment in mice with HFD-feeding. The black arrow indicated NC treatment. The purple arrow indicated HFD treatment. The red arrow indicated Cd treatment. (B) The fertility rate was calculated. $n=4$, Degree of freedom (DOF)=11, $F=18.20$, $P=0.0007$. (C) The pregnancy rate was counted. $n=4$, $P=0.3911$. (D and E) Epididymal sperm counts were measured. $n=10$, DOF=39, $F=37.68$, $P<0.0001$. (F) Sperm motility was recorded. $n=10$, DOF=39, $F=14.01$, $P<0.0001$. (G) Testicular H&E staining. (H) The number of Testicular seminiferous tubules number at different stages were evaluated. $n=4$, DOF=15, $F=30.25$, $P<0.0001$. n.s., not significant. $*P < 0.05$; $**P < 0.01$.

In the revised Materials and Methods section (Lines 626-632)

Fertility test

For fertility rate, one male mouse was mated with two female mice at night, and vaginal plug were checked in the next morning. The number of fertile male mice was recorded and divided by the total number of male mice to calculate the fertility rate of each group on that day. This test was repeated twice again every 4 days, a total of 4 repetitions. For pregnant rate, female mice with vaginal plug were fed at single caged, and the pregnancy results were recorded. The litter sizes were also observed after delivery.

7. CdCl₂ was provided in the drinking water at 100 mg/L (as used by this group previously, Nan et al, 2020). There is little discussion of this dose other than it has been used previously, or how it compares to other exposures that may impact the sperm/testis.

Response: Thanks for reviewer's comments. Cadmium (Cd), one of the common contaminants in drinking water, impairs testicular development and spermatogenesis. It's well-known that Cd exposure impairs testicular spermatogenesis mainly via the dose of internal exposure. In the current study, we selected the dose of Cd based on the internal exposure dose in human body. Several population studies had found that the blood Cd concentration of patients with low semen quality could reach 10 µg/L [1,2]. In our pervious study, the sera Cd concentration was 7.21±0.74 µg/L when mice were exposed to CdCl₂ (100 mg/L) through drinking water [3], which was close to the blood Cd of patients with low semen quality. Meaningfully, the sera Cd concentration was related to Cd level of human in severe Cd-contaminated areas and smokers [4-6]. As above, 100 mg/L CdCl₂ was used in the present study and our previous study [7].

The detailed revision are as follows,

In the revised Materials and methods section (Lines 572-578):

Animal treatments

.....Here, Cd was used as an inducer of reproductive toxicity to induce spermatogenesis disorders in mice. It's well-known that Cd exposure impairs testicular spermatogenesis mainly via the dose of internal exposure. In the present study, the dosage of 100 mg/L CdCl₂ was chosen according to the internal exposure level in population studies and previous research^{18, 58}. Our pervious study showed the sera Cd concentration when mice were exposed to CdCl₂ (100 mg/L) through drinking water was close to the blood Cd of patients with low semen quality and was related to Cd level of human in severe Cd-contaminated areas and smokers^{58, 59}. Meanwhile, single- and double-generations of HFD-fed fathers named HFD and HFD1+HFD group, and their offspring named HFD1 and HFD2 group, were created.....

[1] Mínguez-Alarcón L, Mendiola J, Roca M, et al. Correlations between Different Heavy Metals in Diverse Body Fluids: Studies of Human Semen Quality. *Adv Urol*. 2012; 2012:420893.

[2] Telisman S, Cvitković P, Jurasović J, Pizent A, Gavella M, Rocić B. Semen quality and reproductive endocrine function in relation to biomarkers of lead, cadmium, zinc, and copper in men. *Environ Health Perspect*. 2000; 108(1): 45-53.

[3] Xiong YW, Tan LL, Zhang J, et al. Combination of high-fat diet and cadmium impairs testicular spermatogenesis in an m6A-YTHDF2-dependent manner. *Environ Pollut*. 2022; 313:120112.

[4] Kiziler AR, Aydemir B, Onaran I, et al. High levels of cadmium and lead in seminal fluid and blood of smoking men are associated with high oxidative stress and damage in infertile subjects. *Biol Trace Elem Res*. 2007;120(1-3):82-91.

[5] Xu X, Liao W, Lin Y, Dai Y, Shi Z, Huo X. Blood concentrations of lead, cadmium, mercury and their association with biomarkers of DNA oxidative damage in preschool children living in an e-waste recycling area. *Environ Geochem Health*. 2018; 40(4):1481-1494.

[6] Böhlant A, Schierl R, Diemer J, et al. High concentrations of cadmium, cerium and lanthanum in indoor air due to environmental tobacco smoke. *Sci Total Environ*. 2012; 414:738-741.

[7] Nan Y, Yi SJ, Zhu HL, et al. Paternal cadmium exposure increases the susceptibility to diet-induced testicular injury and spermatogenic disorders in mouse offspring. *Chemosphere*. 2020; 246:125776.

8. After WT1 expression was shown to be downregulated, to investigate the

role of *Wt1* in the response AAV9-*Wt1* was injected – the methods states a local testicular injection was performed but no detail is provided - how was this done (how many sites etc); I note the same question applies to STM2457 treatment; site and number of injections, dose used etc would be useful.

Response: Thanks for reviewer's comment and suggestion. We apologize for missing the details of the local testicular injection. Local testicular injections were performed according to previous study with corresponding modification [1]. In brief, the mice were injected intraperitoneally with 250 mg/kg 2,2,2-tribromoethanol. After anesthesia, both testes of the mice were found by touch pressure. Then, 20 μ L AAV9 or STM2457 were injected into the middle of each testes using a micropipette. After injection, the mice were placed flat in cage and observed for 30 min until they were fully awake and returned to normal. Among them, 1.03×10^{12} vg/ml AAV9-*Wt1* were injected once into mice at PND35. STM2457 were injected at a dose of 50.0 mg/kg, once a week for 5 weeks. The dose of STM2457 was chosen according to the previous study [2].

To determine whether the local injection of AAV9-*Wt1* can be integrated into the entire testicular tissue, a preliminary experiment was first performed. Results showed that after injection of the Ponceau S, the entire testis appeared red compared to the normal saline (Fig. S12). In addition, Figs. 5H and J-K also presented that the expression of WT1 in the testes was significantly upregulated after local testicular injection of AAV9-*Wt1*. These results suggested that local injection of AAV9-*Wt1* could be integrated into the entire testicular tissue.

Fig. S12

Figs. 5H and J-K

Fig.5. Paternal HFD aggravates environmental stress-impaired testicular spermatogenesis via inhibiting WT1-mediated retinoic acid synthesis in offspring. (A) Experimental design flowchart..... (G) Testicular retinoic acid level was detected by ELISA. $n=6$, $DOF=23$, $F=13.88$, $P<0.0001$. (H–J) Testicular ALDH1A1 and WT1 protein levels were measured by immunoblotting. $n=6$, $DOF=23$, $F=30.92$ and $P<0.0001$ for ALDH1A1; $F=48.60$ and $P<0.0001$ for WT1. (K) Testicular WT1 expression was measured by immunohistochemistry. * $P < 0.05$; ** $P <$

0.01 vs NCD. #*P* < 0.05; ##*P* < 0.01 vs HFD1D.

The detailed revision are as follows,

In the revised Materials and methods section (Lines 638-649):

Local testicular injection

Local testicular injections were performed according to previous study with corresponding modification⁶⁰. In brief, the mice were injected intraperitoneally with 250 mg/kg 2,2,2-tribromoethanol. After anesthesia, both testes of the mice were found by touch pressure. Then, 20 μ L AAV9 or STM2457 were injected into the middle of each testes using a micropipette. After injection, the mice were placed flat in cage and observed for 30 min until they were fully awake and returned to normal. Among them, 1.03×10^{12} vg/ml AAV9-*Wt1* were injected once into mice at PND35. AAV9-NC was used as control. AAV9-*Wt1* and AAV9-NC were synthesized by General Biol Co., Ltd. STM2457 were injected at a dose of 50.0 mg/kg, once a week for 5 weeks. 20% SBE- β -CD was used as control. STM2457 and SBE- β -CD were purchase from MedChemExpress. The dose of STM2457 was choose according the previous study⁶¹.

[1] Wu H, Zhang X, Yang J, et al. Taurine and its transporter TAUT positively affect male reproduction and early embryo development. *Hum Reprod.* 2022;37(6):1229-1243.

[2] Yankova E, Blackaby W, Albertella M, et al. Small-molecule inhibition of METTL3 as a strategy against myeloid leukaemia. *Nature.* 2021;593(7860):597-601.

9. Line 516 states that mice were fasted for 12 h prior to GTT without water - is this the case?? Normally no food is provided during a fast, but mice need to be hydrated. And GTT glucose data are normally analyzed by repeated measures ANOVA.

Response: Thanks for reviewer's question and suggestion. We are very sorry for the misunderstanding caused by our inappropriate expression. The mice were fasted for 12 h prior to GTT, and maintained normal water intake. In addition, the GTT glucose data have been analyzed by repeated measures ANOVA. Results showed that HFD exposure induced glucose tolerance impairment in mice.

The detailed revision are follows,

In the revised Materials and Methods section (Lines 635 and 790)

Glucose tolerance test (GTT)

The F0 generation male mice were intraperitoneally injected with 2.0 g/kg glucose solution after fasting (maintaining normal water intake) for 12 h. The blood glucose was then detected in the caudal vein using a glucometer, and recorded at different time

Statistical analysis

All statistical analyses were conducted by SPSS23.0 software, and data were expressed as *means* ± *SEM*. One-way *ANOVA* was used to compare means in multiple groups. Mean comparisons between the two groups were performed using *Student's t-test*. Repeated-measures *ANOVA* was applied to analyze the GTT data. $P < 0.05$ meant the difference was statistically significant.

In the revised Fig. S1E section (Lines 7)

Fig.S1. HFD induces obesity phenotype in mice. F0 generation male mice were fed NC or HFD from 5 weeks to 15 weeks old.....(D) Fasting blood glucose (FBG). $n=8$, $t=-4.07$, $P=0.0025$. (E) Glucose tolerance test (GTT) was performed. $n=8$, $P<0.0001$ (F) AUC for GTT. $n=5$, $t=-2.57$, $P=0.0332$. (G) Liver H&E staining. (H) Serum triglyceride (TG) was detected by ELISA. $n=6$, $t=-4.33$, $P=0.0024$. * $P < 0.05$; ** $P < 0.01$ vs NC.

10. The gene expression data are key to the argument; did the authors check that there was no influence of any of the interventions on testicular expression level of 18S? Typically, the expression of rRNA is much higher than the target gene and its degradation is reduced compared with mRNA; as there is much

more 18S than transcripts of the gene of interest, this contradicts the assumptions for reference genes.

Response: Thanks for reviewer's comment and question. We checked that there was no effect of different treatments on the expression of 18S in mouse testes. Therefore, 18S could be chosen as the internal control.

The Mean Cp of 18S in the RT-PCR experiment after different treatments are shown in the table below.

Table 1. Multigenerational paternal HFD exacerbated environmental stress-downregulated testicular *Wt1* expression in offspring.

Chart	Pairing	Sample Name	Targets	References	Mean Cp of Wt1	Mean Cp of 18S	Target/Ref	Normalized
TRUE		NC1	Wt1	18S	17.887	8.765	0.000	1.000
TRUE	A2/G2	NC2	Wt1	18S	17.937	8.759	0.054	0.963
TRUE	A3/G3	NC3	Wt1	18S	17.438	8.665	-0.350	1.274
TRUE	A4/G4	NCD1	Wt1	18S	18.437	8.703	0.611	0.655
TRUE	A5/G5	NCD2	Wt1	18S	18.201	8.648	0.430	0.742
TRUE	A6/G6	NCD3	Wt1	18S	18.366	8.991	0.253	0.839
TRUE	A7/G7	HFD1D1	Wt1	18S	18.634	8.699	0.813	0.569
TRUE	A8/G8	HFD1D2	Wt1	18S	18.674	8.756	0.795	0.576
TRUE	A9/G9	HFD1D3	Wt1	18S	19.025	8.721	1.181	0.441
TRUE	A10/G10	HFD2D1	Wt1	18S	19.150	8.735	1.292	0.408
TRUE	A11/G11	HFD2D2	Wt1	18S	19.989	8.824	2.042	0.243
TRUE	A12/G12	HFD2D3	Wt1	18S	19.982	9.140	1.719	0.304

Table 2. The decreased level of *Wt1* mRNA in HFD group was reversed after STM2457 treatment.

Chart	Pairing	Sample Name	Targets	References	Mean Cp of Wt1	Mean Cp of 18S	Target/Ref	Normalized
TRUE		NCD1	Wt1	18S	21.056	7.800	0.000	1.000
TRUE	A2/G2	NCD2	Wt1	18S	20.932	7.390	0.287	0.820
TRUE	A3/G3	NCD3	Wt1	18S	20.947	7.205	0.487	0.714
TRUE	A4/G4	NCD+STM1	Wt1	18S	21.213	7.956	0.002	0.998
TRUE	A5/G5	NCD+STM2	Wt1	18S	21.225	7.727	0.243	0.845
TRUE	A6/G6	NCD+STM3	Wt1	18S	21.325	7.938	0.132	0.913
TRUE	A7/G7	HFD1D1	Wt1	18S	21.740	7.523	0.962	0.513
TRUE	A8/G8	HFD1D2	Wt1	18S	21.693	7.161	1.277	0.413
TRUE	A9/G9	HFD1D3	Wt1	18S	21.399	7.583	0.561	0.678
TRUE	A10/G10	HFD+STM1	Wt1	18S	21.021	7.549	0.216	0.861
TRUE	A11/G11	HFD+STM2	Wt1	18S	21.337	7.472	0.609	0.656
TRUE	A12/G12	HFD+STM3	Wt1	18S	21.229	7.490	0.484	0.715

11. I note that researchers within the field of transgenerational epigenetic inheritance or DOHaD will want to see the HFD1 and HFD2 groups also in most of the figures (or in Supp information). For example, it would be interesting to know whether single- or double-generation HFD have effects on sperm and testicular parameters, retinoic acid metabolism gene expression, WT1 (and *Mettl3*) in HFD2 groups. Perhaps the authors can comment on this.

Response: Thanks for reviewer's comment and suggestion. The effects of paternal single- or double-generation HFD exposure on sperm and testicular parameters, retinoic acid level, WT1 and METTL3 expression in offspring mice were further investigated. As present in **Figs. S8-S10**, results showed that the gradual reduction in sperm count, the gradual impairment in testicular germ cell development, the gradual inhibition in retinoic acid synthesis, the gradual downregulation in WT1 expression, the gradual upregulation in METTL3 expression and the gradual elevation in m6A level were observed in offspring testes with the increase of HFD generation. The corresponding information have been added in the revised manuscript.

The detailed revision are as follows,

In the revised Results section (Lines 416-424)

Elevated sperm m6A level and decreased sperm concentration is observed in overweight/obesity donors

The effects of paternal single- or double-generation HFD exposure on sperm and testicular parameters, retinoic acid level, WT1 and METTL3 expression in offspring mice were investigated. Analogously, the gradual reduction in sperm count, the gradual impairment in testicular germ cell development, the gradual inhibition in retinoic acid synthesis, the gradual downregulation in WT1 expression, the gradual upregulation in METTL3 expression and the gradual elevation in m6A level were observed in offspring testes with the increase of HFD generation (Figs. S8-S10). To further verify the relationship among sperm m6A level, sperm concentration and BMI, a case-control study were established.....

In the revised Supplementary materials section (Lines 79-130)

Figure S8

Fig.S8. The effects of paternal single- or double-generation HFD exposure on testicular development and spermatogenesis in offspring mice. F0 generation male mice were fed NC or HFD from 5 weeks to 15 weeks old, and then mated with NC-fed female mice to breed F1 generation. Similarly, F1 generation male mice whose fathers were exposed to HFD continued to be fed HFD for 10 weeks, and mated with normal

female to breed F2 generation. Male mice of F1 and F2 generations named NC, HFD1 or HFD2 group respectively. All mice were euthanized at 15 weeks of age. (A) Experimental design flowchart. The black arrow indicated NC treatment. The purple arrow indicated HFD treatment. (B and C) Epididymal sperm counts were measured. $n=10$, $DOF=29$, $F=15.19$, $P<0.0001$. (D) Testicular weight. $n=10$, $DOF=29$, $F=3.09$, $P=0.0618$. (E and F) Testicular DDX4 protein expression was detected by immunoblotting. $n=4$, $DOF=11$, $F=22.85$, $P=0.0003$. (G and H) Testicular PLZF, C-KIT and SYCP3 protein expression were detected by immunoblotting. $n=4$, $DOF=11$, $F=6.20$ and $P=0.0203$ for PLZF; $F=15.91$ and $P=0.0011$ for C-KIT; $F=11.60$ and $P=0.0032$ for SYCP3. n.s., not significant. $*P < 0.05$; $**P < 0.01$.

Figure S9

Fig.S9. The effects of paternal single- or double-generation HFD exposure on testicular retinoic acid synthesis and WT1 expression in offspring mice. F0

generation male mice were fed NC or HFD from 5 weeks to 15 weeks old, and then mated with NC-fed female mice to breed F1 generation. Similarly, F1 generation male mice whose fathers were exposed to HFD continued to be fed HFD for 10 weeks, and mated with normal female to breed F2 generation. Male mice of F1 and F2 generations named NC, HFD1 or HFD2 group respectively. All mice were euthanized at 15 weeks of age. (A) Testicular retinoic acid level was detected by ELISA. $n=4$, $DOF=11$, $F=30.91$, $P<0.0001$. (B and C) Testicular RAR α and STRA8 protein expression were measured by immunoblotting. $n=4$, $DOF=11$, $F=26.55$ and $P=0.0002$ for RAR α ; $F=24.69$ and $P=0.0002$ for STRA8. (D-F) Testicular ALDH1A1 and ALDH1A2 protein expression were measured by immunoblotting. $n=4$, $DOF=11$, $F=73.44$ and $P<0.0001$ for ALDH1A1; $F=24.17$ and $P=0.0002$ for ALDH1A2. (G and H) Testicular WT1 protein expression was measured by immunoblotting. $n=4$, $DOF=11$, $F=23.93$, $P=0.0002$. (I) Testicular *Wt1* mRNA level was tested using RT-PCR. $n=4$, $DOF=11$, $F=20.85$, $P=0.0004$. (J) Representative testicular images of immunofluorescent staining for WT1 and SOX9. Scale bar, 20 μ m. Hoechst33258 were used to tag the nuclei. * $P < 0.05$; ** $P < 0.01$.

Figure S10

Fig.S10. The effects of paternal single- or double-generation HFD exposure on

testicular METTL3 expression and *Wtl* m6A level in offspring mice. F0 generation male mice were fed NC or HFD from 5 weeks to 15 weeks old, and then mated with NC-fed female mice to breed F1 generation. Similarly, F1 generation male mice whose fathers were exposed to HFD continued to be fed HFD for 10 weeks, and mated with normal female to breed F2 generation. Male mice of F1 and F2 generations named NC, HFD1 or HFD2 group respectively. All mice were euthanized at 15 weeks of age. (A) Testicular total RNA m6A level was measured. $n=4$, $DOF=11$, $F=18.14$, $P=0.0007$. (B and C) Testicular METTL3 protein expression was detected by immunoblotting. $n=4$, $DOF=11$, $F=20.14$, $P=0.0005$. (D and E) Testicular IGF2BP1 protein expression was detected by immunoblotting. $n=4$, $DOF=11$, $F=38.67$, $P<0.0001$. (F) *Wtl* site1 m6A level was measured by MeRIP-qPCR. $n=4$, $DOF=11$, $F=17.14$, $P=0.0009$. * $P < 0.05$.

12. Regarding mechanism, others have reported that cadmium exposure alters the sperm methylome (e.g. Saintilnord et al Toxicol Sci. 2021 Apr; 180(2): 262–276 – is this worth adding?

Response: Thanks for reviewer's question. In a preliminary experiment, we measured the levels of RNA m6A and DNA 5-mC in mouse sperm after exposure to HFD. As present in **Figs. S13A and B**, sperm m6A levels were increased by a larger fold after HFD exposure compared with 5-mC. In addition, little studies have explored the role of sperm m6A modification in the transmission of paternal-inherited diseases, although previous studies found that paternal exposure to harmful factors, such as cadmium and phthalate, impaired the health of their offspring by altering sperm DNA methylation ^[1, 2]. In the present study, we aim to investigate the role of sperm m6A modification in multigenerational paternal HFD-impaired testicular development and spermatogenesis. Our results showed that METTL3-mediated sperm m6A modification was one of the main causes of multigenerational paternal HFD-induced testicular spermatogenesis disorder in offspring. Further studies are warranted to explore the role of altered sperm DNA methylation in the spermatogenesis

impairment of offspring induced by multigenerational paternal HFD.

The detailed revision are as follows,

In the revised Fig. S13 section

Fig.S13. The effect of HFD exposure on the levels of RNA m6A and DNA 5-mC in mouse sperm. F0 generation male mice were fed NC or HFD from 5 weeks to 15 weeks old. Mouse sperm were obtained. (A) Standardized m6A level. (B) Standardized 5-mC level. $n = 4$. * $P < 0.05$ vs NC.

In the revised Discussion section (Lines 512-513)

Paternally acquired phenotypes are transmitted through sperm to offspring^{19, 50}. Previous studies found that paternal exposure to harmful factors impaired the health of their offspring by altering sperm DNA methylation and histone modification^{51, 52, 53}. However, few studies have explored the role of sperm m6A modification in the transmission of paternal-inherited diseases.....

[1] Saintilnord WN, Tenlep SYN, Preston JD, et al. Chronic Exposure to Cadmium Induces Differential Methylation in Mice Spermatozoa. *Toxicol Sci.* 2021;180(2):262-276.

[2] Oluwayiose OA, Marcho C, Wu H, et al. Paternal preconception phthalate exposure alters sperm methylome and embryonic programming. *Environ Int.* 2021;155:106693.

13. The study would be strengthened further with data from human studies such

as NC, HFD and HFD1+HFD sperm m6A levels in human donors.

Response: Thanks for reviewer's kind suggestion. To further verify the relationship among sperm m6A level, sperm concentration and overweight/obesity, a case-control study were established. A total of 428 human sperm were obtained from the reproductive medicine center of the First Affiliated Hospital of Anhui Medical University with the donor's informed consent. After removal of smoking or alcohol drinking donors, a case-control study containing 30 pairs were obtained by body mass index (BMI), sperm concentration and age. Sperm m6A levels were measured in this case-control study. As shown in Fig. 10, elevated sperm m6A level and decreased sperm concentration was observed in overweight/obesity donors.

The detailed revision are as follows,

In the revised Results section (Lines 424-443)

Elevated sperm m6A level and decreased sperm concentration is observed in overweight/obesity donors

..... To further verify the relationship among sperm m6A level, sperm concentration and BMI, a case-control study were established. As presented in Figs. 10A and B, the BMI of the case group was higher than that of the control group, but the sperm concentration was lower. Figure C showed that sperm m6A levels were elevated in the case group compared to the control group. The correlations of sperm m6A level and BMI or sperm concentration were further analyzed. The positive association was observed between sperm m6A level and BMI ($r = 0.57$, $P < 0.01$; Fig. 10E). The negative correlation was also observed between sperm m6A level and sperm concentration ($r = -0.51$, $P < 0.01$; Fig. 10F). The above results suggest that elevated sperm m6A level and decreased sperm concentration is observed in overweight/obesity donors

Fig.10. Elevated sperm m6A level and decreased sperm concentration is observed in overweight/obesity donors. After removal of smoking or alcohol drinking donors, a case-control study containing 30 pairs were obtained by matching body mass index (BMI), sperm concentration and age. (A) BMI. $n=30, t=-11.20, P<0.0001$. (B) Sperm concentration. $n=30, t=10.50, P < 0.0001$. (C) Sperm m6A level. $n=30, t=-2.74, P=0.0083$. (D) The association between sperm concentration and BMI. (E) The

association between sperm m6A level and BMI. (F) The association between sperm m6A level and sperm concentration. (G) Graphical abstract. * $P < 0.05$; ** $P < 0.01$.

In the Materials and Methods section (Lines 615-625)

Case-control study

To investigate the effect of overweight/obesity on sperm m6A levels, a case-control study was established. A total of 428 human sperm were obtained from the reproductive medicine center of the First Affiliated Hospital of Anhui Medical University with the donor's informed consent. After removal of smoking or alcohol drinking donors, a case-control study containing 30 pairs were obtained by matching body mass index (BMI), sperm concentration and age. Semen specimens were centrifuged at 600 g for 10 min to remove seminal plasma. After PBS resuspension, sperm were incubated in somatic cell lysate (0.5% Triton X and 0.1% sodium dodecyl sulfate in nuclease-free water) at 4 °C for 10 min to remove somatic cells. After washing, sperm pellets were collected for m6A deletion.

14. The final concluding sentence makes rather a sweeping statement- perhaps this could be modified.

Response: Thanks for reviewer's suggestion. We have rewritten the final concluding sentence in the revised manuscript.

The detailed revision are as follows,

In the revised Conclusion section (Lines 532-536)

In conclusion, our results suggest that multigenerational paternal obesity gradually enhances the susceptibility to spermatogenesis disorders in male offspring via downregulating *Wtl* mRNA expression, and METTL3-mediated sperm m6A modification is involved in paternal HFD-induced a decrease of *Wtl* mRNA stability in offspring testes (Fig. 10G). As result of these findings, it opens an avenue for mechanistic exploration of paternally acquired male subfertility and intergenerational transmission, including m6A modification and *Wtl* expression. Among them, sperm m6A modification may be one of the genetic markers for early screening of paternally

acquired male subfertility.

Minor points

15. The abstract may contain a typo, 'mechanically' should probably be 'mechanistically'.

Response: Thanks for reviewer's kind correction. The word "mechanically" has been corrected as "mechanistically" in the revised Abstract section.

The detailed revision is as follows,

In the revised Abstract section (Lines 31)

.....Additionally, multigenerational paternal HFD gradually increased methylase METTL3 and *Wtl* m6A levels in offspring testes. Mechanistically, paternal treatment using STM2457, a specific inhibitor of METTL3 activity, restored HFD-reduced sperm count, and reversed HFD-increased *Wtl* m6A in offspring testes.....

16. Line 147 and 149 Fig 2C and 2D should be 3C and 3 D

Response: Thanks for reviewer's kind correction. Fig 2C and 2D have been corrected as Fig 3C and 3D in the revised Results section.

The detailed revision is as follows,

In the revised Results section (Lines 158-169)

.....Compared with HFD1D group and HFD2D group, 229 mRNAs were upregulated and 268 mRNAs were downregulated, screened for a 1.2-fold change and adjusted with $P < 0.05$ (Figs. S3A-C). As illustrated in Figs. 3A and B, GO analysis revealed that downregulated mRNAs were related to multiple biological processes, including the "retinol metabolic process," the "reproductive process," and the "spermatogenesis"..... Nevertheless, the levels of serum vitamin A (retinol) and testicular retinol-binding protein 4 (RBP4) were not reduced among the four groups (Figs. S4A-C). Notably, a gradual decrease of testicular retinoic acid level was observed in the offspring with the increase of HFD generation (Fig. 3C). Further experiments showed that RAR α and STRA8 proteins expression were gradually downregulated in

offspring testes with the increase of HFD generation (Figs. 3D and E). To further explore and verify the expression of retinoic acid synthetase and retinoic acid metabolic enzyme in the four groups, the analysis of gene expression was conducted using RT-qPCR.....

17. Line 183 'conformably' perhaps another word can be substituted there (eg 'In line with this,')

Response: Thanks for reviewer's kind suggestion. The word "conformably" has been substituted as " In line with this" in the revised Results section.

The detailed revision is as follows,

In the revised Results section (Lines 209)

.....As shown in Figs. 4B-D, the mRNA and protein levels of WT1 were significantly reduced in the HFD1D group, and persistently lowered in the HFD2D group compared to those of the NC and NCD mouse testes. **In line with this**, the gradual reductions of testicular WT1-positive cells were observed in the offspring with the increase of HFD generation (Figs. 4E and F).....

18. Line 372 'Nevertheless' is repeated.

Response: Thanks for reviewer's kind correction. The repeated word "Nevertheless" has been deleted in the revised Discussion section.

19. Line 420 'The modification of m6A involves in testicular ...' - review grammar.

Response: Thanks for reviewer's suggestion. The incorrect sentence "The modification of m6A involves in testicular development and spermatogenesis" has been corrected as "m6A modification participates in the processes of testicular development and spermatogenesis" in the revised Discussion section.

The detailed revision is as follows,

In the revised Discussion section (Lines 493-494)

.....Methylases (METTL3 and METTL14) and demethylases (ALKBH5 and FTO)

regulate the balance of m6A in mammal^{40, 41, 42}. m6A modification participates in the processes of testicular development and spermatogenesis^{43, 44}. This study found that multigenerational paternal HFD gradually exacerbated Cd-increased METTL3 and m6A in offspring testes.....

20. Line 456 remove 'a'

Response: Thanks for reviewer's kind correction. The word "a" has been removed in the revised Discussion section.

21. Why is Cd used to indicate cadmium exposure in Figure 7 but 'D' in the previous figures?

Response: Thanks for reviewer's question. In animal studies, cadmium exposure was labeled "D" due to the variety of treatments in mice. In the cell experiment, the treatment mode is relatively simple, and cadmium exposure is labeled "Cd".

Reviewer #2 (Remarks to the Author):

The study by Xiong et al explores the role that a paternal high fat diet has on offspring reproductive fitness and spermatogenesis in a mouse model. The authors report a role for the transcription factor Wt1 and retinoic acid in regulating offspring sperm production. While there is merit in the study, I have several major and minor issues that need to be addressed.

1. First, could the authors provide data on litter sizes, pregnancy rates and general fertility measurements in their mice? While i appreciate there is a decline in sperm number, this doesn't necessarily translate into a decline in fertility. therefore, data showing whether mating, embryo number, litter sizes and or offspring survival are necessary to place the changes in sperm number into context.

Response: Thanks for reviewer's question and suggestion. The data of multigenerational paternal HFD exposure on the fertility rate, pregnancy rate and litter size in mice were provided. As shown in Fig. 1B-

C and Fig. S2A, the gradual reductions of fertility rate and pregnancy rate were observed in mice with the increase of HFD generation, but not embryo number and litter size. In addition, multigenerational paternal HFD gradually reduced the body weight of offspring mice, but had no effect on their survival (**Fig. S2B**).

The detailed revision are as follows,

In the revised Results section (Lines 83-86)

Multigenerational paternal HFD gradually enhances susceptibility to spermatogenesis disorder in their offspring

..... Furthermore, HFD exposure induced fasting blood glucose elevation, glucose tolerance impairment, hepatic lipid deposition and serum triglyceride (TG) increase in mice (Figs. S1D-H). **The effect of multigenerational paternal HFD exposure on the fertility rate, pregnancy rate and litter size in mice were then investigated. As presented in Fig. 1A-C and Fig. S2A, the gradual reductions of fertility rate were observed in mice with the increase of HFD generation, but not litter size.** The effect of paternal HFD exposure on the susceptibility of environmental stress-induced spermatogenesis

Fig.1. Multigenerational paternal HFD enhances susceptibility to spermatogenesis disorder in offspring.(A) Experimental design flowchart of Cd-induced spermatogenesis impairment in mice with HFD-feeding. The black arrow indicated NC treatment. The purple arrow indicated HFD treatment. The red arrow indicated Cd treatment. (B) The fertility rate was calculated. $n=4$, Degree of Freedom (DOF)=11, $F=18.20$, $P=0.0007$. (C) The pregnancy rate was counted. $n=4$, $P=0.3911$. (D and E) Epididymal sperm counts were measured. $n=10$, DOF=39, $F=37.68$, $P<0.0001$. (F) Sperm motility was recorded; $n=10$, DOF=39, $F=14.01$, $P<0.0001$. (G) Testicular H&E staining. (H) The number of Testicular seminiferous tubules number at different stages were evaluated. $n=4$, DOF=15, $F=30.25$, $P<0.0001$. n.s., not significant. $*P < 0.05$; $**P < 0.01$.

In the revised Fig. S2 section (Lines 11-18)

Figure S2

Fig.S2. The effect of multigenerational paternal HFD on litter size and body weight

in offspring. F0 generation male mice were fed NC or HFD from 5 weeks to 15 weeks

old, and then mated with NC-fed female mice to breed F1 generation. Similarly, partial

F1 generation male mice continued to be fed HFD for 10 weeks, and mated with normal

female to breed F2 generation. (A) Litter size. $n=10$, $DOF=29$, $F=0.48$, $P=0.6246$. (B)

Body weight of male. $n = 10$. $*P < 0.05$.

In the revised Materials and Methods section (Lines 626-632)

Fertility test

For fertility rate, one male mouse was mated with two female mice at night, and vaginal

plug were checked in the next morning. The number of fertile male mice was recorded

and divided by the total number of male mice to calculate the fertility rate of each group

on that day. This test was repeated twice again every 4 days, a total of 4 repetitions. For

pregnant rate, female mice with vaginal plug were fed at single caged, and the

pregnancy results were recorded. The litter sizes were also observed after delivery.

2. While the authors took their experimental offspring groups out to an F2

generation, why did they not do the same with the control (NC) group? I find it odd that they are comparing phenotypic changes in the experimental F2 offspring against a control F1 group.

Response: Thanks for reviewer's question and comment. As presented in **Fig. S11**, our results showed that there were no significant differences in sperm count, retinoic acid level, WT1 expression and m6A modification level in mice between NC1 group and NC2 group. Therefore, we did not show the relevant data of the NC group in F2 generation. In addition, the aim of this study was to investigate the effect and mechanism of multigenerational paternal HFD on the susceptibility of offspring to spermatogenesis disorders. Therefore, the two groups we mainly compared were the HFD1D (offspring were treated with Cd after paternal exposure to one-generational HFD) group and HFD2D (the offspring were treated with Cd after paternal exposure to bi-generational HFD) group.

The detailed revision are as follows,

In the revised Fig. S11 section

Fig.S11. The effects of paternal single- or double-generation HFD exposure on testicular spermatogenesis, retinoic acid level, WT1 expression and m6A modification in offspring mice. F0 generation male mice were fed NC or HFD from 5 weeks to 15 weeks old, and then mated with NC-fed female mice to breed F1 generation. Similarly, F1 generation male mice whose fathers were exposed to HFD continued to be fed HFD for 10 weeks, and mated with normal female to breed F2 generation. Male mice of F1 and F2 generations named NC1, NC2, HFD1 or HFD2 group respectively. All mice were euthanized at 15 weeks of age. (A) Experimental design flowchart. The black arrow indicated NC treatment. The purple arrow indicated HFD treatment. (B and C) Epididymal sperm counts were measured. (D) Testicular retinoic acid level was detected by ELISA. (E-G) Testicular WT1 and METTL3 protein

expression were measured by immunoblotting. (H) Testicular total RNA m6A level was measured. a $P < 0.05$ vs NC1. b $P < 0.05$ vs NC2. c $P < 0.05$ vs HFD1.

Minor comments

3. In line 17, did the authors mean they looked at 'different testicular germ cell markers...'?

Response: Thanks for reviewer's question. In this study, we detected the mRNA and protein levels of different testicular germ cell markers to determine which germ cells were impaired. As presented in Fig. 2D, the mRNA levels of *Izumo3* (elongated spermatids marker), *Acrv1* (round spermatids marker), *Smc3* (spermatocytes marker) and *C-kit* (differentiating-spermatogonia marker) were gradually reduced with the increase of HFD generation, but not *Plzf* (undifferentiated-spermatogonia marker). Correspondingly, the C-KIT and SYCP3 (spermatocytes marker) proteins expression were also downregulated in the HFD1D and HFD2D groups (Figs. 2E and F). These results suggested that the mRNA and protein levels of differentiating-spermatogonia and subsequent germ cell marker were gradually reduced in offspring with the increase of HFD generation. The details have been included in the revised manuscript.

The detailed revision is as follows,

In the revised Abstract section (Lines 25)

.....However, the effects of multigenerational paternal obesity on the susceptibility of cadmium (a reproductive toxicant)-induced spermatogenesis disorders in offspring remain unknown. Here, results showed that sperm count, testicular germ cell marker and retinoic acid (RA) levels were gradually reduced in offspring with the increase of high-fat diet (HFD) generation. Furthermore, we identified that WT1, RA synthetases upstream transcription factor, was decreased over generations.....

In the Figs. 2D-F section (Lines 143-149)

Fig.2. Multigenerational paternal HFD exacerbates environmental stress-impaired testicular germ cell development in offspring..... (B and C) Testicular DDX4 protein expression was detected by immunoblotting. $n=6$, $DOF=23$, $F=25.05$, $P < 0.0001$. (D) Testicular *Izumo3*, *Acrv1*, *Smc3*, *C-kit* and *Plzf* mRNA levels were detected by RT-PCR. $n=6$, $DOF=23$, $F=10.68$ and $P=0.0002$ for *Plzf*; $F=16.53$ and $P < 0.0001$ for *C-kit*; $F=25.24$ and $P < 0.0001$ for *Smc3*; $F=31.55$ and $P < 0.0001$ for *Acrv1*; $F=29.16$ and $P < 0.0001$ for *Izumo3*. (E and F) Testicular PLZF, C-KIT and SYCP3 protein expression were detected by immunoblotting. $n=6$, $DOF=23$, $F=7.15$ and $P=0.002$ for PLZF; $F=27.78$ and $P < 0.0001$ for C-KIT; $F=24.44$ and $P < 0.0001$ for

SYCP3. n.s., not significant. * $P < 0.05$; ** $P < 0.01$.

4. In line 23, did the authors mean 'mechanistically' rather than 'mechanically'?

Response: Thanks for reviewer's question and kind correction. We mean "mechanistically" rather than "mechanically". The word "mechanically" has been corrected as "mechanistically" in the revised Abstract section.

The detailed revision is as follows,

In the revised Abstract section (Lines 31)

.....Additionally, multigenerational paternal HFD gradually increased methylase METTL3 and *Wtl* m6A levels in offspring testes. **Mechanistically**, paternal treatment using STM2457, a specific inhibitor of METTL3 activity, restored HFD-reduced sperm count, and reversed HFD-increased *Wtl* m6A in offspring testes.....

5. Line 32, did the authors mean that the number of obese people is on the rise, rather than the people themselves being on the rise?

Response: Thanks for reviewer's question and kind correction. We mean that the number of obese and overweight people are on the rise, rather than the people themselves being on the rise. The sentence "obese and overweight people are on the rise" has been corrected as "the number of obese and overweight people are on the rise" in the revised Introduction section.

The detailed revision is as follows,

In the revised Introduction section (Lines 40)

.....In the last 45 years (1973–2018), global semen quality has decreased by more than 62%¹. Globally, **the number of obese and overweight people are on the rise**². Body mass index (BMI) and the quality of semen were negatively correlated in population surveys^{3,4}.....

6. Lines 40-44, the introduction of the transgenerational concept feels very clunky and i feel that a reference focused on transgenerational reproductive

changes would be more appropriate.

Response: Thanks for reviewer's kind suggestion. The reference focused on transgenerational reproductive changes has been added in the revised Introduction section.

The detailed revision is as follows,

In the revised Introduction section (Lines 48-50)

.....Based on the above studies, paternal obesity may lead to spermatogenesis disorders in offspring. Recent study found that parental obesity intergenerationally induced reproductive damages in offspring¹¹. Nevertheless, the intergenerational effect of multigenerational paternal obesity on the susceptibility to spermatogenesis disorders in offspring and its mechanism remains unknown.

7. The rationale for using cadmium in this study is not explained.

Response: Thanks for reviewer's comment. Cadmium (Cd), a well-known reproductive toxicant, impairs testicular development and spermatogenesis. Population-based studies found that Cd concentrations in blood and semen were negatively correlated with sperm quality [1,2]. A series of animal experiments also confirmed that Cd exposure impaired testicular development and reduced sperm count in mice [3,4]. In this study, we used Cd as an inducer of reproductive toxicity to investigate the effect of multigenerational paternal HFD on the susceptibility to spermatogenesis disorders in offspring. The detailed reason for using Cd have been added to the revised manuscript.

The detailed revision are as follows,

In the revised Abstract section (Lines 23-24)

There is strong evidence that obesity is a risk factor for poor semen quality. However, the effects of multigenerational paternal obesity on the susceptibility of cadmium (a reproductive toxicant)-induced spermatogenesis disorders in offspring remain unknown.....

In the revised Materials and Methods section (Lines 566-571)

.....Then, 10 male mice from each group of F1 and F2 generations were exposed to CdCl₂ (0 or 100 mg/L) by drinking water for 10 weeks, and named NC (normal control), NCD (the offspring were treated with Cd after paternal treatment with normal chow), HFD1D (the offspring were treated with Cd after paternal exposure to one-generational HFD) or HFD2D (the offspring were treated with Cd after paternal exposure to bi-generational HFD) groups, respectively. Cd, a well-known reproductive toxicant, impairs testicular development and spermatogenesis. Population-based studies found that Cd concentrations in blood and semen were negatively correlated with sperm quality^{55,56}. A series of animal experiments also confirmed that Cd exposure impaired testicular development and reduced sperm count in mice^{18,57}. Here, Cd was used as an inducer of reproductive toxicity to induce spermatogenesis disorders in mice. It's well-known that Cd exposure impairs testicular spermatogenesis mainly via the dose of internal exposure.

[1] He Y, Zou L, Luo W, et al. Heavy metal exposure, oxidative stress and semen quality: Exploring associations and mediation effects in reproductive-aged men. *Chemosphere*. 2020; 244: 125498.

[2] Huang X, Zhang B, Wu L, et al. Association of Exposure to Ambient Fine Particulate Matter Constituents With Semen Quality Among Men Attending a Fertility Center in China. *Environ Sci Technol*. 2019;53(10):5957-5965.

[3] Li X, Yao Z, Yang D, et al. Cyanidin-3-O-glucoside restores spermatogenic dysfunction in cadmium-exposed pubertal mice via histone ubiquitination and mitigating oxidative damage. *J Hazard Mater*. 2020; 387:121706.

[4] Xiong YW, Tan LL, Zhang J, et al. Combination of high-fat diet and cadmium impairs testicular spermatogenesis in an m6A-YTHDF2-dependent manner. *Environ Pollut*. 2022;313:120112.

8. In line 72, which 'white fat' depot was measured? And if it was only one then why didn't they look at the others.

Response: Thanks for reviewer's questions. In line with previous study [1], epididymal white fat weight was one of the indexes to evaluate HFD-induced obesity phenotype in mice, which combines with impaired

glucose tolerance, elevated body weight, increased hepatic lipid deposition and serum triglyceride, suggesting successful induction of obesity phenotype in mice. In the present study, we only measured epididymal white fat in mice, and we will supplement others in future experiments.

The detailed revision are as follows,

In the revised Results section (Lines 80)

.....The obesity phenotype was first identified in male mice. As presented in Figs. S1A-C, body weight and epididymal white fat weight in male mice increased after HFD exposure.....

In the revised Supplementary materials section (Lines 5)

Fig.S1. HFD induces obesity phenotype in mice. F0 generation male mice were fed NC or HFD from 5 weeks to 15 weeks old. (A) Body weight of mice. $n=10$, $t=-4.59$, $P=0.0008$. (B) The weight of **epididymal** white fat. $n=10$, $t=-6.45$, $P<0.0001$* $P < 0.05$; ** $P < 0.01$ vs NC.

[1] Coyne ES, Bédard N, Gong YJ, Faraj M, Tchernof A, Wing SS. The deubiquitinating enzyme USP19 modulates adipogenesis and potentiates high-fat-diet-induced obesity and glucose intolerance in mice. *Diabetologia*. 2019; 62(1):136-146.

9. Throughout the manuscript, the authors use terms such as '...paternal exposure to multigenerational HFD...' (See line 85). However, this makes it sound like it was the previous generations, before the F0 father, who were exposed to the HFD. This is not the case and so the wording of such phrases needs to be changed to better reflect the design of the study.

Response: Thanks for reviewer's comment and suggestion. The phrases of "...paternal exposure to multigenerational HFD..." have been changed to "...multigenerational paternal HFD..." in the revised manuscript.

The detailed revision are as follows,

In the revised Results section (Lines 77-367)

Multigenerational paternal HFD gradually enhances susceptibility to spermatogenesis disorder in their offspring.

..... The above results indicate that **multigenerational paternal HFD** gradually enhances the susceptibility to spermatogenesis disorder in their offspring.

10. Similarly, I am unsure what the authors mean by a 'partial F1 generation' such as in line 91.

Response: Thanks for reviewer's comment. We mean that some of the males in F1 generation whose fathers were exposed to HFD continued to be treated with HFD, while others were exposed to Cd.

The detailed revision are as follows,

In the revised Materials and Methods section (Lines 558-560)

.....After mating, 10 pregnant mice were obtained in each group. Similarly, 10 F1 generation male mice (whose fathers were exposed to HFD) from different litter continued to be fed HFD for 10 weeks, and mated with normal females to breed F2 generation.....

11. I believe that the more accurate name for MVH (line 105) is Ddx4.

Response: Thanks for reviewer's comment. The word of "MVH" has been corrected as "DDX4" in the revised manuscript.

The detailed revision are as follows,

In the revised Results section (Lines 118)

.....It is well known that spermatogenesis is determined by the development of testicular germ cells. Compared to the group, testicular weight and DDX4 (marker of testicular germ cell) protein level were obviously lowered in the HFD1D group, which was further reduced in the HFD2D group (Figs. 2A-C).

In the revised Fig.2B and C section (Lines 141-142)

Fig.2. Multigenerational paternal HFD exacerbates environmental stress-impaired testicular germ cell development in offspring. All mice were euthanized at 15 weeks of age. (A) Testicular weight. $n=10$, $DOF=39$, $F=8.12$, $P=0.0003$. (B and C) Testicular DDX4 protein expression was detected by immunoblotting. $n=6$, $DOF=23$, $F=25.05$, $P<0.0001$. (D) Testicular *Izumo3*, *Acrv1*, *Smc3*, *C-kit* and *Plzf* mRNA levels were detected by RT-PCR.....

12. IN Figure 3, I find it odd that after conducting RNA-seq on the offspring testes, the authors then conduct additional analysis of gene expression using RT-qPCR. Why didn't they check the expression of the genes in panel F within

their RNA-Seq data?

Response: Thanks for reviewer's comment and question. In RNA-Seq data, GO analysis revealed that downregulated mRNAs were related to "retinol metabolic process" (Fig. 3B). The retinol metabolic-related genes were presented in Fig. S2D. Results showed that the expressions of *Aldh1a1*, *Aldh1a2*, *Aldh1a3*, *Rara* and *Stra8* were downregulated in the HFD1D group compared with the HFD2D group, while the expressions of *Rdh10*, *Cyp26a1*, *Cyp26b1* and *Cyp26c1* were unchanged (Fig. S2D). To further explore and verify the expression of retinoic acid synthetase and retinoic acid metabolic enzyme in the four groups (NC, NCD, HFD1D and HFD2D groups), the analysis of gene expression was conducted using RT-qPCR. The results presented that *Aldh1a1*, *Aldh1a2* and *Aldh1a3* mRNA levels were lowered in HFD1D and HFD2D groups, and *Aldh1a1* and *Aldh1a2* levels in HFD2D group were less than those in HFD1D group (Fig. 3F).

The detailed revision are as follows,

In the revised Results section (Lines 161-172)

Multigenerational paternal HFD gradually aggravates environmental stress-inhibited testicular retinoic acid synthesis in offspring

.....As illustrated in Figs. 3A and B, GO analysis revealed that downregulated mRNAs were related to multiple biological processes, including the "retinol metabolic process," the "reproductive process," and the "spermatogenesis". The retinol metabolic-related genes were presented in Fig. S3D. Results showed that the expressions of *Aldh1a1*, *Aldh1a2*, *Aldh1a3*, *Rara* and *Stra8* were downregulated in the HFD1D group compared with the HFD2D group. Nevertheless, the levels of serum vitamin A (retinol) and testicular retinol-binding protein 4 (RBP4) were not reduced among the four groups (Figs. S4A-C). Notably, a gradual decrease of testicular retinoic acid level was observed in the offspring with the increase of HFD generation (Fig. 3C). Further experiments showed that RAR α and STRA8 proteins expression were gradually downregulated in

offspring testes with the increase of HFD generation (Figs. 3D and E). To further explore and verify the expression of retinoic acid synthetase and retinoic acid metabolic enzyme in the four groups, the analysis of gene expression was conducted using RT-qPCR. The results presented that *Aldh1a1*, *Aldh1a2* and *Aldh1a3* mRNA levels were lowered in HFD1D and HFD2D groups, and *Aldh1a1* and *Aldh1a2* levels in HFD2D group were less than those in HFD1D group (Fig. 3F).....

Fig.3. Multigenerational paternal HFD gradually aggravates environmental stress-inhibited testicular retinoic acid synthesis in offspring. (A and B) GO enrichment analysis of differentially expressed mRNAs in mouse testes between HFD1D and HFD2D groups. (C) Testicular retinoic acid level was detected by ELISA. $n=4$, $DOF=15$, $F=41.13$, $P<0.0001$. (D and E) Testicular RAR α and STRA8 protein expression were measured by immunoblotting. $n=6$, $DOF=23$, $F=43.08$ and $P<0.0001$ for RAR α ; $F=46.65$ and $P<0.0001$ for STRA8. (F) Testicular *Aldh1a1*, *Aldh1a2*, *Aldh1a3*, *Rdh10*, *Cyp26a1*, *Cyp26b1* and *Cyp26c1* mRNA levels were tested by RT-PCR. $n=6$, $DOF=23$, $F=32.26$ and $P<0.0001$ for *Aldh1a1*; $F=27.78$ and $P<0.0001$ for *Aldh1a2*; $F=25.82$ and $P<0.0001$ for *Aldh1a3*; $F=11.75$ and $P=0.0001$ for *Rdh10*; $F=0.18$ and $P=0.9063$ for *Cyp26a1*; $F=0.78$ and $P=0.5217$ for *Cyp26b1*; $F=3.20$ and $P=0.0459$ for *Cyp26c1*. (G–I) Testicular ALDH1A1 and ALDH1A2 proteins were measured using immunoblotting. $n=6$, $DOF=23$, $F=61.35$ and $P < 0.0001$ for ALDH1A1; $F=20.01$ and $P<0.0001$ for ALDH1A2. n.s., not significant. * $P < 0.05$; ** $P < 0.01$.

In the revised Fig. S3D section (Lines 32)

Fig.S3. Multigenerational paternal HFD gradually exacerbates environmental stress-induced testicular dysregulated expression of mRNAs in offspring.(A) The distribution of differentially expressed mRNAs in testes were presented. **(B)** The number of up-regulated and down-regulated mRNAs were shown. **(C)** Heatmap of differentially expressed mRNAs in testes were presented. **(D) Heatmap of retinol metabolic process-related mRNAs in testes were presented.**

13. In Figure 4E, is that the average number of WT1+ cells per tubule or per unit area? It would be of interest to see which cells are expressing WT1 through some double staining for Sox9 if the authors think its in the Sertoli cells.

Response: Thanks for reviewer's question and suggestion. In Fig. 4E,

the average number of WT1 positive cells per tubule were recorded. In addition, the results of double staining of WT1 and SOX9 showed that WT1 was mainly located in the Sertoli cells (In Fig. S9J).

The detailed revision are as follows,

In the revised Fig. 4E section (Lines 225-226)

Fig.4. Multigenerational paternal HFD gradually exacerbates environmental stress-downregulated testicular WT1 expression in offspring..... (C and D)

Testicular WT1 protein expression was measured by immunoblotting. $n=6$; $P<0.0001$;

$F=21.59$; $DOF=23$. (E and F) The number of testicular $WT1^+$ positive cells per tubule were counted by immunohistochemistry. $n=6$, $DOF=23$, $F=23.66$, $P<0.0001$. $*P <$

0.05 ; $**P < 0.01$.

In the revised Fig. S9J section (Lines 113-114)

Fig.S9. The effects of paternal single- or double-generation HFD exposure on testicular retinoic acid synthesis and WT1 expression in offspring mice..... ((J)

Representative testicular images of immunofluorescent staining for WT1 and SOX9.

Scale bar, 20 μm . Hoechst33258 were used to tag the nuclei. * $P < 0.05$; ** $P < 0.01$.

14. For the analysis of retinoic acid levels, why have the authors normalized the control group to 1? As this was an ELISA, why not display it in the units of RA?

Response: Thanks for reviewer's questions. To easily observe the multiples of increase and decrease, the level of retinoic acid in control group was normalized to 1 in the original manuscript. At present, the normalized data has been corrected to the detection level in the revised manuscript.

The detailed revision are as follows,

In the revised Fig. 3C section (**Lines 188-189**)

Fig.3. Multigenerational paternal HFD gradually aggravates environmental stress-inhibited testicular retinoic acid synthesis in offspring..... (C) Testicular retinoic acid level was detected by ELISA. $n=4$, $DOF=15$, $F=41.13$, $P<0.0001$. (D and E) Testicular RAR α and STRA8 protein expression were measured by immunoblotting.....

15. When the authors refer to siR (e.g, line 283) do the authors mean siRNA?

Response: Thanks for reviewer's question. The siR means siRNA. It has been marked in the revised manuscript

The detailed revision is as follows,

In the revised Results section (**Lines 316**)

.....Compared with the WT sequence, the MUT sequence markedly reversed Cd-reduced *Wt1* level in Sertoli cells (Fig. 7D). Further studies found ***Mettl3* siRNA (siR)** pretreatment markedly restored Cd-elevated *Wt1* site1 m6A level, and reversed Cd-lowered WT1 protein and mRNA level in Sertoli cells (Figs. 7E-H).....

16. In figure 8F, the genes in blue are regarded as being hypermethylated but the genes in yellow are regarded as being down regulated? These needs amending so that its either hyper/hypo or up/down regulated.

Response: Thanks for reviewer's question and comment. In this study, **m6A-mRNA&IncRNA Epitranscriptomic Microarray analyses were performed on mouse sperm. Hyper-methylated mRNAs were chosen from Differentially Methylated Genes, while downregulated mRNAs were chosen from Differentially Expressed Genes. Since the previous results revealed that mRNA hypermethylation might lead to downregulation of its expression, the intersection between hyper-methylated mRNAs and downregulated mRNAs was performed.**

The details are as follows,

In the Fig. 8F section (**Lines 378-379**)

Fig.8. Multigenerational paternal HFD gradually increases m6A level and downregulates *Wt1* mRNA expression in paternal sperm..... (E) The distribution of differentially expressed mRNAs in sperm were presented. (F) **Hyper-methylated mRNAs and down-regulated mRNAs in sperm were intersected.** (G) Heatmap of overlapping mRNAs in sperm was depicted. (H) Sperm *Wt1* mRNA expression was tested using RT-PCR. $n=4$, $DOF=11$, $F=21.19$, $P=0.0004$. $*P < 0.05$.

17. The Materials and Methods lacks significant amounts of information.

Response: Thanks for reviewer's comment and suggestion. The lacked information in Materials and Methods have been add into revised

manuscript.

The detailed revision are as follows,

In the revised Materials and Methods section (**Lines 626-764**)

MATERIALS AND METHODS

Chemicals and Reagents

Detailed information on chemical reagents and primary antibodies are listed in Table S1.

.....

Fertility test

For fertility rate, one male mouse was mated with two female mice at night, and vaginal plug were checked in the next morning. The number of fertile male mice was recorded and divided by the total number of male mice to calculate the fertility rate of each group on that day. This test was repeated twice again every 4 days, a total of 4 repetitions. For pregnant rate, female mice with vaginal plug were fed at single caged, and the pregnancy results were recorded. The litter sizes were also observed after delivery.

Glucose tolerance test (GTT)

The F0 generation male mice were intraperitoneally injected with 2.0 g/kg glucose solution after fasting (**maintaining normal water intake**) for 12 h. The blood glucose was then detected in the caudal vein using a glucometer, and recorded at different time points.

Local testicular injection

Local testicular injections were performed according to previous study with corresponding modification⁶⁰. In brief, the mice were injected intraperitoneally with 250 mg/kg 2,2,2-tribromoethanol. After anesthesia, both testes of the mice were found by touch pressure. Then, 20 μ L AAV9 or STM2457 were injected into the middle of each testes using a micropipette. After injection, the mice were placed flat in cage and observed for 30 min until they were fully awake and returned to normal. Among them, 1.03×10^{12} vg/ml AAV9-*Wt1* were injected once into mice at PND35. AAV9-*NC* was used as control. AAV9-*Wt1* and AAV9-*NC* were synthesized by General Biol Co., Ltd.

STM2457 were injected at a dose of 50.0 mg/kg, once a week for 5 weeks. 20% SBE- β -CD was used as control. STM2457 and SBE- β -CD were purchase from MedChemExpress. The dose of STM2457 was choose according the previous study⁶¹.

Evaluation of sperm count

.....

Sperm isolation

To obtain sperm, fresh mouse epididymis was completely cut into pieces in preheated medium and incubated at 37°C for 10 min until mature sperm were fully released. Then, the tissue debris were removed by 40- μ m cell strainer. The sperm mixture was incubated with somatic cell lysate (0.5% Triton X and 0.1% sodium dodecyl sulfate in nuclease-free water) on ice for 40 min to clear somatic cell. Next, the mixture was centrifuged at 600 g for 5 minutes and the precipitates were resuspended with PBS. After washing twice, sperm pellets were collected for the methylated RNA microarray.

MTT assay

First, TM4 cells were treated with different concentrations CdCl₂ for 6, 12, or 24 h after culture in 96 well plates for 12 h. Following that, the cells were treated with 5 mg/ml MTT, and were persistently cultured for 2 h in a 37 °C incubator. Prior to this, the cells were grown in DMEM/F-12 medium and supplemented with 5% horse serum, 2.5% fetal bovine serum and 1% penicillin-streptomycin solution. After aspirating the medium, DMSO was added to the culture plate. As a final step, the cellular absorbance at 570 nm was detected by a multimode reader (SYNERGY4, Biotek, USA).

Immunohistochemistry

.....

Immunoblotting

The protein lysates from TM4 cells and mouse testes were obtained after homogenizing with RIPA buffer, and Immunoblotting was implemented as previously described⁶². Briefly, cellular and testicular protein extracts (6-30 ug) were separated by 10.0-12.5% SDS-PAGE electrophoresis and then transferred into PVDF membrane. After blocking with 5% skim milk for 1.5 h, the membranes were incubated with

antibodies of MVH (1:1000), PLZF (1:200), C-KIT (1:200), SYCP3 (1:200), STRA8 (1:1000), RAR α (1:1000), RBP4 (1:1000), ALDH1A1 (1:1000), ALDH1A2 (1:1000), WT1 (1:1000), METTL3 (1:2000), METTL14 (1:1000), ALKBH5 (1:1000), FTO (1:1000), YTHDF1 (1:1000), YTHDF2 (1:1000) and IGF2BP1 (1:1000) for 1-3 h at room temperature. Antibody of β -Actin (1:10000) for loading control. After washing, the corresponding secondary antibody were used to incubate membranes for 1-2 h. Finally, the imaging system Bio-Rad ChemiDoc™ MP was used to detect immunoreactive protein signal.

Enzyme-linked immunosorbent assay (ELISA)

The quantity of retinoic acid (RA) in mouse testes and retinol in mouse serum were measured using the mouse RA-ELISA kit (MyBiosource, MBS706971) and the mouse retinol-ELISA kit (CUSABIO, CSB-E07891m), respectively. Assays were implemented according to the manufacturer's specifications. To detect testicular RA, 50 mg of testes was accurately weighed and homogenized in 900 μ l ice PBS with protease inhibitor. After centrifugation at 6000 g for 15 min, the supernatant was collected for RA detection. The concentration of testicular RA was presented as ng/g.

m6A-mRNA&IncRNA Epitranscriptomic Microarray

Sperm total RNA was initially extracted with TRIzol reagent, and quantified using the NanoDrop ND-1000. The methylated RNA epitranscriptomic microarray were performed at Aksomics (Shanghai, China). Briefly, the total RNAs were immunoprecipitated with anti-m6A antibody. The modified RNAs were eluted from the immunoprecipitated magnetic beads as the IP. The unmodified RNAs were recovered from the supernatant as Sup. After Sup and IP RNAs labeling with Cy3 and Cy5 respectively, the cRNAs were combined together and hybridized onto Arraystar Mouse mRNA&IncRNA Epitranscriptomic Microarray. Then, the arrays were scanned in two-color channels by an Agilent Scanner G2505C. Agilent Feature Extraction software was used to analyze the collected chip fluorescence signals. The raw signal strength of IP and Sup was normalized using the mean value of Log₂-converted spik-in RNA fluorescence intensity. After standardization, the difference in m6A methylation levels

between the two groups was calculated using a T-test statistical model. FC (Fold Change) >1.5 and P-value <0.05 were set as screening thresholds for differential expression.

RNA sequencing

Total RNA of mouse testes was extracted, and quality inspections were performed. The sequencing library was constructed with 2.0 μg total RNA from each sample at Aksonomics (Shanghai, China). In brief, mRNA enrichment was performed using NEB Next[®] Poly(A) mRNA Magnetic Isolation Module kit. The library from enriched RNA was constructed by KAPA Stranded RNA-Seq Library Prep Kit (Illumina) and inspected by Agilent 2100 Bioanalyzer. Then, the final quantification of library was conducted by qPCR. After denaturing with 0.1 M NaOH, 8 pM single stranded DNA was amplified in situ at the NovaSeq S4 Reagent Kit. The end of generated fragment was sequenced using Illumina NovaSeq 6000 sequencer for 150 cycles. After StringTie comparison of the sequencing results to the known transcriptome, the transcriptional abundances at Gene Level and Transcript Level were calculated using Ballgown. The unit of expression quantity is expressed by FPKM (Fragments Per Kilobase of gene/transcript model per Million mapped fragments). A gene or transcript with an expression value of FPKM > 0.5 was considered to be effectively expressed in the group. Finally, Ballgown software was used for statistical analysis by the F-test statistical model, FC >1.2 and P-value <0.05 were set as the screening threshold for differential expression.

Isolation of total RNA and real-time RT-PCR

.....

Quantification of m6A modifications

.....

MeRIP qPCR

Sperm total RNA was initially extracted with TRIzol reagent, and mRNA was enriched with Arraystar Seq-Star[™] poly (A) mRNA isolation kit. The interrupted mRNA (1/10 of which was left as the input control) was mixed with 2 μg anti-m6A rabbit polyclonal antibody and IgG Dynabeads into a 500 μL IP reaction system (50 mM Tris-HCl, pH

7.4, 0.1% NP-40, 150 mM NaCl and 40 U/ml RNase inhibitor), and incubated at 4 °C for 2 h. After washing and elution, the binding RNA was leached by TRIzol^{16, 63}. Afterward, reverse transcription and amplification were performed. The SRAMP website (<http://www.cuilab.cn/sramp/>) predicted that *Wt1* mRNA has four highly credible m6A-modified sites. The corresponding primer sequences are shown in Table S3. The different site m6A level of *Wt1* mRNA was quantified by RT-qPCR, which was calculated with the following formula: Site m6A level (%) = $2^{Ct(\text{Input})-Ct(\text{MeRIP})} \times Fd \times 100\%$. Among them, Fd is the input dilution factor.

.....

18. Line 480, and throughout this section, how were the mice euthanised?

Response: Thanks for reviewer's question. For euthanasia, all mice were injected intraperitoneally with 2,2,2-tribromoethanol (250 mg/kg), and then cervical dislocation was executed under anesthesia.

The detailed revision is as follows,

In the revised Materials and Methods section (Lines 549-551)

.....In the different generations, 8 to 11 pregnant mice were obtained in each group after mating with different male mice, and the balances of sexes between pups (male: female, 3: 3) were performed on postnatal day (PND) 1. At PND28, one male mouse from each litter was selected. For euthanasia, all mice were injected intraperitoneally with 2,2,2-tribromoethanol (250 mg/kg), and then cervical dislocation was executed under anesthesia. This study was approved by the Laboratory Animal Ethics Committee of Anhui Medical University (ethical approval number: LLSC82273664).....

19. The sections on the AAV9 and STM2457 lack detail. where did the AAV9-wt1 construct come from? How much was used? How did they inject it into the testes, what was the surgical procedure? Was any anaesthetic and analgesia used? Did they inject into both testes? How did they test that it had incorporated into the testicular tissue? What was the control? These questions also apply to the STM2457 construct.

Response: Thanks for reviewer's questions. We apologize for missing

the details of the local testicular injection. Local testicular injections were performed according to previous study with corresponding modification [1]. In brief, the mice were injected intraperitoneally with 250 mg/kg 2,2,2-tribromoethanol. After anesthesia, both testes of the mice were found by touch pressure. Then, 20 μ L AAV9 or STM2457 were injected into the middle of each testes using a micropipette. After injection, the mice were placed flat in cage and observed for 30 min until they were fully awake and returned to normal. Among them, 1.03×10^{12} vg/ml AAV9-*Wt1* were injected once into mice at PND35. AAV9-NC was used as control. AAV9-*Wt1* and AAV9-NC were synthesized by General Biol Co., Ltd. STM2457 were injected at a dose of 50.0 mg/kg, once a week for 5 weeks. 20% SBE- β -CD was used as control. STM2457 and SBE- β -CD were purchased from MedChemExpress. The dose of STM2457 was chosen according to the previous study [2].

To determine whether the local injection of AAV9-*Wt1* can be integrated into the entire testicular tissue, a preliminary experiment was first performed. Results showed that after injection of the Ponceau S, the entire testis appeared red compared to the normal saline (Fig. S12). In addition, Figs. 2H and J-K also presented that the expression of WT1 in the testes was significantly upregulated after local testicular injection of AAV9-*Wt1*. These results suggested that local injection of AAV9-*Wt1* can be integrated into the entire testicular tissue.

Fig. S12

Figs. 2H and J-K

The detailed revision are as follows,

In the revised Materials and methods section (Lines 638-649):

Local testicular injection

Local testicular injections were performed according to previous study with corresponding modification⁶⁰. In brief, the mice were injected intraperitoneally with 250 mg/kg 2,2,2-tribromoethanol. After anesthesia, both testes of the mice were found by touch pressure. Then, 20 μ L AAV9 or STM2457 were injected into the middle of

each testes using a micropipette. After injection, the mice were placed flat in cage and observed for 30 min until they were fully awake and returned to normal. Among them, 1.03×10^{12} vg/ml AAV9-*Wt1* were injected once into mice at PND35. AAV9-NC was used as control. AAV9-*Wt1* and AAV9-NC were synthesized by General Biol Co., Ltd. STM2457 were injected at a dose of 50.0 mg/kg, once a week for 5 weeks. 20% SBE- β -CD was used as control. STM2457 and SBE- β -CD were purchase from MedChemExpress. The dose of STM2457 was choose according the previous study⁶¹.

[1] Wu H, Zhang X, Yang J, et al. Taurine and its transporter TAUT positively affect male reproduction and early embryo development. *Hum Reprod.* 2022;37(6):1229-1243.

[2] Yankova E, Blackaby W, Albertella M, et al. Small-molecule inhibition of METTL3 as a strategy against myeloid leukaemia. *Nature.* 2021;593(7860):597-601.

20. Did the authors normalise their MTT assay to cell number of protein content?

Response: Thanks for reviewer's question. It's an apology that we didn't normalize MTT assay to cell number of protein content. In this study, the main purpose of the MTT assay was to determine the dosage of Cd damage to Sertoli cells. During the assay, we performed standard procedures to ensure that all conditions, such as number of cells inoculated and incubating time, were consistent except for the Cd dose. Results showed that the cell viability decreased to 70% after exposure to 20 μ M CdCl₂ for 24 h, which was similar to previous study [1]. In addition, the expression of *Wt1* mRNA were obviously downregulated and the level of *Wt1* m6A were markedly increased in Sertoli cells at 24 h after 20 μ M CdCl₂ treatment. These results suggest that the phenotypes of Cd-damaged Sertoli cells have been established.

[1] Wang M, Wang XF, Li YM, et al. Cross-talk between autophagy and apoptosis regulates testicular injury/recovery induced by cadmium via PI3K with mTOR-independent pathway. *Cell Death Dis.* 2020;11(1):46.

21. For the RA ELISA, how did authors lyse the testis tissue and did they normalize for protein content?

Response: Thanks for reviewer's questions. To detect testicular RA, 50 mg of testes was accurately weighed and homogenized in 900 ml ice PBS with protease inhibitor. After centrifugation at 6000 g for 15 min, the supernatant was collected for RA detection. We normalized RA level using testicular weight, but not protein content. The concentration of testicular RA was presented as ng/g. More details for the assessment of RA levels have been added in the revised manuscript.

The detailed revision are as follows,

In the revised Materials and Method section (**Lines 697-700**):

Enzyme-linked immunosorbent assay (ELISA)

The quantity of retinoic acid (RA) in mouse testes and retinol in mouse serum were measured using the mouse RA-ELISA kit (MyBiosource, MBS706971) and the mouse retinol-ELISA kit (CUSABIO, CSB-E07891m), respectively. Assays were implemented according to the manufacturer's specifications. **To detect testicular RA, 50 mg of testes was accurately weighed and homogenized in 900 ml ice PBS with protease inhibitor. After centrifugation at 6000 g for 15 min, the supernatant was collected for RA detection. The concentration of testicular RA was presented as ng/g.**

22. How were the sperm obtained for the methylated RNA sequencing?

Response: Thanks for reviewer's question. To obtain sperm, fresh mouse epididymis was completely cut into pieces in preheated medium and incubated at 37°C for 10 min until mature sperm were fully released. Then, the tissue debris were removed by 40-µm cell strainer. The sperm mixture was incubated with somatic cell lysate (0.5% Triton X and 0.1% sodium dodecyl sulfate in nuclease-free water) on ice for 40 min to clear somatic cell. Next, the mixture was centrifuged at 600 g for 5 min and the precipitates were resuspended with PBS. After washing twice, sperm pellets were collected for the methylated RNA microarray. More details for sperm isolation have been added in the revised manuscript.

The detailed revision are as follows,

In the revised Materials and Method section (Lines 657-663):

Sperm isolation

To obtain sperm, fresh mouse epididymis was completely cut into pieces in preheated medium and incubated at 37°C for 10 min until mature sperm were fully released. Then, the tissue debris were removed by 40-µm cell strainer. The sperm mixture was incubated with somatic cell lysate (0.5% Triton X and 0.1% sodium dodecyl sulfate in nuclease-free water) on ice for 40 min to clear somatic cell. Next, the mixture was centrifuged at 600 g for 5 minutes and the precipitates were resuspended with PBS. After washing twice, sperm pellets were collected for the methylated RNA microarray.

23. Have the authors deposited their sequencing data into a publicly accessible repository?

Response: Thanks for reviewer's question. The m6A-mRNA&IncRNA Epitranscriptomic Microarray data (GSE241195) and RNA sequencing data (GSE241413) have been deposited in Gene Expression Omnibus with the accession number unmncaujhmxyp and afubgaoydjiftqd, respectively.

The detailed revision are as follows,

In the revised Data availability section (Lines 795-798):

Data availability

The m6A-mRNA&IncRNA Epitranscriptomic Microarray data (GSE241195) and RNA sequencing data (GSE241413) have been deposited in Gene Expression Omnibus with the accession number unmncaujhmxyp and afubgaoydjiftqd, respectively. Source data are provided with this paper.

24. What statistical approach was used for the analysis of the sequencing data?

Response: Thanks for reviewer's question. After StringTie comparison of the sequencing results to the known transcriptome, the transcriptional abundances at Gene Level and Transcript Level were calculated using Ballgown. The unit of expression quantity was expressed by FPKM (Fragments Per Kilobase of gene/transcript model

per Million mapped fragments). A gene or transcript with an expression value of FPKM > 0.5 was considered to be effectively expressed in the group. Then, Ballgown software was used for statistical analysis by the F-test statistical model, FC (Fold Change)>1.2 and P-value<0.05 were set as the screening threshold for differential expression.

The detailed revision are as follows,

In the revised Materials and Method section (Lines 726-734):

RNA sequencing

..... The end of generated fragment was sequenced using Illumina NovaSeq 6000 sequencer for 150 cycles. After StringTie comparison of the sequencing results to the known transcriptome, the transcriptional abundances at Gene Level and Transcript Level were calculated using Ballgown. The unit of expression quantity was expressed by FPKM (Fragments Per Kilobase of gene/transcript model per Million mapped fragments). A gene or transcript with an expression value of FPKM > 0.5 was considered to be effectively expressed in the group. Finally, Ballgown software was used for statistical analysis by the F-test statistical model, FC>1.2 and P-value<0.05 were set as the screening threshold for differential expression.

25. The details on the RNA-Seq is very scant and require a lot more information.

Response: Thanks for reviewer's comment and suggest. More details for the RNA-Seq have been added in the revised manuscript.

The detailed revision are as follows,

In the revised Materials and Method section (Lines 718-734):

RNA sequencing

Total RNA of mouse testes was extracted, and quality inspections were performed. The sequencing library was constructed with 2.0 µg total RNA from each sample at Aksomics (Shanghai, China). In brief, mRNA enrichment was performed using NEB Next® Poly(A) mRNA Magnetic Isolation Module kit. The library from enriched RNA was constructed by KAPA Stranded RNA-Seq Library Prep Kit (Illumina) and inspected by Agilent 2100 Bioanalyzer. Then, the final quantification of library was

conducted by qPCR. After denaturing with 0.1 M NaOH, 8 pM single stranded DNA was amplified in situ at the NovaSeq S4 Reagent Kit. The end of generated fragment was sequenced using Illumina NovaSeq 6000 sequencer for 150 cycles. After StringTie comparison of the sequencing results to the known transcriptome, the transcriptional abundances at Gene Level and Transcript Level were calculated using Ballgown. The unit of expression quantity was expressed by FPKM (Fragments Per Kilobase of gene/transcript model per Million mapped fragments). A gene or transcript with an expression value of FPKM > 0.5 was considered to be effectively expressed in the group. Finally, Ballgown software was used for statistical analysis by the F-test statistical model, FC>1.2 and P-value<0.05 were set as the screening threshold for differential expression.

26. For the MTT assay, what was the medium that the cells were grown in?

Response: Thanks for reviewer's question. Before adding to DMSO, the cells were grown in DMEM/F-12 medium and supplemented with 5% horse serum, 2.5% fetal bovine serum and 1% penicillin-streptomycin solution. More details for MTT assay have been added in the revised manuscript.

The detailed revision are as follows,

In the revised Materials and Method section (Lines 667-670):

MTT assay

First, TM4 cells were treated with different concentrations CdCl₂ for 6, 12, or 24 h after culture in 96 well plates for 12 h. Following that, the cells were treated with 5 mg/ml MTT, and were persistently cultured for 2 h in a 37 °C incubator. **Prior to this, the cells were grown in DMEM/F-12 medium and supplemented with 5% horse serum, 2.5% fetal bovine serum and 1% penicillin-streptomycin solution. After aspirating the medium,** DMSO was added to the culture plate. As a final step, the cellular absorbance at 570 nm was detected by a multimode reader (SYNERGY4, Biotek, USA).

27. For the determination of % RNA methylation by MeRIP qPCR, did the authors use a standard curve? If so this needs to be detailed.

Response: Thanks for reviewer's question and suggestion. The different site m6A level of *Wt1* mRNA were determined by RT-qPCR and following formula: Site m6A level (%) = $2^{Ct(\text{Input})-Ct(\text{MeRIP})} \times Fd \times 100\%$, but not a standard curve. Among them, Fd is the input dilution factor. For example, if 50 μL sample is used for MeRIP and 5 μL sample is used as input, the value of Fd is 1/10. The details of the MeRIP qPCR experiments have been added in the revised manuscript.

The detailed revision are as follows,

In the revised Materials and Method section (**Lines 761-763**):

MeRIP qPCR

The mRNAs were isolated as described previously..... The SRAMP website (<http://www.cuilab.cn/sramp/>) predicted that *Wt1* mRNA has four highly credible m6A-modified sites. The corresponding primer sequences are shown in Table S3. **The different site m6A level of *Wt1* mRNA was quantified by RT-qPCR, which was calculated with the following formula: Site m6A level (%) = $2^{Ct(\text{Input})-Ct(\text{MeRIP})} \times Fd \times 100\%$. Among them, Fd is the input dilution factor.**

28. There are no Materials and Methods details on any of the Western blotting experiments.

Response: Thanks for reviewer's comment. The details of the Western blotting experiments have been added in the revised manuscript.

The detailed revision are as follows,

In the revised Materials and Method section (**Lines 682-692**):

Immunoblotting

The protein lysates from TM4 cells and mouse testes were obtained after homogenizing with RIPA buffer, and Immunoblotting was implemented as previously described⁶². **Briefly, cellular and testicular protein extracts (6-30 μg) were separated by 10.0-12.5% SDS-PAGE electrophoresis and then transferred into PVDF membrane. After blocking with 5% skim milk for 1.5 h, the membranes were incubated with antibodies of DDX4 (1:1000), PLZF (1:200), C-KIT (1:200), SYCP3 (1:200), STRA8 (1:1000), RAR α**

(1:1000), RBP4 (1:1000), ALDH1A1 (1:1000), ALDH1A2 (1:1000), WT1 (1:1000), METTL3 (1:2000), METTL14 (1:1000), ALKBH5 (1:1000), FTO (1:1000), YTHDF1 (1:1000), YTHDF2 (1:1000) and IGF2BP1 (1:1000) for 1-3 h at room temperature. Antibody of β -Actin (1:10000) for loading control. After washing, the corresponding secondary antibody were used to incubate membranes for 1-2 h. Finally, the imaging system Bio-Rad ChemiDoc™ MP was used to detect immunoreactive protein signal.

If you have any questions, please feel free to contact us by E-mail (wanghuadev@ahmu.edu.cn).

Sincerely,

Hua Wang, PhD, MD

Department of Toxicology

Anhui Medical University

REVIEWER COMMENTS

Reviewer #1 (Remarks to the Author):

The authors have made extensive changes to the revised manuscript and added new data, including human data, in providing a detailed rebuttal.

The revised version has gone some way to addressing the concerns raised regarding the data and the statistical analysis. Despite these modifications, here are very few changes to the description of the data or the conclusions drawn.

I have several comments, as outlined below, which I hope will be useful.

Thanks for adding cadmium to the abstract –

Note, I would say 'on the susceptibility to cadmium (not of)

Regarding the regarding on statistical analysis, the figures now contain n for each group as well as ANOVA outcomes. It is good to see individual data points. The authors state that new experiments were undertaken.

'To further determine the statistical significance of our findings, additional repetitions were added for all animal experiments whose sample size n was 3. As shown in Figure 6, Figure 9 and other Figures, the sample size for Immunoblotting, RT-PCR, and MeRIP qPCR was increased from 3 to 6'

All of the western blot data appear the same as the original blots- so were new data added from separate blots to increase the n to 6 per group?. Did that require new western blots to be done, and re-analysis?

And,

Were new breeding experiments undertaken ? If so, how was the analysis of 2 cohorts handled.

Can the authors explain why in Figure2, panel C, contains data on DDX4 in the revised MS, and MVH in the original document -

The revision to the methods section details more information regarding the breeding process (new methods section, lines 542 -), which is helpful - the information in the methods section differs from that provided in the rebuttal ..

Rebuttal states:

'After mating, 10 pregnant mice were obtained in each group. Similarly, 10 F1 generation male mice (whose fathers were exposed to HFD) from different litters continued to be fed HFD for 10 weeks, and mated with normal females to breed F2 generation'

The MS version, line 600 mentions 8-11 mice; please check.

Could the authors check the description of data in Supp Figure 6?

For testicular site2Wt1 m6A level, the graph comprises n=3, with P=0.0791, so I am unclear why there is an asterisk on the Figure, NC versus HFD2D.

In the new Figure 10, BMI and sperm concentrations were analysed in donors, termed

control and case – presumably matched on age.

The description in the Figure 10 legend states
'After removal of smoking or alcohol drinking donors, a case-control study containing 30 pairs were obtained by matching body mass index (BMI), sperm concentration and age.

Presumably, only age was matched at the outset, and then BMI and sperm concentration were determined?

Please clarify.

Also, a series of relationships are depicted in 10D-F. The graphs show linear relationships – were other possibilities tested?

There is no description of the correlation analysis. It also appears that there are 2 distinct populations of sperm concentration in case vs controls- and thus there may be two distinct relationships for the comparison of sperm concentration and m6A level, shown in Fig 10F – the authors need to report on whether the relationship holds when just 'Case' subjects are analysed, and just 'Control' subjects.

OTHER suggestions

The description of the experiment is rather complex and I believe improvements to the language could be made, to help the reader's comprehension.

some examples-

1. It is unclear to me what the authors mean when they say in many of the figure legends: 'Similarly, partial F1 generation male mice continued to be fed HFD

Do you simply mean that a subset of F1 males were continued on HFD for 10 weeks ...

2. Throughout the MS, terms such as '.... multigenerational paternal HFD gradually enhances susceptibility to spermatogenesis disorder in offspring' where I believe the 'gradual' refers to the difference between F1 and F2 generations where 1 or 2 rounds of paternal HFD applied - gradually is not the right adjective here I believe-

Possibly progressively ?? incrementally ?? might be more accurate - throughout, as gradually infers an effect over time, but here you have 2 distinct generations, and offspring were only measured at 1 time point.

'Gradually' appears 33 times in the manuscript - it might be good to change this if the authors agree.

3. Line 86,
Instead of '..but not litter size',
'...with no effect on litter size'.

The interpretation of the result of Figure 3 could be improved.

Lines 176-178- could be changed to
'....multigenerational HFD progressively exacerbates the inhibition of synthesis of testicular

retinoic acid in offspring exposed to environmental stress' -

I think that is a more accurate description of the effect observed.

Regarding the role of role of Wt1 in the response, more detail has been provided, re the AAV9-Wt1 injection, re how the local testicular injection was performed, but it is not clear what the control injection is- was it scrambled AAV ? vehicle?

The labelling of Fig 5A schematic could be improved, to detail exactly who received AAV9-WT1- it looks as if all mice did (in 5A) but in fact only 2 groups did - could be made clearer in Fig 5A, and the legend –

NOTE - The legend states

'After injections of AAV9-Wt1 or NC, mice were exposed

Please clarify.

Perhaps AAV9-NC injection was used as control would be clearer in the methods description.

The expression of the legend could be improved – e.g.

At postnatal day 35 Wt1 was overexpressed in F1 mice by local testicular injection of,

Figure S2

For body weight, what does the asterisk refer to in Figure S2B - an overall effect of treatment ?

Line 262

'...on the level of testicular m6A in offspring is shown in Fig 6A.' (not was).

Methods section

'STM2457 and SBE- β -CD were purchase from MedChemExpress. The dose of STM2457 was choose according the previous study⁶¹'

Change to

STM2457 and SBE- β -CD were purchased from MedChemExpress. The dose of STM2457 was chosen according to previous research⁶¹

Line 438

Overweight/obesity donors ...

Perhaps ' ...donors who were overweight or obese'.

Line 460-61

....Based on these findings indicate that mutigenerational

Maybe change this to -

Taken together these findings suggest that

Note, other editorial input of the language is required.

Reviewer #2 (Remarks to the Author):

The authors have addressed all my comments. I have no further comments to make.

Manuscript No.: NCOMMS-23-17775B

Title: Multigenerational paternal obesity enhances the susceptibility to poor semen quality in male offspring via *Wt1* m6A modification

Dear reviewers,

Thank you for your letter. According to reviewers' comments and questions, the detailed revisions are as follows,

Reviewer comments:

Reviewer #1(Remarks to the Author):

The authors have made extensive changes to the revised manuscript and added new data, including human data, in providing a detailed rebuttal. The revised version has gone some way to addressing the concerns raised regarding the data and the statistical analysis. Despite these modifications, here are very few changes to the description of the data or the conclusions drawn. I have several comments, as outlined below, which I hope will be useful.

1. Thanks for adding cadmium to the abstract – Note, I would say ‘on the susceptibility to cadmium (not of)

Response: Thanks for reviewer's kind correction. The phrase "on the susceptibility of cadmium" has been corrected as "on the susceptibility to cadmium" in the revised Abstract section.

The detailed revision is as follows,

In the revised Abstract section (Line 23)

There is strong evidence that obesity is a risk factor for poor semen quality. However, the effects of multigenerational paternal obesity on the susceptibility to cadmium (a reproductive toxicant)-induced spermatogenesis disorders in offspring remain unknown.....

2. Regarding the regarding on statistical analysis, the figures now contain n for each group as well as ANOVA outcomes. It is good to see individual data points. The authors state that new experiments were undertaken. 'To further determine the statistical significance of our findings, additional repetitions were added for all animal experiments whose sample size n was 3. As shown in Figure 6, Figure 9 and other Figures, the sample size for Immunoblotting, RT-PCR, and MeRIP qPCR was increased from 3 to 6'. All of the western blot data appear the same as the original blots- so were new data added from separate blots to increase the n to 6 per group? Did that require new western blots to be done, and re-analysis? And, were new breeding experiments undertaken? If so, how was the analysis of 2 cohorts handled.

Response: Thanks for reviewer's questions. In the supplementary experiment, additional 3 different mouse testicular tissues were homogenized to extract protein and RNA. Among them, original and supplementary blots data were obtained from 6 different mouse testicular lysate. Hence, we added new western blots to increase the n to 6 per group, and conducted a re-analysis. In order to control quality, we performed standard procedures to ensure that all conditions, such as experimenter, reagents and antibodies incubation, were consistent. As shown in the figures below, the analysis of the original and supplementary blots showed that the data distribution was slightly different, the overall trend remained unchanged (more information were available in Fig. 6 and Fig. 9). Therefore, no significant changes have been made in the blot pictures and the description of the results in the revised manuscript. In our previous animal experiments, about 10 offspring mice were obtained from different litters each group. The samples per group were sufficient to meet the demand, so no new breeding experiments were carried out.

In the Figures section (Lines 99-449)

The original blots in Figs. 6B and C

B

C

The supplementary blots in Figs. 6B and C

B

C

The analysis of blots in Figs. 6B and C

Blots	Analysis of METTL3 (mean ± SEM)			
	NC	NCD	HFD1D	HFD2D
Original	1.00 ± 0.09	1.34 ± 0.08	2.02 ± 0.11	3.19 ± 0.14
Supplementary	1.00 ± 0.14	1.65 ± 0.05	1.98 ± 0.18	3.00 ± 0.08
Total	1.00 ± 0.08	1.50 ± 0.08	2.00 ± 0.08	3.09 ± 0.08

The original blots in Figs. 6E and F

E

F

The supplementary blots in Figs. 6E and F

E

F

The analysis of blots in Figs. 6E and F

Blots	Analysis of YTHDF1 (mean ± SEM)			
	NC	NCD	HFD1D	HFD2D
Original	1.00 ± 0.03	0.80 ± 0.04	0.50 ± 0.05	0.19 ± 0.11
Supplementary	1.00 ± 0.13	0.40 ± 0.09	0.31 ± 0.05	0.09 ± 0.01
Total	1.00 ± 0.06	0.60 ± 0.10	0.40 ± 0.05	0.14 ± 0.05

Blots	Analysis of YTHDF2 (mean ± SEM)			
	NC	NCD	HFD1D	HFD2D
Original	1.00 ± 0.15	1.49 ± 0.05	3.43 ± 0.76	3.97 ± 0.43
Supplementary	1.00 ± 0.15	2.69 ± 0.37	3.29 ± 0.20	4.94 ± 0.64
Total	1.00 ± 0.09	2.09 ± 0.32	3.36 ± 0.35	4.45 ± 0.41

Blots	Analysis of IGF2BP1 (mean ± SEM)			
	NC	NCD	HFD1D	HFD2D
Original	1.00 ± 0.17	0.75 ± 0.05	0.54 ± 0.07	0.14 ± 0.04
Supplementary	1.00 ± 0.21	0.54 ± 0.13	0.26 ± 0.06	0.08 ± 0.01
Total	1.00 ± 0.11	0.64 ± 0.08	0.40 ± 0.08	0.11 ± 0.02

The original blots in Figs. 9D and E

The supplementary blots in Figs. 9D and E

The analysis of blots in in Figs. 9D and E

Blots	Analysis of WT1 (mean ± SEM)			
	NCD	NCD+STM	HFD1D	HFD1D+STM
Original	1.00±0.14	1.32±0.14	0.53±0.08	1.27±0.25
Supplementary	1.00±0.09	1.25±0.15	0.52±0.03	0.71±0.05
Total	1.00±0.07	1.28±0.09	0.53±0.04	0.99±0.17

The original blots in Figs. 9H and I

The supplementary blots in Figs. 9H and I

The analysis of blots in in Figs. 9H and I

3. Can the authors explain why in Figure2, panel C, contains data on DDX4 in the revised MS, and MVH in the original document -

Response: Thanks for reviewer's question. Based on another reviewer's suggestion about "I believe that the more accurate name for MVH (line 105) is Ddx4" and previous literature [1, 2], we also agree that a more accurate name for Anti-DDX4/MVH antibody is DDX4. Therefore, the word of "MVH" has been corrected as "DDX4" in the revised manuscript.

[1] Yu S, Zhou C, He J, et al. BMP4 drives primed to naïve transition through PGC-like state. Nat

Commun. 2022;13(1):2756.

[2] Zhao J, Lu P, Wan C, et al. Cell-fate transition and determination analysis of mouse male germ cells throughout development. Nat Commun. 2021;12(1):6839.

4. The revision to the methods section details more information regarding the breeding process (new methods section, lines 542 -), which is helpful - the information in the methods section differs from that provided in the rebuttal. Rebuttal states: 'After mating, 10 pregnant mice were obtained in each group. Similarly, 10 F1 generation male mice (whose fathers were exposed to HFD) from different litters continued to be fed HFD for 10 weeks, and mated with normal females to breed F2 generation' The MS version, line 600 mentions 8-11 mice; please check.

Response: Thanks for reviewer's comment and suggestion. Upon careful examination, the information in the methods section does not contradict from that provided in the rebuttal. In this study, we conducted three different animal experiments. In experiment 1 there were 10 mice in each group, and in experiment 3 there were 8-11 mice in each group.

Experiment 1. The experimental design was used to investigate the susceptibility to spermatogenesis disorders in the offspring of fathers stressed by multigenerational HFD. After mating, 10 pregnant mice were obtained in each group. **Experiment 3.** The experimental design was used to investigate the effect of reduced sperm m6A level on paternal HFD-inhibited testicular WT1 expression and spermatogenesis in offspring. After mating, 8-11 pregnant mice were obtained in each group. Therefore, 8 to 11 pregnant mice were obtained in each group after mating with different male mice in animal treatments.

5. Could the authors check the description of data in Supp Figure 6? For testicular site2 Wt1 m6A level, the graph comprises n=3, with P=0.0791, so I am unclear why there is an asterisk on the Figure, NC versus HFD2D.

Response: Thanks for reviewer's questions and correction. We are

sorry for the misunderstanding caused by our mismarking. After careful checking, the asterisk that should have been marked on supplementary Figure 6B ($F=7.97$ and $P=0.0087$ for Site3) was mistakenly marked on supplementary Figure 6A ($F=3.29$ and $P=0.0791$ for Site2). In fact, no effect was observed on the description of results. In the revised manuscript, we carefully checked the statistical data and made corrections.

The detailed revision are as follows,

In the revised Fig. S6 section (Lines 57-58)

Fig.S6. Effect of multigenerational paternal HFD on testicular *Wt1* Site2-4 m6A

modification in offspring. (A-C) Testicular *Wt1* site 2, 3 and 4 m6A level were detected by MeRIP qPCR. $n=3$, $DOF=11$, $F=3.29$ and $P=0.0791$ for *Site2*; $F=7.97$ and $P=0.0087$ for *Site3*; $F=4.35$ and $P=0.0428$ for *Site4*. n.s., not significant. $*P < 0.05$.

6. In the new Figure 10, BMI and sperm concentrations were analyzed in donors, termed control and case – presumably matched on age. The description in the Figure 10 legend states ‘After removal of smoking or alcohol drinking donors, a case-control study containing 30 pairs were obtained by matching body mass index (BMI), sperm concentration and age. Presumably, only age was matched at the outset, and then BMI and sperm concentration were determined? Please clarify. Also, a series of relationships are depicted in 10D-F. The graphs show linear relationships – were other possibilities tested? There is no description of the correlation analysis. It also appears that there are 2 distinct populations of

sperm concentration in case vs controls- and thus there may be two distinct relationships for the comparison of sperm concentration and m6A level, shown in Fig 10F – the authors need to report on whether the relationship holds when just ‘Case’ subjects are analyzed, and just ‘Control’ subjects.

Response: Thanks for reviewer's questions and suggestions. To establish a case-control study, a total of 428 human sperm were obtained from the reproductive medicine center of the First Affiliated Hospital of Anhui Medical University with the donor's informed consent. After removal of smoking or alcohol drinking donors, 168 sperm samples were available. Finally, 30 pairs of overweight/obesity cases and corresponding controls were obtained by matching age. Sperm m6A levels were measured in this case-control study. As shown in Fig. 10, elevated sperm m6A level and decreased sperm concentration is observed in donors who were overweight or obese.

In addition, restricted cubic splines were used to describe the nonlinear correlation among sperm concentration, BMI and sperm m6A level. As shown in Figs. S11A-C, a linear relationship was observed between the three sets of data (P for overall <0.001 , and P for nonlinear >0.05). Further analysis found that sperm m6A levels were negatively correlated with sperm concentration in both the case and control groups (Figs. S11D and E). Therefore, the negative correlation between sperm m6A level and sperm concentration was also remained even if the case group or the control group were analyzed separately.

The detailed revision are as follows,

In the revised Results section (Lines 435-445)

Elevated sperm m6A level and decreased sperm concentration is observed in donors who were overweight or obese

.....The positive association was observed between sperm m6A level and BMI ($r = 0.57$, $P < 0.01$; Fig. 10E). The negative correlation was also observed between sperm

m6A level and sperm concentration ($r=-0.51$, $P < 0.01$; Fig. 10F). In addition, restricted cubic splines were used to describe the nonlinear correlation among sperm concentration, BMI and sperm m6A level. As shown in Figs. S11A-C, a linear relationship was observed between the three sets of data (P for overall < 0.001 , and P for nonlinear > 0.05). The above results suggest that elevated sperm m6A level and decreased sperm concentration is observed in donors who were overweight or obese.

Fig.10. Elevated sperm m6A level and decreased sperm concentration is observed in donors who were overweight or obese. After removal of smoking or alcohol drinking donors, 30 pairs of overweight/obesity cases and corresponding controls were obtained by matching age. (A) BMI. $n=30$, $t=-11.20$, $P < 0.0001$. (B) Sperm concentration. $n=30$, $t=10.50$, $P < 0.0001$

In the revised Materials and methods section (Lines 626-628)

Case-control study

To investigate the effect of overweight/obesity on sperm m6A levels, a case-control study was established. A total of 428 human sperm were obtained from the reproductive medicine center of the First Affiliated Hospital of Anhui Medical University with the donor's informed consent. After removal of smoking or alcohol drinking donors, 168 sperm samples were available. Finally, 30 pairs of overweight/obesity cases and corresponding controls were obtained by matching age. Semen specimens were centrifuged at 600 g for 10 min to remove seminal plasma.

In the revised Fig. S11 section (Lines 130-138)

Fig.S11. The association among sperm concentration, BMI and sperm m6A level

were analyzed. After removal of smoking or alcohol drinking donors, 30 pairs of overweight/obesity cases and corresponding controls were obtained by matching age. (A-C) Restricted cubic splines were used to describe the nonlinear correlation among sperm concentration, BMI and sperm m6A level. (D and E) The linear correlation analysis of sperm concentration and sperm m6A level was performed in both the case and control groups, respectively.

OTHER suggestions

The description of the experiment is rather complex and I believe improvements to the language could be made, to help the reader's comprehension. some examples-

1. It is unclear to me what the authors mean when they say in many of the figure legends: 'Similarly, partial F1 generation male mice continued to be fed HFD'. Do you simply mean that a subset of F1 males were continued on HFD for 10 weeks ...

Response: Thanks for reviewer's comment and question. The sentence "partial F1 generation male mice continued to be fed HFD for 10 weeks" mean that a subset of the males in F1 generation were continued to be treated with HFD, while others were exposed to Cd.

The detailed revision are as follows,

In the revised Figure legends section (Lines 103-287)

Fig.2. Multigenerational paternal HFD exacerbates environmental stress-impaired testicular germ cell development in offspring. F0 generation male mice were fed NC or HFD from 5 weeks to 15 weeks old, and then mated with normal female to breed F1 generation. Similarly, a subset of the males in F1 generation were continued to be treated with HFD for 10 weeks, and mated with normal female to breed F2 generation.....

2. Throughout the MS, terms such as '.... multigenerational paternal HFD gradually enhances susceptibility to spermatogenesis disorder in offspring' where I believe the 'gradual' refers to the difference between F1 and F2

generations where 1 or 2 rounds of paternal HFD applied - gradually is not the right adjective here I believe- Possibly progressively ?? incrementally ?? might be more accurate - throughout, as gradually infers an effect over time, but here you have 2 distinct generations, and offspring were only measured at 1 time point. 'Gradually' appears 33 times in the manuscript - it might be good to change this if the authors agree.

Response: Thanks for reviewer's kind suggestion. We agree with you. In the revised manuscript, the word of "gradually" has been replaced with "progressively".

The detailed revision are as follows,

In the revised Abstract section (Lines 26-30)

.....Here, results showed that sperm count, testicular germ cell marker and retinoic acid (RA) levels were progressively reduced in offspring with the increase of high-fat diet (HFD) generation. Furthermore, we identified that WT1, RA synthetases upstream transcription factor, was decreased over generations. The impacts mentioned above were restored by injecting AAV9-*Wt1* in the testes of the offspring. Additionally, multigenerational paternal HFD progressively increased methylase METTL3 and *Wt1* m6A levels in offspring testes.....

In the revised Results section (Lines 77-370)

Multigenerational paternal HFD progressively enhances susceptibility to spermatogenesis disorder in their offspring

.....Additionally, testicular HE staining showed that the gradual reductions of mature seminiferous tubules were observed in offspring with the increase of HFD generation (Figs. 1G and H). The above results indicate that multigenerational paternal HFD progressively enhances the susceptibility to spermatogenesis disorder in their offspring.

In the revised Discussion section (Lines 455-536)

.....The present study revealed that fertility rate and sperm count were progressively reduced in the offspring with the increase of HFD generation. During successive

generations of HFD with Cd stress, offspring testes were progressively depleted of differentiating spermatogonia, spermatocytes, round spermatids, and elongated spermatids.....

3. Line 86, Instead of ‘..but not litter size’, ‘...with no effect on litter size’.

Response: Thanks for reviewer's kind correction. The phrase "but not litter size" has been corrected as "with no effect on litter size" in the revised Results section.

The detailed revision is as follows,

In the revised Results section (Line 86)

.....As presented in Fig. 1A-C and Fig. S2A, the gradual reductions of fertility rate were observed in mice with the increase of HFD generation, with no effect on litter size. The effect of paternal HFD exposure on the susceptibility of environmental stress-induced spermatogenesis impairment in offspring was also explored.....

4. The interpretation of the result of Figure 3 could be improved. Lines 176-178- could be changed to ‘...multigenerational HFD progressively exacerbates the inhibition of synthesis of testicular retinoic acid in offspring exposed to environmental stress’ - I think that is a more accurate description of the effect observed.

Response: Thanks for reviewer's comment and kind suggestion. The sentence "multigenerational HFD gradually exacerbates the synthesis of testicular retinoic acid in the offspring inhibited by environmental stress" has been changed to "multigenerational HFD progressively exacerbates the inhibition of synthesis of testicular retinoic acid in offspring exposed to environmental stress" in the revised Results section.

The detailed revision is as follows,

In the revised Results section (Lines 178-180)

.....Similarly, the gradual downregulation in the expression of ALDH1A1 and ALDH1A2 proteins was observed in offspring testes with the increase of HFD

generation (Figs. 3G-I). The compilation of these studies suggests that multigenerational HFD progressively exacerbates the inhibition of synthesis of testicular retinoic acid in offspring exposed to environmental stress.

5. Regarding the role of *Wt1* in the response, more detail has been provided, re the AAV9-*Wt1* injection, re how the local testicular injection was performed, but it is not clear what the control injection is- was it scrambled AAV? vehicle?

Response: Thanks for reviewer's comment and question. In the *Wt1* overexpression experiment, the control was injected with vehicle. More details have been added to the revised manuscript.

The detailed revision are as follows,

In the revised Materials and methods section (Lines 651-652)

.....After injection, the mice were placed flat in cage and observed for 30 min until they were fully awake and returned to normal. Among them, 1.03×10^{12} vg/ml AAV9-*Wt1* were injected once into mice at PND35. The control was injected with vehicle. AAV9-*Wt1* and vehicle were synthesized by General Biol Co., Ltd.....

6. The labelling of Fig 5A schematic could be improved, to detail exactly who received AAV9-WT1- it looks as if all mice did (in 5A) but in fact only 2 groups did - could be made clearer in Fig 5A, and the legend –

Response: Thanks for reviewer's suggestions. In the revised manuscript, we improved Figure 5A to exactly state which group of mice received AAV9-*Wt1* or vehicle treatment. In addition, detailed annotations have been added to the revised Figure legends.

The detailed revision are as follows,

In the revised Figure and Figure legends section (Lines 245-251)

Fig.5. Paternal HFD aggravates environmental stress-impaired testicular spermatogenesis via inhibiting WT1-mediated retinoic acid synthesis in offspring. F0 generation male mice were fed NC or HFD from 5 weeks to 15 weeks old, and mated with normal female to breed F1 generation. At postnatal day 35, *Wt1* was overexpressed in F1 mice by local testicular injection of adeno-associated virus 9 (AAV9). After injection of AAV9-*Wt1* or vehicle, mice were exposed to CdCl₂ for 10 weeks, and

named NCD, NCD+WT1, HFD1D or HFD1D+WT1 group, respectively.

7. NOTE - The legend states 'After injections of AAV9-Wt1 or NC, mice were exposed

Please clarify. Perhaps AAV9-NC injection was used as control would be clearer in the methods description.

Response: Thanks for reviewer's comment and suggestion. In the revised manuscript, we annotated that vehicle injection was used as control.

The detailed revision are as follows,

In the revised Figure legends section (Line 251)

.....At postnatal day 35, Wt1 was overexpressed in F1 mice by local testicular injection of adeno-associated virus 9 (AAV9). After injection of AAV9-Wt1 or vehicle, mice were exposed to CdCl₂ for 10 weeks, and named NCD, NCD+WT1, HFD1D or HFD1D+WT1 group, respectively.....

In the revised Materials and methods section (Lines 652-653)

.....Among them, 1.03×10¹² vg/ml AAV9-Wt1 were injected once into mice at PND35. The control was injected with vehicle. AAV9-Wt1 and vehicle were synthesized by General Biol Co., Ltd.....

8. The expression of the legend could be improved – e.g. At postnatal day 35 Wt1 was overexpressed in F1 mice by local testicular injection of,

Response: Thanks for reviewer's kind suggestion. The sentence "At postnatal day (PND) 35, F1 generation mice overexpressed WT1 by testicular local injection of adeno-associated virus 9 (AAV9)" has been changed to "At postnatal day 35, Wt1 was overexpressed in F1 mice by local testicular injection of adeno-associated virus 9 (AAV9)" in the revised Figure legends section.

The detailed revision is as follows,

In the revised Figure legends section (Lines 249-251)

.....F0 generation male mice were fed NC or HFD from 5 weeks to 15 weeks old,

and mated with normal female to breed F1 generation. At postnatal day 35, Wt1 was overexpressed in F1 mice by local testicular injection of adeno-associated virus 9 (AAV9). After injection of AAV9-Wt1 or NC, mice were exposed to CdCl₂ for 10 weeks, and named NCD, NCD+WT1, HFD1D or HFD1D+WT1 group, respectively.....

9. Figure S2 For body weight, what does the asterisk refer to in Figure S2B - an overall effect of treatment?

Response: Thanks for reviewer's question. In Figure S2, the asterisk represented the overall effect of weight change in mice treated with HFD and HFD1+HFD. The corresponding details have been added to the revised manuscript.

The detailed revision are as follows,

In the revised Supplementary Materials section (Line 18)

Fig.S2. The effect of multigenerational paternal HFD on litter size and body weight in offspring. (A) Litter size. $n=10$, $DOF=29$, $F=0.48$, $P=0.6246$. (B) Body weight of male. $n = 10$, $F=31.45$, $P<0.0001$. * $P < 0.05$.

10. Line 262 ‘...on the level of testicular m6A in offspring is shown in Fig 6A.’ (not was).

Response: Thanks for reviewer's correction. The word "was" has

been corrected as "is" in the revised results section.

The detailed revision is as follows,

In the revised results section (Line 264)

.....The effect of paternal HFD exposure on the level of testicular m6A in offspring **is** shown in Fig. 6A.....

11. Methods section ' STM2457 and SBE- β -CD were purchase from MedChemExpress. The dose of STM2457 was choose according the previous study⁶¹' Change to STM2457 and SBE- β -CD were purchased from MedChemExpress. The dose of STM2457 was chosen according to previous research⁶¹

Response: Thanks for reviewer's kind corrections. The sentences "STM2457 and SBE- β -CD were purchase from MedChemExpress. The dose of STM2457 was choose according the previous study" have been changed to "STM2457 and SBE- β -CD were purchased from MedChemExpress. The dose of STM2457 was chosen according to previous research" in the revised Materials and Methods section.

The detailed revision are as follows,

In the revised Materials and Methods section (Lines 655-657)

.....STM2457 were injected at a dose of 50.0 mg/kg, once a week for 5 weeks. 20% SBE- β -CD was used as control. STM2457 and SBE- β -CD were purchased from MedChemExpress. The dose of STM2457 was chosen according to previous research⁶¹.

12. Line 438 Overweight/obesity donors ...Perhaps ' ...donors who were overweight or obese'.

Response: Thanks for reviewer's suggestion. The sentence "Elevated sperm m6A level and decreased sperm concentration is observed in overweight/obesity donors" has been changed to "Elevated sperm m6A level and decreased sperm concentration is observed in donors who were

overweight or obese" in the revised Results and Figure legends section.

The detailed revision are as follows,

In the revised Results section (Lines 419-420)

Elevated sperm m6A level and decreased sperm concentration is observed in donors who were overweight or obese

The effects of paternal single- or double-generation HFD exposure on sperm and testicular parameters, retinoic acid level, WT1 and METTL3 expression in offspring mice were investigated.....

In the revised Figure legends section (Lines 442-443)

Fig.10. Elevated sperm m6A level and decreased sperm concentration is observed in donors who were overweight or obese. After removal of smoking or alcohol drinking donors, a case-control study containing 30 pairs were obtained by matching body mass index (BMI), sperm concentration and age.....

13. Line 460-61 ...Based on these findings indicate that mutigenerational Maybe change this to -Taken together these findings suggest that

Response: Thanks for reviewer's suggestion. The sentence "Based on these findings indicate that....." has been changed to "Taken together these findings suggest that....." in the revised Discussion section.

The detailed revision is as follows,

In the revised Discussion section (Lines 465-466)

.....In line with our findings, previous study found that paternal exposure to HFD decreased sperm count and motility in F1 offspring⁹. **Taken together these findings suggest that** multigenerational paternal HFD enhances susceptibility to testicular spermatogenesis disorder and germ cells impairment in their offspring.

14. Note, other editorial input of the language is required.

Response: Thanks for reviewer's suggestion. The entire manuscript has been fully checked and inappropriate language have been corrected.

The detailed revision are as follows,

In the revised Introduction section (Lines 45-47)

.....Paternal Origins of Health and Disease (POHaD) theory manifests that paternal exposure to adverse factors leads to the occurrence and development of adult offspring with chronic diseases^{7, 8}. Studies showed that paternal obesity impaired the structure of testicular seminiferous tubules and reduced the number of epididymal sperm in offspring^{9, 10}. Based on the above studies, paternal obesity may lead to spermatogenesis disorders in offspring. Recent study found that parental obesity intergenerationally induced reproductive damages in offspring¹¹.....

In the revised Results section (Lines 119-440)

Multigenerational paternal HFD progressively exacerbates environmental stress-impaired testicular germ cell development in offspring

It is well known that spermatogenesis is determined by the development of testicular germ cells. Compared to the NCD group, testicular weight and DDX4 (marker of testicular germ cell) protein level were obviously lowered in the HFD1D group, which was further reduced in the HFD2D group (Figs. 2A-C). Furthermore, the mRNA levels of *Izumo3* (elongated spermatids marker), *Acrv1* (round spermatids marker), *Smc3* (spermatocytes marker) and *C-kit* (differentiating-spermatogonia marker) were reduced in HFD1D and HFD2D groups compared to NCD group, and *Smc3*, *Acrv1* and *Izumo3* mRNA levels were further decreased in HFD2D groups compared to HFD1D group (Fig. 2D). Correspondingly, the C-KIT and SYCP3 (spermatocytes marker) protein expressions were downregulated in the HFD1D and HFD2D groups compared with that of NCD group, and SYCP3 expression was persistently downregulated in the HFD2D group compared to the HFD1D group (Figs. 2E and F).

Multigenerational paternal HFD progressively aggravates environmental stress-inhibited testicular retinoic acid synthesis in offspring

..... The retinol metabolic-related genes were presented in Fig. S3D. Results showed that the expressions of *Aldh1a1*, *Aldh1a2*, *Aldh1a3*, *Rara* and *Stra8* were downregulated in the HFD2D group compared with the HFD1D group. Nevertheless,

the levels of serum vitamin A (retinol) and testicular retinol-binding protein 4 (RBP4) were not reduced among the four groups (Figs. S4A-C).....

In the revised Materials and methods section (Lines 622-799)

Case-control study

To investigate the effect of overweight/obesity on sperm m6A levels, a case-control study was established. A total of 428 human sperm were obtained from the reproductive medicine center of the First Affiliated Hospital of Anhui Medical University with the donor's informed consent. After removal of smoking or alcohol drinking donors, 168 sperm samples were available. Finally, 30 pairs of overweight/obesity cases and corresponding controls were obtained by matching age. Semen specimens were centrifuged at 600 g for 10 min to remove seminal plasma. After PBS resuspension, sperm were incubated in somatic cell lysate (0.5% Triton X and 0.1% sodium dodecyl sulfate in nuclease-free water) at 4 °C for 10 min to remove somatic cells. After washing, sperm pellets were collected for m6A detection.

.....

Statistical analysis

..... Mean comparisons between the two groups were performed using two-tailed *Student's t-test*. Repeated-measures *ANOVA* was applied to analyze the GTT data. Two-way *ANOVA* was applied to analyze RNA stability assay and mouse body weight data. $P < 0.05$ meant the difference was statistically significant.

If you have any questions, please feel free to contact us by E-mail (wanghuadev@ahmu.edu.cn).

Sincerely,

Hua Wang, PhD, MD

Department of Toxicology

Anhui Medical University

REVIEWERS' COMMENTS

Reviewer #1 (Remarks to the Author):

Thanks for making those changes to the manuscript, and for clarifying the analysis. I think the paper is much clearer now.

There is one small clarification remaining.

The labelling of Supp Fig3 departs a bit from other figures - the legend says:

'Male mice of F1 and F2 generations were exposed to CdCl₂ (0 or 100 mg/L) by drinking water for 10 weeks, and named NC, NCD, HFD1D or HFD2D group respectively' –

But the labels on graph read HFD1D1, HFD2D1 etc

Manuscript No.: NCOMMS-23-17775C

Title: Multigenerational paternal obesity enhances the susceptibility to male subfertility in offspring via *Wt1* m6A modification

Dear reviewers,

Thank you for your letter. According to reviewers' comments and questions, the detailed revisions are as follows,

Reviewer comments:

Reviewer #1(Remarks to the Author):

Thanks for making those changes to the manuscript, and for clarifying the analysis. I think the paper is much clearer now. There is one small clarification remaining. The labelling of Supp Fig3 departs a bit from other figures - the legend says: 'Male mice of F1 and F2 generations were exposed to CdCl₂ (0 or 100 mg/L) by drinking water for 10 weeks, and named NC, NCD, HFD1D or HFD2D group respectively' –But the labels on graph read HFD1D1, HFD2D1 etc.

Response: Thanks for reviewer's kind correction. In supplementary Figure 3, we presented the results of testicular RNA sequencing in the HFD1D and HFD2D groups. Therefore, the description of NC and NCD groups in the figure legend was redundant. The sentence "Male mice of F1 and F2 generations were exposed to CdCl₂ (0 or 100 mg/L) by drinking water for 10 weeks, and named NC, NCD, HFD1D or HFD2D group respectively" has been corrected as "Male mice of F1 and F2 generations were exposed to CdCl₂ (0 or 100 mg/L) by drinking water for 10 weeks, and named HFD1D or HFD2D group respectively" in the revised supplementary Figure 3 section.

The detailed revision is as follows,

In the revised supplementary Figure 3 section (Lines 27-28)

.....Male mice of F1 and F2 generations were exposed to CdCl₂ (0 or 100 mg/L) by drinking water for 10 weeks, and named HFD1D or HFD2D group respectively. Testes of HFD1D and HFD2D groups were collected for RNA-seq.....

If you have any questions, please feel free to contact us by E-mail (wanghuadev@ahmu.edu.cn).

Sincerely,

Hua Wang, PhD, MD

Department of Toxicology

Anhui Medical University